# On The Concurrence of Layer-wise Preconditioning Methods and Provable Feature Learning

**Thomas T. Zhang** [* 1]  **Behrad Moniri** [* 1]  **Ansh Nagwekar** [1]  **Faraz Rahman** [1]  **Anton Xue** [1]
**Hamed Hassani** [1]  **Nikolai Matni** [1]

## Abstract

Layer-wise preconditioning methods are a family of memory-efficient optimization algorithms that introduce preconditioners per axis of each layer's weight tensors. These methods have seen a recent resurgence, demonstrating impressive performance relative to entry-wise ("diagonal") preconditioning methods such as Adam(W) on a wide range of neural network optimization tasks. Complementary to their practical performance, we demonstrate that layer-wise preconditioning methods are provably necessary from a statistical perspective. To showcase this, we consider two prototypical models, *linear representation learning* and *single-index learning*, which are widely used to study how typical algorithms efficiently learn useful *features* to enable generalization. In these problems, we show SGD is a suboptimal feature learner when extending beyond ideal isotropic inputs $\mathbf{x} \sim \mathsf{N}(\mathbf{0}, \mathbf{I})$ and well-conditioned settings typically assumed in prior work. We demonstrate theoretically and numerically that this suboptimality is fundamental, and that layer-wise preconditioning emerges naturally as the solution. We further show that standard tools like Adam preconditioning and batch-norm only mildly mitigate these issues, supporting the unique benefits of layer-wise preconditioning.

## 1. Introduction

Well-designed optimization algorithms have been an enabler to the staggering growth and success of machine learning. For the broader ML community, the Adam (Kingma & Ba, 2015) optimizer is likely the go-to scalable and performant choice for most tasks. However, despite its popularity in practice, it has been notoriously challenging to understand

Adam-like optimizers theoretically, especially from a *statistical* (e.g. generalization) perspective. [1] In fact, there exist many theoretical settings where Adam and similar methods underperform in convergence or generalization relative to, e.g., well-tuned SGD (see e.g. Wilson et al. (2017); Keskar & Socher (2017); Reddi et al. (2018); Gupta et al. (2021); Xie et al. (2022); Dereich et al. (2024)), further complicating a principled understanding the role of Adam-like optimizers in deep learning. Given these challenges, is there an alternative algorithmic paradigm that is comparable to the Adam family in practice that is also well-motivated from a statistical learning perspective? Encouragingly, in a recent large-scale deep learning optimization competition, AlgoPerf (MLCommons, 2024), Adam and its variants were outperformed in various "hold-out error per unit-compute"[2] metrics by a method known as Shampoo (Gupta et al., 2018), a member of a layer-wise "Kronecker-Factored" family of preconditioners, formally described in Section 2, contrasted with "diagonal" preconditioning methods like Adam.

Notable members of the Kronecker-Factored preconditioning family include Shampoo and KFAC (Martens & Grosse, 2015), as well as their many variants and descendants. These algorithms are motivated from an approximation-theoretic perspective, aiming to approximate some *ideal* curvature matrix (e.g. the Hessian or Fisher Information) in a way that mitigates the computational and memory challenges associated with second-order algorithms such as Newton's Method (NM) or Natural Gradient Descent (NGD). However, towards establishing the benefit of these preconditioners, an approximation viewpoint is bottlenecked by our limited understanding of how the *idealized* second-order methods perform on neural-network learning tasks, even disregarding the computational considerations. It in fact remains unclear whether these second-order methods are inherently superior to the approximations designed to emulate them. For example, recent work has shown that, surprisingly, KFAC generally *outperforms* its ideal counterpart NGD in convergence rate and generalization error on typical deep learning tasks (Benzing, 2022). Thus, a key question remains:

---
[*]Equal contribution  [1]University of Pennsylvania. Correspondence to: T. Zhang, B. Moniri <{ttz2, bemoniri}@seas.upenn.edu>.

*Proceedings of the 42nd International Conference on Machine Learning*, Vancouver, Canada. PMLR 267, 2025. Copyright 2025 by the author(s).

---
[1]To be contrasted with an "optimization" perspective, e.g. guarantees of convergence to a critical point of the *training* objective.
[2]See Dahl et al. (2023, Section 4.2) for details.

*How do we explain the performance of Kronecker-Factored preconditioned optimizers?*

In a seemingly distant area, the learning theory community has been interested in studying the solutions learned by abstractions of "typical" deep learning set-ups, where the overall goal is to theoretically demonstrate how neural networks *learn* features from data to perform better than classical "fixed features" methods, e.g. kernel machines. Much of this line of work focuses on analyzing the performance of SGD on simplified models of deep learning (see e.g. Collins et al. (2021); Damian et al. (2022); Ba et al. (2022); Barak et al. (2022); Abbe et al. (2023); Dandi et al. (2024a); Berthier et al. (2024); Nichani et al. (2024b); Collins et al. (2024)). Almost invariably, certain innocuous-looking assumptions are made, such as isotropic covariates $\mathbf{x} \sim \mathsf{N}(\mathbf{0}, \mathbf{I})$. Under these conditions, SGD has been shown to exhibit desirable generalization properties. However, some works deviate from these assumptions in specific settings (Amari et al., 2021; Zhang et al., 2024b), and suggest that SGD can exhibit severely suboptimal generalization. Thus, toward extending our understanding of feature learning, it seems beneficial to consider a broader family of optimizers. This raises the following question:

*What is a* practical *family of optimization algorithms that overcomes the deficiencies of* SGD *for standard feature learning tasks?*

We answer the above two questions by focusing on two prototypical problems used to theoretically study feature learning: *linear representation learning* and *single-index learning*. In both problems, we show that SGD is clearly suboptimal outside ideal settings, such as when the ubiquitous isotropic data $\mathsf{N}(\mathbf{0}, \mathbf{I})$ assumption is violated. By inspecting the root cause behind these suboptimalities, we show that Kronecker-Factored preconditioners arise naturally as a *first-principles* solution to these issues. We provide novel non-approximation-theoretic motivations for this class of algorithms, while establishing new and improved learning-theoretic guarantees. We hope that this serves as strong evidence of an untapped synergy between deep learning optimization and feature learning theory.

**Contributions.**

- We study the linear representation learning problem under general anisotropic $\mathbf{x} \sim \mathsf{N}(\mathbf{0}, \mathbf{\Sigma_x})$ covariates and show that the convergence of SGD can be drastically slow, even under mild anisotropy. Also, the convergence rate suffers an undesirable dependence on the "conditioning" of the instance even for ideal step-sizes. We arrive at a variant of KFAC as the natural solution to these deficiencies of SGD, giving rise to the first *condition-number-free convergence rate* for the problem (Section 3.1).

- Next, we consider the problem of learning a single-index model using a two-layer neural network in the high-dimensional proportional limit. We show that for anisotropic covariates $\mathbf{x} \sim \mathsf{N}(\mathbf{0}, \mathbf{\Sigma_x})$, SGD fails to learn useful features, whereas it is known that it learns suitable features in the isotropic setting. Furthermore, we show that KFAC is a natural fix to SGD, greatly enhancing the learned features in anisotropic settings (Section 3.2).

- Lastly, we carefully numerically verify our theoretical predictions. Notably, we confirm the findings in Benzing (2022) that full second-order methods heavily underperform KFAC in convergence rate and stability. We also show standard tools like Adam-like preconditioning and batch-norm (Ioffe & Szegedy, 2015) do not fix the issues we identify, even for our simple models, and may even *hurt* generalization in the latter's case.

In addition to the works discussed earlier, we provide extensive related work and background in Appendix A.

**Notation.** We denote vector quantities by **bold** lower-case, and matrix quantities by **bold** upper-case. We use $\odot$ to denote element-wise (Hadamard) product, $\otimes$ for Kronecker product, and $\mathrm{vec}(\cdot)$ the *column-major* vectorization operator. Positive (semi-)definite matrices are denoted by $\mathbf{Q} \succ (\succeq) \mathbf{0}$, and the corresponding partial order $\mathbf{P} \preceq \mathbf{Q} \implies \mathbf{Q} - \mathbf{P} \succeq \mathbf{0}$. We use $\|\cdot\|_{\mathrm{op}}, \|\cdot\|_F$ to denote the operator (spectral) and Frobenius norms. We use $\mathsf{E}[f(\mathbf{x})]$ to denote the expectation of $f(\mathbf{x})$, and $\mathbb{P}[A(\mathbf{x})]$ to denote the probability of event $A(\mathbf{x})$. Given a batch $\{\mathbf{x}_i\}_{i=1}^n$, we denote the *empirical expectation* $\widehat{\mathsf{E}}[f(\mathbf{x})] = \frac{1}{n} \sum_{i=1}^n f(\mathbf{x}_i)$. Given an indexed set of vectors, we use the upper case to denote the (row-wise) stacked matrix, e.g. $\mathbf{X} \triangleq \begin{bmatrix} \mathbf{x}_1 & \cdots & \mathbf{x}_n \end{bmatrix}^\top \in \mathbb{R}^{n \times d_{\mathbf{x}}}$. We reserve $\mathbf{\Sigma}$ ($\widehat{\mathbf{\Sigma}}$) for (sample) covariance matrices, e.g. $\mathbf{\Sigma_x} = \mathsf{E}[\mathbf{x}\mathbf{x}^\top]$, $\widehat{\mathbf{\Sigma}}_{\mathbf{x}} = \widehat{\mathsf{E}}[\mathbf{x}\mathbf{x}^\top] = \frac{1}{n}\mathbf{X}^\top\mathbf{X}$. We use $\lesssim, \gtrsim, \approx$ to omit universal numerical constants, and standard asymptotic notation $o(\cdot), \mathcal{O}(\cdot), \Omega(\cdot), \Theta(\cdot)$. Lastly, we use the index shorthand $[n] = \{1, \ldots, n\}$, and subscript $_+$ to denote the "next iterate", e.g. $\mathbf{G}_+ = \mathrm{Next}(\mathbf{G})$.

## 2. Kronecker-Factored Approximation

One of the longest-standing research efforts in optimization literature is dedicated to understanding the role of (local) curvature toward accelerating convergence rates of optimization methods. An example is Newton's method, where the curvature matrix (Hessian) serves as a preconditioner of the gradient, enabling one-shot convergence in quadratic optimization, in which gradient descent enjoys at best a linear convergence rate dictated by the conditioning of the problem. However, for high-dimensional variables, computing and storing the full curvature matrix is often infeasible. Thus enter Quasi-Newton and (preconditioned) Conjugate Gradient methods, where the goal is to reap the benefits of curvature under computational or structural specifications, such as {block-diagonal, low-rank, sparsity, etc.} constraints (e.g. BFGS family (Goldfarb, 1970; Liu & Nocedal, 1989; Nocedal & Wright, 1999)), or accessing the curvature matrix

only through matrix-vector products (see e.g. Pearlmutter (1994); Schraudolph (2002); Martens (2010)).

Nevertheless, the use of these methods for neural network optimization introduces new considerations. Consider an $L$-layer fully-connected neural network (omitting biases) $f_{\boldsymbol{\theta}}(\mathbf{x}) \triangleq \mathbf{W}_L \sigma(\mathbf{W}_{L-1} \cdots \sigma(\mathbf{W}_1 \mathbf{x}) \cdots)$, where $\mathbf{W}_\ell \in \mathbb{R}^{d \times d}$, $\ell \in [L]$ and $\boldsymbol{\theta} \in \mathbb{R}^{Ld^2}$ is the concatenation of $\boldsymbol{\theta}_\ell \triangleq \text{vec}(\mathbf{W}_\ell)$, $\ell \in [L]$. Firstly, establishing convergence of SGD (Arora et al., 2019), NGD (Zhang et al., 2019), or Gauss-Newton (Cai et al., 2019) (or their corresponding gradient flows) to global minima of the *training* objective is non-trivial, as optimization over $\boldsymbol{\theta}$ is non-convex. Moreover, these results do not directly characterize the structure of the resulting features learned by the algorithms. Secondly, on the practical front, full preconditioners on $\boldsymbol{\theta}$ require memory $\mathcal{O}(L^2 d^4)$, which grows prohibitively with depth and width. Block-diagonal approximations (where one curvature block $\mathbf{M}_\ell \in \mathbb{R}^{d^2 \times d^2}$ corresponds to a layer $\boldsymbol{\theta}_\ell$) still require $\mathcal{O}(Ld^4)$. Thus, entry-wise preconditioning as in Adam, with footprint $\mathcal{O}(Ld^2) \approx \dim(\boldsymbol{\theta})$, is usually considered the only scalable class of preconditioners.

However, a distinct notion of "Kronecker-Factored" preconditioning emerged approximately concurrently with Adam, with representative examples such as KFAC and Shampoo. As its name suggests, since full block-diagonal approximations are too expensive, a Kronecker-Factored approximation is made instead, where $\mathbf{M}_\ell^{-1} \nabla_{\boldsymbol{\theta}_\ell} \mathcal{L}(\boldsymbol{\theta}) = (\mathbf{Q}_\ell \otimes \mathbf{P}_\ell)^{-1} \nabla_{\boldsymbol{\theta}_\ell} \mathcal{L}(\boldsymbol{\theta})$, $\mathbf{P}_\ell, \mathbf{Q}_\ell \succeq \mathbf{0}$. Using properties of the Kronecker product (see Lemma E.1), this has the convenient interpretation of pre- and post-multiplying the weights in their *matrix form*:

$$(\mathbf{Q}_\ell \otimes \mathbf{P}_\ell)^{-1} \nabla_{\boldsymbol{\theta}_\ell} \mathcal{L}(\boldsymbol{\theta}) \iff \mathbf{P}_\ell^{-1} \nabla_{\mathbf{W}_\ell} \mathcal{L}(\boldsymbol{\theta}) \mathbf{Q}_\ell^{-1}. \quad (1)$$

As such, the memory requirement of Kronecker-Factored layer-wise preconditioning is $\mathcal{O}(Ld^2)$, matching that of entry-wise preconditioning. The notion of curvature differs from case to case, e.g., for KFAC, this is the Fisher Information matrix corresponding to the distribution parameterized by $f_{\boldsymbol{\theta}}(\mathbf{x})$, whereas for Shampoo this is the full-matrix Adagrad preconditioner, in turn closely related to the Gauss-Newton matrix.[3] We provide some sample derivations and background in Appendix D. However, as aforementioned, an approximation viewpoint falls short of explaining the practical performance of Kronecker-Factored methods, as they typically converge *faster* than their corresponding second-order method (Benzing, 2022) on deep learning tasks. This motivates understanding the unique benefits of layer-wise preconditioning methods from first principles, which brings us to the following section.

---

[3]This is itself a positive-definite approximation of the Hessian.

## 3. Feature Learning via Kronecker-Factored Preconditioning

We present two prototypical models of feature learning, *linear representation learning* and *single-index learning*, and demonstrate how typical guarantees for the features learned by SGD break down outside of idealized settings. We then show how to rectify these issues by deriving a modified algorithm from first principles, and demonstrate that both cases in fact coincide with a particular Kronecker-factored preconditioning method. We now set-up the model architecture and algorithm primitive considered in both problems. We consider two-layer feedforward neural network predictors:

$$f_{\mathbf{F},\mathbf{G}}(\mathbf{x}) = \mathbf{F}\sigma(\mathbf{G}\mathbf{x}), \quad (2)$$

where $\mathbf{F} \in \mathbb{R}^{d_Y \times d_h}$, $\mathbf{G} \in \mathbb{R}^{d_h \times d_X}$ denote the weight matrices and $\sigma(\cdot)$ is a predetermined activation function. For scalar outputs $d_Y = 1$, we use $f_{\mathbf{f},\mathbf{G}}(\mathbf{x}) = \mathbf{f}^\top \sigma(\mathbf{G}\mathbf{x})$. For our purposes, we omit the bias vectors from both layers. We further denote the intermediate covariate pre- and post-activation $\mathbf{h} \triangleq \mathbf{G}\mathbf{x}$, $\mathbf{z} \triangleq \sigma(\mathbf{G}\mathbf{x})$. We consider a standard mean-squared-error (MSE) regression objective and its (batch) empirical counterpart:

$$\mathcal{L}(\mathbf{F}, \mathbf{G}) \triangleq \frac{1}{2} \mathsf{E}_{(\mathbf{x},\mathbf{y})} \big[ \|\mathbf{y} - f_{\mathbf{F},\mathbf{G}}(\mathbf{x})\|^2 \big]$$
$$\widehat{\mathcal{L}}(\mathbf{F}, \mathbf{G}) \triangleq \frac{1}{2} \widehat{\mathsf{E}} \big[ \|\mathbf{y} - f_{\mathbf{F},\mathbf{G}}(\mathbf{x})\|^2 \big] \quad (3)$$

Given a batch of inputs $\{\mathbf{x}_i\}_{i=1}^n$, we define the left and right preconditioners of the two layers (recall equation (1)):

$$\mathbf{Q}_{\mathbf{G}} = \widehat{\boldsymbol{\Sigma}}_{\mathbf{x}} = \widehat{\mathsf{E}}[\mathbf{x}\mathbf{x}^\top], \quad \mathbf{Q}_{\mathbf{F}} = \widehat{\boldsymbol{\Sigma}}_{\mathbf{z}} = \widehat{\mathsf{E}}[\mathbf{z}\mathbf{z}^\top], \quad (4)$$

$$\mathbf{P}_{\mathbf{G}} = \widehat{\mathsf{E}} \left[ \left( \frac{\partial f_{\mathbf{F},\mathbf{G}}}{\partial \mathbf{h}} \right)^\top \frac{\partial f_{\mathbf{F},\mathbf{G}}}{\partial \mathbf{h}} \right] (\text{or } \mathbf{I}_{d_h}), \quad \mathbf{P}_{\mathbf{F}} = \mathbf{I}_{d_Y}.$$

We introduce the flexibility of $\mathbf{P}_{\mathbf{G}} = \mathbf{I}_{d_h}$ for when $\mathbf{P}_{\mathbf{G}}$ does not play a significant role; notably, this recovers certain Kronecker-Factored preconditioners that avoid extra backwards passes (see Appendix D). We consider a stylized alternating descent primitive, where we iteratively perform

$$\mathbf{G}_+ = \mathbf{G} - \eta_{\mathbf{G}} \mathbf{P}_{\mathbf{G}}^{-1} \nabla_{\mathbf{G}} \widehat{\mathcal{L}}(\mathbf{F}, \mathbf{G}) (\mathbf{Q}_{\mathbf{G}} + \lambda_{\mathbf{G}} \mathbf{I}_{d_X})^{-1}$$
$$\mathbf{F}_+ = \mathbf{F} - \eta_{\mathbf{F}} \mathbf{P}_{\mathbf{F}}^{-1} \nabla_{\mathbf{F}} \widehat{\mathcal{L}}(\mathbf{F}, \mathbf{G}_+) \mathbf{Q}_{\mathbf{F}}^{-1}, \quad (5)$$

where $\eta_{\mathbf{G}}, \eta_{\mathbf{F}} > 0$ are layer-wise learning rates, and $\lambda_{\mathbf{G}} \in \mathbb{R}$ is a regularization parameter. In line with most prior work, we consider an alternating scheme for analytical convenience. We also assume that $\mathbf{G}_+, \mathbf{F}_+$ are computed on *independent batches* of data, equivalent to sample-splitting strategies in prior analysis (Collins et al. (2021); Zhang et al. (2024b); Ba et al. (2022); Moniri et al. (2024) etc).

The preconditioners (4) and update (5) bear a striking resemblance to KFAC (cf. Appendix D). In fact, the preconditioners align exactly with KFAC if we view $\mathcal{L}(\mathbf{F}, \mathbf{G})$ as a

negative log-likelihood of a conditionally Gaussian model with fixed variance: $\widehat{\mathbf{y}}(\mathbf{x}) \sim \mathsf{N}(f_{\mathbf{F},\mathbf{G}}(\mathbf{x}), \sigma^2\mathbf{I})$. This is in some sense a coincidence (and a testament to the prescience of KFAC's design): rather than deriving the above preconditioners via approximating the Fisher Information matrix, we will show shortly how they arise as a natural adjustment to SGD in our featured problems. We note that Kronecker-Factored preconditioning methods often involve further moving parts such as damping exponents $\mathbf{P}^{-\rho}$, additional ridge parameters on various preconditioners, and momentum. Though of great importance in practice, they are beyond the scope of this paper,[4] and we only feature parameters that play a role in our analysis.

A convenient observation that unifies various stylized algorithms on two-layer networks is that the $\mathbf{F}$-update in (5) can be interpreted as an exponential moving average (EMA) over least-squares estimators conditioned on $\mathbf{z}$.

**Lemma 3.1.** *Given* $\mathbf{G}$*, define the least-squares estimator:*

$$\widehat{\mathbf{F}}_{\mathrm{ls}} \triangleq \operatorname*{argmin}_{\widehat{\mathbf{F}}} \frac{1}{2}\widehat{\mathsf{E}}\big[\|\mathbf{y} - \widehat{\mathbf{F}}\,\underbrace{\sigma(\mathbf{G}\mathbf{x})}_{\mathbf{z}}\|^2\big] = \mathbf{Y}^\top\mathbf{Z}\,(\mathbf{Z}^\top\mathbf{Z})^{-1}$$

*Given* $\eta_{\mathbf{F}} \in (0, 1]$*, then the* $\mathbf{F}$*-update in* (5) *can be re-written as an EMA of* $\widehat{\mathbf{F}}_{\mathrm{ls}}$*; i.e.,* $\mathbf{F}_+ = (1 - \eta_{\mathbf{F}})\mathbf{F} + \eta_{\mathbf{F}}\widehat{\mathbf{F}}_{\mathrm{ls}}$.

In particular, many prior works (e.g. Collins et al. (2021); Nayer & Vaswani (2022); Thekumparampil et al. (2021); Zhang et al. (2024b)) consider an alternating "minimization-descent" approach, where out of analytical convenience $\mathbf{F}$ is updated by performing least-squares regression holding the hidden layer $\mathbf{G}$ fixed. In light of Lemma 3.1, this corresponds to the case where $\eta_{\mathbf{F}} = 1$.

### 3.1. Linear Representation Learning

Assume we have data generated by the following process

$$\mathbf{x}_i \overset{\text{i.i.d.}}{\sim} \mathsf{N}(0, \boldsymbol{\Sigma}_{\mathbf{x}}), \quad \mathbf{y}_i = \mathbf{F}_\star\mathbf{G}_\star\mathbf{x}_i + \boldsymbol{\varepsilon}_i, \qquad (6)$$

where $\boldsymbol{\Sigma}_{\mathbf{x}}$ is the input covariance, $\mathbf{F}_\star \in \mathbb{R}^{d_{\mathsf{Y}} \times k}$, $\mathbf{G}_\star \in \mathbb{R}^{k \times d_{\mathsf{X}}}$ are (unknown) rank-$k$ matrices, and $\boldsymbol{\varepsilon}_i \overset{\text{i.i.d.}}{\sim} \mathsf{N}(0, \boldsymbol{\Sigma}_{\boldsymbol{\varepsilon}})$ is additive label noise independent of all other randomness. We consider Gaussian data throughout the paper for conciseness; all results in this section can be extended to subgaussian $\mathbf{x}, \varepsilon$ via standard tools, affecting only the universal constants (see Appendix E). Let us define $\sigma_{\boldsymbol{\varepsilon}}^2 \triangleq \lambda_{\max}(\boldsymbol{\Sigma}_{\boldsymbol{\varepsilon}})$. Accordingly, our predictor model is a two-layer linear net (2) with $\mathbf{F} \in \mathbb{R}^{d_{\mathsf{Y}} \times k}$, $\mathbf{G} \in \mathbb{R}^{k \times d_{\mathsf{X}}}$.

The goal of linear representation learning is to learn the low-dimensional feature space that $\mathbf{G}_\star$ maps to, which is

equivalent to determining its row-space $\operatorname{rowsp}(\mathbf{G}_\star)$. Recovering $\mathbf{F}_\star, \mathbf{G}_\star$ is an ill-posed problem, as for any invertible $\mathbf{L} \in \mathbb{R}^{k \times k}$, the matrices $\mathbf{F}_\star\mathbf{L}$, $\mathbf{L}^{-1}\mathbf{G}_\star$ remain optimal. Therefore, we measure recovery of $\operatorname{rowsp}(\mathbf{G}_\star)$ via a *subspace distance*.

**Definition 3.2** (Subspace Distance (Stewart & Sun, 1990)). Let $\mathbf{G}, \mathbf{G}_\star \in \mathbb{R}^{k \times d_{\mathsf{X}}}$ be matrices whose rows are orthonormal. Let $\mathcal{P}_\star^\perp \in \mathbb{R}^{d_{\mathsf{X}} \times d_{\mathsf{X}}}$ be the projection matrix onto $\operatorname{rowsp}(\mathbf{G}_\star)^\perp$. Define the distance between the subspaces spanned by the rows of $\mathbf{G}$ and $\mathbf{G}_\star$ by

$$\operatorname{dist}(\mathbf{G}, \mathbf{G}_\star) \triangleq \|\mathbf{G}\mathcal{P}_\star^\perp\|_{\mathrm{op}} \qquad (7)$$

The subspace distance quantitatively captures the alignment between two subspaces, ranging between $0$ (occurring iff $\operatorname{rowsp}(\mathbf{G}_\star) = \operatorname{rowsp}(\mathbf{G})$) and $1$ (occurring iff $\operatorname{rowsp}(\mathbf{G}_\star) \perp \operatorname{rowsp}(\mathbf{G})$). We further make the following non-degeneracy assumptions.

**Assumption 3.3.** We assume $\mathbf{G}_\star$ is row-orthonormal, and $\mathbf{F}_\star$ is full-rank, $\operatorname{rank}(\mathbf{F}_\star) = k \leq d_{\mathsf{Y}}$. This is without loss of generality: if $k > d_{\mathsf{Y}}$, then recovering a $k$-dimensional row-space from $\mathbf{y}_i$ is underdetermined. If $\operatorname{rank}(\mathbf{F}_\star) = k' < k$, then it suffices to consider $\mathbf{G}_\star \in \mathbb{R}^{k' \times d_{\mathsf{X}}}$.

The linear representation learning problem has often been studied in the context of multi-task learning (Du et al., 2021; Tripuraneni et al., 2020; Collins et al., 2021; Thekumparampil et al., 2021; Zhang et al., 2024b).

*Remark* 3.4 (Multi-task Learning). Multi-task learning considers data generated as $\mathbf{y}_i^{(t)} = \mathbf{F}_\star^{(t)}\mathbf{G}_\star\mathbf{x}_i^{(t)} + \boldsymbol{\varepsilon}_i^{(t)}$ for distinct tasks $t = 1, \ldots, T$, with the same goal of recovering the *shared* representation $\mathbf{G}_\star$. Our algorithm and guarantees naturally extend here, see Appendix B.3 for full details. In particular, by embedding $\mathbf{F}_\star = \begin{bmatrix} \mathbf{F}_\star^{(1)\top} & \cdots & \mathbf{F}_\star^{(T)\top} \end{bmatrix}^\top$, Assumption 3.3 is equivalent to the "task-diversity" conditions in the above works: $\operatorname{rank}(\mathbf{F}_\star) = \operatorname{rank}\left(\sum_{t=1}^T \mathbf{F}_\star^{(t)\top}\mathbf{F}_\star^{(t)}\right) = k$.

We maintain the "single-task" setting in this section for concise bookkeeping while preserving the essential features of the representation learning problem. Various algorithms have been proposed toward provably recovering the representation $\mathbf{G}_\star$. A prominent example is an alternating minimization-SGD scheme (Collins et al., 2021; Vaswani, 2024). In the cited works, a local convergence result[5] is established for isotropic data $\boldsymbol{\Sigma}_{\mathbf{x}} = \mathbf{I}_{d_{\mathsf{X}}}$. In Zhang et al. (2024b), it is shown that using SGD can drastically slow convergence even under mild anisotropy; their proposed algorithmic adjustment equates to applying the right-preconditioner $\mathbf{Q}_{\mathbf{G}} = \widehat{\boldsymbol{\Sigma}}_{\mathbf{x}}$. However, their local convergence result suffers a dependence on the condition

---

[4]We refer the interested reader to Ishikawa & Karakida (2023) for discussion of these settings in the "maximal-update parameterization" framework (Yang & Hu, 2021b).

[5]By local convergence here we mean $\operatorname{dist}(\mathbf{G}, \mathbf{G}_\star)$ is sufficiently (but non-vanishingly) small.

number of $\mathbf{F}_\star$, slowing the linear convergence rate for ill-conditioned $\mathbf{F}_\star$. Let us now specify the algorithm template used in this section, that also encompasses the above work:

$$\overline{\mathbf{G}}_+ \text{ via (5)}, \ \mathbf{G}_+ = \text{Ortho}(\overline{\mathbf{G}}_+), \ \mathbf{F}_+ \text{ via (5)}. \quad (8)$$

Notably, we row-orthonormalize the representation after each update. Besides ease of analysis, we have observed this numerically mitigates the elements of $\mathbf{G}$ from blow-up when running variants of SGD. The alternating min-SGD algorithms in Collins et al. (2021); Vaswani (2024) are equivalent to iterating (8) setting $\mathbf{P}_\mathbf{G} = \mathbf{Q}_\mathbf{G} = \mathbf{I}$, $\eta_\mathbf{F} = 1$ in (4), whereas Zhang et al. (2024b) use $\mathbf{P}_\mathbf{G} = \mathbf{I}$, $\mathbf{Q}_\mathbf{G} = \widehat{\Sigma}_\mathbf{x}$, $\eta_\mathbf{F} = 1$. Let us now write out the full-batch gradient update.

**Full-Batch SGD.** Given a fresh batch of data $\{(\mathbf{x}_i, \mathbf{y}_i)\}_{i=1}^n$, and current weights $(\mathbf{F}, \mathbf{G})$, we have the representation gradient and corresponding SGD step:

$$\nabla_\mathbf{G}\widehat{\mathcal{L}}(\mathbf{F}, \mathbf{G}) = \frac{1}{n}\mathbf{F}^\top\left(\mathbf{FGX}^\top\mathbf{X} - \mathbf{Y}^\top\mathbf{X}\right)$$
$$\mathbf{G}_+ = \mathbf{G} - \eta_\mathbf{G}\nabla_\mathbf{G}\widehat{\mathcal{L}}(\mathbf{F}, \mathbf{G}). \quad (9)$$

When $\mathbf{x}$ is isotropic $\Sigma_\mathbf{x} = \mathbf{I}_{d_x}$, the key observation is that by multiplying both sides of (9) by $\mathcal{P}_\star^\perp$, recalling $\mathbf{Y}^\top = \mathbf{F}_\star\mathbf{G}_\star\mathbf{X}^\top + \mathcal{E}^\top$, we have

$$\overline{\mathbf{G}}_+\mathcal{P}_\star^\perp = \left(\mathbf{G} - \eta_\mathbf{G}\mathbf{F}^\top\left((\mathbf{FG} - \mathbf{F}_\star\mathbf{G}_\star)\widehat{\Sigma}_\mathbf{x} - \frac{1}{n}\mathcal{E}^\top\mathbf{X}\right)\right)\mathcal{P}_\star^\perp$$
$$\approx (\mathbf{I}_k - \eta_\mathbf{G}\mathbf{F}^\top\mathbf{F})\mathbf{G}\mathcal{P}_\star^\perp + \frac{\eta_\mathbf{G}}{n}\mathbf{F}^\top\mathcal{E}^\top\mathbf{X}\mathcal{P}_\star^\perp,$$

where the approximate equality hinges on covariance concentration $\widehat{\Sigma}_\mathbf{x} \approx \mathbf{I}_{d_x}$ and $\mathbf{G}_\star\mathcal{P}_\star^\perp = \mathbf{0}$. Therefore, in the isotropic setting, for sufficiently large $n \gtrsim d_x$, and appropriately chosen $\eta_\mathbf{G} \approx \frac{1}{\lambda_{\max}(\mathbf{F}_\star^\top\mathbf{F}_\star)}$, then (omitting many details) we have the one-step contraction (Collins et al., 2021; Vaswani, 2024): defining $\Delta = \mathcal{O}\left(\sigma_\varepsilon\sqrt{d_x/n}\right)$,

$$\text{dist}(\mathbf{G}_+, \mathbf{G}_\star) \lesssim \left(1 - \frac{\lambda_{\min}(\mathbf{F}_\star^\top\mathbf{F}_\star)}{\lambda_{\max}(\mathbf{F}_\star^\top\mathbf{F}_\star)}\right)\text{dist}(\mathbf{G}, \mathbf{G}_\star) + \Delta \quad (10)$$

Therefore, in low-noise/large-batch settings, this demonstrates SGD on the representation $\mathbf{G}$ converges geometrically to $\mathbf{G}_\star$ (in subspace distance). However, there are clear suboptimalities to SGD. Firstly, the above analysis critically relies on $\Sigma_\mathbf{x} = \mathbf{I}_{d_x}$ such that $\mathbf{G}_\star\widehat{\Sigma}_\mathbf{x}\mathcal{P}_\star^\perp \approx \mathbf{0}$. As aforementioned, this is demonstrated to be crucial in Zhang et al. (2024b) for $\text{dist}(\mathbf{G}, \mathbf{G}_\star)$ to converge using SGD. Secondly, the convergence of SGD is bottlenecked by the conditioning of $\mathbf{F}_\star$. In fact, we show the dependence on $\mathbf{F}_\star$ in the contraction rate bound (10) cannot be improved in general, even under the most benign assumptions. Following Collins et al. (2021); Vaswani (2024), we define $\mathbf{G}_T = \text{SGD}(\mathbf{G}_0; \eta_\mathbf{G}, T)$ as the output of alternating min-SGD, i.e. iterating (8) setting $\mathbf{P}_\mathbf{G} = \mathbf{Q}_\mathbf{G} = \mathbf{I}$, $\eta_\mathbf{F} = 1$ in (4), for $T$ steps with fixed step-size $\eta_\mathbf{G}$ starting from $\mathbf{G}_0$.

**Proposition 3.5.** *Let* $\Sigma_\mathbf{x} = \mathbf{I}_{d_x}$, $n = \infty$. *Choose any* $d_x > k$, $d_y \geq k \geq 2$. *Let the learner be given knowledge of* $\mathbf{F}_\star, \mathbf{G}_\star$ *and* $\text{dist}(\mathbf{G}_0, \mathbf{G}_\star)$. *However, assume the learner must fix* $\eta_\mathbf{G} > 0$ *before observing* $\mathbf{G}_0$. *Then, there exists* $\mathbf{F}_\star \in \mathbb{R}^{d_y \times k}$, $\mathbf{G}_\star, \mathbf{G}_0 \in \mathbb{R}^{k \times d_x}$, *such that* $\mathbf{G}_T = \text{SGD}(\mathbf{G}_0; \eta_\mathbf{G}, T)$ *satisfies:*

$$\text{dist}(\mathbf{G}_T, \mathbf{G}_\star) \geq \left(1 - 4\frac{\lambda_{\min}(\mathbf{F}_\star^\top\mathbf{F}_\star)}{\lambda_{\max}(\mathbf{F}_\star^\top\mathbf{F}_\star)}\right)^T \text{dist}(\mathbf{G}_0, \mathbf{G}_\star).$$

The proof can be found in Appendix B.1. Since we set $\Sigma_\mathbf{x} = \mathbf{I}$, the lower bound also holds for the algorithm in Zhang et al. (2024b). We remark departing from a worst-case analysis to a generic performance lower bound, e.g. random initialization or varying step-sizes, is a nuanced topic even for the simple case of convex quadratics; see e.g. Bach (2024); Altschuler & Parrilo (2024). In light of Proposition 3.5 and (9), a sensible alteration might be to *pre- and post- multiply* $\nabla_\mathbf{G}\widehat{\mathcal{L}}(\mathbf{F}, \mathbf{G})$ by $(\mathbf{F}^\top\mathbf{F})^{-1}$ and $\widehat{\Sigma}_\mathbf{x}^{-1}$. These observations bring us to the proposed recipe in (5).

**Stylized KFAC.** By analyzing the shortcomings of the SGD update, we arrive at the proposed representation update:

$$\overline{\mathbf{G}}_+ = \mathbf{G} - \eta_\mathbf{G}(\mathbf{F}^\top\mathbf{F})^{-1}\nabla_\mathbf{G}\widehat{\mathcal{L}}(\mathbf{F}, \mathbf{G})\widehat{\Sigma}_\mathbf{x}^{-1}.$$

We can verify from (4) and (5) that $\mathbf{P}_\mathbf{G} = \mathbf{F}^\top\mathbf{F}$ and $\mathbf{Q}_\mathbf{G} = \widehat{\Sigma}_\mathbf{x}$. Thus, we have recovered a stylized variant of KFAC as previewed. Our main result in this section is a local convergence guarantee.

**Theorem 3.6.** *Consider running* (8) *with* $\lambda_\mathbf{G} = 0$, $\eta_\mathbf{G} \in [0, 1]$, *and* $\eta_\mathbf{F} = 1$. *Define* $\overline{\sigma}^2 \triangleq \frac{\sigma_\varepsilon^2}{\sigma_{\min}(\mathbf{F}_\star)^2\lambda_{\min}(\Sigma_\mathbf{x})}$. *As long as* $\text{dist}(\mathbf{G}, \mathbf{G}_\star) \leq 0.01\kappa^{-1}(\Sigma_\mathbf{x})\kappa^{-1}(\mathbf{F}_\star)$ *and* $n \gtrsim \overline{\max}\{1, \sigma^2\}(d_x + \log(1/\delta))$, *we have:*

$$\text{dist}(\mathbf{G}_+, \mathbf{G}_\star) \leq (1 - 0.9\eta_\mathbf{G})\text{dist}(\mathbf{G}, \mathbf{G}_\star) + \eta_\mathbf{G}\Delta,$$

*with probability* $\geq 1 - \delta$, *where* $\Delta \triangleq \mathcal{O}\left(\overline{\sigma}\sqrt{\frac{d_x + \log(1/\delta)}{n}}\right)$.

Crucially, our contraction factor is condition-number-free, subverting the lower bound in Proposition 3.5 for sufficiently ill-conditioned $\mathbf{F}_\star$. Therefore, setting $\eta_\mathbf{G}$ near 1 ensures a universal constant contraction rate. Curiously, our proposed stylized KFAC (8) aligns with an alternating "min-min" scheme (Jain et al., 2013; Thekumparampil et al., 2021), where $\mathbf{F}, \mathbf{G}$ are alternately updated via solving the convex quadratic least-squares problem, by setting $\eta_\mathbf{F} = \eta_\mathbf{G} = 1$. However, our experiments (see Figure 5) demonstrate $\eta_\mathbf{G} = 1$ is generally suboptimal, highlighting the flexibility of viewing KFAC as a descent method.

### 3.1.1. TRANSFER LEARNING

The upshot of representation learning is the ability to *transfer* (e.g. fine-tune) to a distinct, but related, task by only

retraining $\mathbf{F}$ (Du et al., 2021; Kumar et al., 2022). Assume we now have target data generated by:

$$\mathbf{x}_i^{(\mathrm{t})} \overset{\mathrm{i.i.d.}}{\sim} \mathsf{N}(\mathbf{0}, \boldsymbol{\Sigma}_\mathbf{x}^{(\mathrm{t})}), \ \mathbf{y}_i^{(\mathrm{t})} = \mathbf{F}_\star^{(\mathrm{t})} \mathbf{G}_\star \mathbf{x}_i^{(\mathrm{t})} + \boldsymbol{\varepsilon}_i, \quad (11)$$

where $\boldsymbol{\varepsilon}_i \overset{\mathrm{i.i.d.}}{\sim} \mathsf{N}(\mathbf{0}, \boldsymbol{\Sigma}_{\boldsymbol{\varepsilon}})$, $\mathbf{F}_\star^{(\mathrm{t})} \in \mathbb{R}^{d_Y \times k}$. Notably, $\mathbf{G}_\star$ is shared with the "training" distribution (6). Given $\widehat{\mathbf{G}}$ (e.g. by running (8) on training task), we consider fitting the last layer $\mathbf{F}$ given a batch of $n^{(\mathrm{t})}$ data from the target task (11).

**Lemma 3.7.** *Let* $\widehat{\mathbf{F}}_{\mathrm{ls}}^{(\mathrm{t})} = \mathrm{argmin}_{\widehat{\mathbf{F}}} \ \widehat{\mathsf{E}}^{(\mathrm{t})}[\|\mathbf{y}^{(\mathrm{t})} - \widehat{\mathbf{F}}\mathbf{z}^{(\mathrm{t})}\|_2^2]$, $\mathbf{z}^{(\mathrm{t})} \triangleq \widehat{\mathbf{G}}\mathbf{x}^{(\mathrm{t})}$ *be the optimal* $\mathbf{F}$ *on the batch of* $n^{(\mathrm{t})}$ *target data* (11) *given* $\widehat{\mathbf{G}}$. *Defining* $\nu = \mathrm{dist}(\widehat{\mathbf{G}}, \mathbf{G}_\star)$, *given* $n^{(\mathrm{t})} \gtrsim k + \log(1/\delta)$, *we have with probability* $\geq 1 - \delta$:

$$\mathcal{L}^{(\mathrm{t})}(\widehat{\mathbf{F}}_{\mathrm{ls}}^{(\mathrm{t})}, \widehat{\mathbf{G}}) \triangleq \mathsf{E}\left[\|\mathbf{y}^{(\mathrm{t})} - \widehat{\mathbf{F}}_{\mathrm{ls}}^{(\mathrm{t})}\widehat{\mathbf{G}}\mathbf{x}^{(\mathrm{t})}\|_2^2\right]$$

$$\lesssim \|\mathbf{F}_\star^{(\mathrm{t})}\|_F^2 \, \lambda_{\max}(\boldsymbol{\Sigma}_\mathbf{x}^{(\mathrm{t})})\nu^2 + \frac{\sigma_{\boldsymbol{\varepsilon}}^2(d_Y k + \log(1/\delta))}{n^{(\mathrm{t})}}.$$

As hoped, the MSE of the fine-tuned predictor decomposes into a bias term scaling with the quality of $\widehat{\mathbf{G}}$, and a noise term scaling with $\dim(\mathbf{F})/n^{(\mathrm{t})}$. We comment the required data is $\approx k$ rather than $\approx d_\mathbf{x}$ resulting from doing regression from scratch (Wainwright, 2019). Additionally, the noise term scales with $\dim(\mathbf{F}) = d_Y k$ rather than $d_Y d_\mathbf{x}$ of the full predictor space. The transfer learning set-up (11) also reveals why data normalization (e.g. whitening, batch-norm (Ioffe & Szegedy, 2015)) can be counterproductive. To illustrate this, consider perfectly whitening the training covariates $\mathbf{v} = \boldsymbol{\Sigma}_\mathbf{x}^{-1/2}\mathbf{x}$. By this change of variables, the ground-truth predictor changes $\mathbf{y} \approx \mathbf{F}_\star \mathbf{G}_\star \mathbf{x} = \mathbf{F}_\star \mathbf{G}_\star \boldsymbol{\Sigma}_\mathbf{x}^{1/2}\mathbf{v}$. This is unproblematic so far—in fact, since the covariates $\mathbf{v}$ are isotropic, SGD now may converge. However, instead of $\mathrm{rowsp}(\mathbf{G}_\star)$, the representation now converges to $\mathrm{rowsp}(\mathbf{G}_\star\boldsymbol{\Sigma}_\mathbf{x}^{1/2})$. Deploying on the target task, since $\boldsymbol{\Sigma}_\mathbf{x} \neq \boldsymbol{\Sigma}_\mathbf{x}^{(\mathrm{t})}$, we have $\mathrm{rowsp}(\widehat{\mathbf{G}}) \approx \mathrm{rowsp}(\mathbf{G}_\star\boldsymbol{\Sigma}_\mathbf{x}^{1/2}) \neq \mathrm{rowsp}(\mathbf{G}_\star(\boldsymbol{\Sigma}_\mathbf{x}^{(\mathrm{t})})^{1/2})$. In other words, in return for stabilizing optimization, normalizing the data destroys the shared structure of the predictor model! We illustrate this effect in Figure 3.

### 3.2. Single Index Learning

Assume that we observe $n$ i.i.d. samples generated according to the following single-index model:

$$\mathbf{x}_i \overset{\mathrm{i.i.d.}}{\sim} \mathsf{N}(\mathbf{0}, \boldsymbol{\Sigma}_\mathbf{x}), \quad y_i = \sigma_\star(\boldsymbol{\beta}_\star^\top \mathbf{x}_i) + \varepsilon_i \qquad (12)$$

where $\boldsymbol{\Sigma}_\mathbf{x} \in \mathbb{R}^{d_\mathbf{x} \times d_\mathbf{x}}$ is the input covariance, $\sigma_\star : \mathbb{R} \to \mathbb{R}$ is the teacher activation function, $\boldsymbol{\beta}_\star \in \mathbb{R}^{d_\mathbf{x}}$ is the (unknown) target direction, and $\varepsilon_i$ is an additive noise $\varepsilon_i \overset{\mathrm{i.i.d.}}{\sim} \mathsf{N}(0, \sigma_\varepsilon^2)$ independent of all other sources of randomness. We also make the following common assumption on $\boldsymbol{\beta}_\star$ (Dicker (2016); Dobriban & Wager (2018); Tripuraneni et al.

(2021a); Moniri et al. (2024); Moniri & Hassani (2024a), etc.) that ensures the covariates $\mathbf{x}_i$ alone do not carry any information about the target direction.

**Assumption 3.8.** The vector $\boldsymbol{\beta}_\star$ is drawn from $\boldsymbol{\beta}_\star \sim \mathsf{N}(\mathbf{0}, d_\mathbf{x}^{-1}\mathbf{I}_{d_\mathbf{x}})$ independent of other sources of randomness.

In this section, we study the problem of fitting a two-layer feedforward neural network $f_{\mathbf{f},\mathbf{G}}$ for prediction of unseen data points drawn independently from (12) at test time. When $\mathbf{G}$ is kept at a random initialization and $\mathbf{f}$ is trained using ridge regression, the model coincides with a random features model (Rahimi & Recht, 2007; Montanari et al., 2019; Hu & Lu, 2023) and has repeatedly used as a toy model to study and explain various aspects of practical neural networks (see Lin & Dobriban (2021); Adlam & Pennington (2020); Tripuraneni et al. (2020); Hassani & Javanmard (2024); Bombari et al. (2023); Lee et al. (2023a); Bombari & Mondelli (2024a;b), etc.).

When the covariates are isotropic $\boldsymbol{\Sigma}_\mathbf{x} = \mathbf{I}_{d_\mathbf{x}}$, it is shown that a single step of full-batch SGD update on $\mathbf{G}$ can drastically improve the performance of the model over random features as a result of *feature learning* by aligning the top right-singular-vector of the updated representation layer $\mathbf{G}$ with the direction $\boldsymbol{\beta}_\star$ (Damian et al., 2022; Ba et al., 2022; Moniri et al., 2024; Cui et al., 2024; Dandi et al., 2024a;b;c). In this section, we assume that the covariates are anisotropic and show that in this case, the one-step full batch SGD is suboptimal and can learn an ill-correlated direction even when the sample size $n$ is large. We then demonstrate that the KFAC update with the preconditioners from (4) is in fact the natural fix to the full batch SGD.

**Full-Batch SGD.** Following the prior work, at initialization, we set $\mathbf{f} = d_\mathsf{h}^{-1/2}\mathbf{f}_0$ with $\mathbf{f}_0 \sim \mathsf{N}(\mathbf{0}, d_\mathsf{h}^{-1}\mathbf{I}_{d_\mathsf{h}})$, and $\mathbf{G} = \mathbf{G}_0$ with i.i.d. $\mathsf{N}(0, d_\mathbf{x}^{-1})$ entries. We update $\mathbf{G}$ with one step of full batch SGD with step size $\eta_\mathbf{G} = \eta\sqrt{d_\mathsf{h}}$; i.e.,

$$\mathbf{G}_{\mathrm{SGD}} \triangleq \mathbf{G}_0 - \eta\sqrt{d_\mathsf{h}} \, \nabla_\mathbf{G}\widehat{\mathcal{L}}(\mathbf{f}_0, \mathbf{G}_0).$$

In the following theorem, we provide an approximation of the updated first layer $\mathbf{G}_{\mathrm{SGD}}$, which is a generalization of (Ba et al., 2022, Proposition 2.1) for $\boldsymbol{\Sigma}_\mathbf{x} \neq \mathbf{I}_{d_\mathbf{x}}$.

**Theorem 3.9.** *Assume that the activation function* $\sigma$ *is* $\mathcal{O}(1)$-*Lipschitz and that Assumption 3.8 holds. In the limit where* $n, d_\mathbf{x}, d_\mathsf{h}$ *tend to infinity proportionally, the matrix* $\mathbf{G}_{\mathrm{SGD}}$, *with probability* $1 - o(1)$, *satisfies*

$$\left\|\mathbf{G}_0 + \alpha\eta \, \mathbf{f}_0\boldsymbol{\beta}_{\mathrm{SGD}}^\top - \mathbf{G}_{\mathrm{SGD}}\right\|_{\mathrm{op}} \to 0,$$

*in which* $\alpha = \mathsf{E}_z[\sigma'(z)]$ *with* $z \sim \mathsf{N}(0, d_\mathbf{x}^{-1}\mathrm{Tr}(\boldsymbol{\Sigma}_\mathbf{x}))$, *and the vector* $\boldsymbol{\beta}_{\mathrm{SGD}}$ *is given by* $\boldsymbol{\beta}_{\mathrm{SGD}} = n^{-1}\mathbf{X}^\top\boldsymbol{y}$.

This theorem shows that one step of full batch SGD update approximately adds a rank-one component $\alpha\eta \, \mathbf{f}_0\boldsymbol{\beta}_{\mathrm{SGD}}^\top$ to the

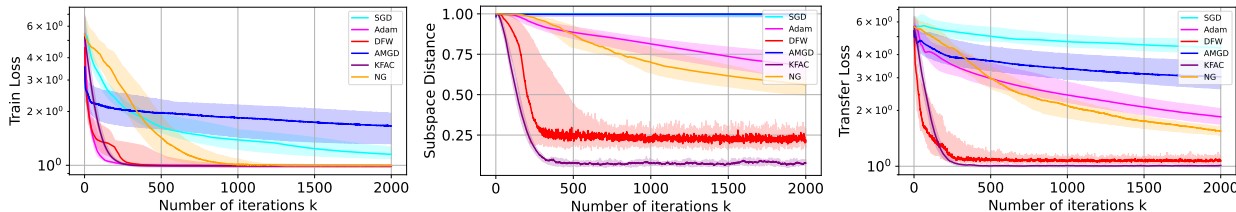

Figure 1. From **left** to **right**: the training loss, subspace distance, and transfer loss induced by various algorithms on a linear representation learning task. We note that various algorithms converge in training loss, but negligibly in subspace distance, and thus transfer loss.

initialized weights $\mathbf{G}_0$. Thus, the pre-activation features for a given input $\mathbf{x} \in \mathbb{R}^{d_x}$ after the update are given by

$$\mathbf{h}_{\text{SGD}} = \mathbf{G}_{\text{SGD}}\mathbf{x} \approx \mathbf{G}_0\mathbf{x} + \alpha\,\eta\,\left(\boldsymbol{\beta}_{\text{SGD}}^\top\mathbf{x}\right)\,\mathbf{f}_0 \in \mathbb{R}^{d_h}$$

where the first and second term correspond to the *random feature*, and the *learned feature* respectively. To better understand the learned feature component, note that defining $c_{\star,1} = \mathsf{E}_{z \sim \mathsf{N}(0, d_x^{-1}\,\mathrm{Tr}(\boldsymbol{\Sigma}_\mathbf{x}))}[\sigma'_\star(z)]$, the target function $\sigma_\star(\boldsymbol{\beta}_\star^\top\mathbf{x})$ can be decomposed as

$$\sigma_\star(\boldsymbol{\beta}_\star^\top\mathbf{x}) = c_{\star,1}\boldsymbol{\beta}_\star^\top\mathbf{x} + \sigma_{\star,\perp}(\boldsymbol{\beta}_\star^\top\mathbf{x})$$

satisfying $\mathsf{E}_\mathbf{x}\left[c_{\star,1}(\boldsymbol{\beta}_\star^\top\mathbf{x})\,\sigma_{\star,\perp}(\boldsymbol{\beta}_\star^\top\mathbf{x})\right] = 0$. Therefore, when $c_{\star,1} \neq 0$, the target function has a *linear part*. Full batch SGD is estimating the direction of $\boldsymbol{\beta}_\star$ using this linear part with the estimator $\boldsymbol{\beta}_{\text{SGD}} = \mathbf{X}^\top\boldsymbol{y}/n$. However, the natural choice for this task is in fact ridge regression $\hat{\boldsymbol{\beta}}_\lambda = (\widehat{\boldsymbol{\Sigma}}_\mathbf{x} + \lambda\,\mathbf{I}_{d_x})^{-1}\mathbf{X}^\top\boldsymbol{y}/n$, and $\boldsymbol{\beta}_{\text{SGD}}$ is missing the prefactor $(\widehat{\boldsymbol{\Sigma}}_\mathbf{x} + \lambda\mathbf{I}_{d_x})^{-1}$. In the isotropic case $\boldsymbol{\Sigma}_\mathbf{x} = \mathbf{I}_{d_x}$, we expect $\widehat{\boldsymbol{\Sigma}}_\mathbf{x} \approx \mathbf{I}_{d_x}$ when $n \gg d_x$. Thus, in this case the estimator $\boldsymbol{\beta}_{\text{SGD}}$ is roughly equivalent to the ridge estimator and can recover the direction $\boldsymbol{\beta}_\star$. However, in the anisotopic case, $\boldsymbol{\beta}_{\text{SGD}}$ is biased even when $n \gg d_x$. To make these intuitions rigorous, we characterize in the following proposition the correlation between the learned direction $\boldsymbol{\beta}_{\text{SGD}}$ and the true direction $\boldsymbol{\beta}_\star$.

**Lemma 3.10.** *Under the assumptions of Theorem 3.9, the correlation between $\boldsymbol{\beta}_\star$ and $\boldsymbol{\beta}_{\text{SGD}}$ satisfies*

$$\left|\frac{\boldsymbol{\beta}_\star^\top\boldsymbol{\beta}_{\text{SGD}}}{\|\boldsymbol{\beta}_{\text{SGD}}\|_2\|\boldsymbol{\beta}_\star\|_2} - \frac{\frac{c_{\star,1}}{d_x}\mathrm{Tr}(\boldsymbol{\Sigma}_\mathbf{x})}{\sqrt{\frac{c_\star^2+\sigma_\varepsilon^2}{n}\mathrm{Tr}(\boldsymbol{\Sigma}_\mathbf{x}) + \frac{c_{\star,1}^2}{d_x}\mathrm{Tr}(\boldsymbol{\Sigma}_\mathbf{x}^2)}}\right| \to 0$$

*with probability $1 - o(1)$, in which $c_{\star,1} = \mathsf{E}_z[\sigma'_\star(z)]$ and $c_\star^2 = \mathsf{E}_z[\sigma_\star^2(z)]$ with $z \sim \mathsf{N}(0, d_x^{-1}\,\mathrm{Tr}(\boldsymbol{\Sigma}_\mathbf{x}))$.*

This lemma shows that the correlation is increasing in the strength of the linear component $c_{\star,1}$ while keeping the signal strength $c_\star$ fixed. Also, based on this lemma, when $n \gg d_x$, the correlation is given by $d_x^{-1}\mathrm{Tr}(\boldsymbol{\Sigma}_\mathbf{x})/\sqrt{d_x^{-1}\mathrm{Tr}(\boldsymbol{\Sigma}_\mathbf{x}^2)}$, which is equal to one *if and only if* $\boldsymbol{\Sigma}_\mathbf{x} = \sigma^2\,\mathbf{I}_{d_x}$ for some $\sigma \in \mathbb{R}$. This means these are the only covariance matrices for which applying one step of full batch SGD update learns the correct direction of $\boldsymbol{\beta}_\star$.

**Stylized KFAC.** This time, we update $\mathbf{G}$ using the stylized KFAC update from (5) with the regularized $\mathbf{P}_\mathbf{G}$. We use the same initialization as full-batch SGD. The updated representation layer in this case is given by

$$\mathbf{G}_{\text{KFAC}} \triangleq \mathbf{G}_0 - \eta\sqrt{d_h}\,\nabla_\mathbf{G}\widehat{\mathcal{L}}(\mathbf{f}_0, \mathbf{G}_0)\,(\mathbf{Q}_\mathbf{G} + \lambda_\mathbf{G}\mathbf{I}_{d_x})^{-1}.$$

The preconditioning factor $(\mathbf{Q}_\mathbf{G} + \lambda_\mathbf{G}\mathbf{I}_{d_x})^{-1}$ with $\mathbf{Q}_\mathbf{G} = \widehat{\boldsymbol{\Sigma}}_\mathbf{x}$ is precisely the factor required so that the direction learned by the one-step update to match the ridge regression estimator with ridge parameter $\lambda_\mathbf{G}$ as shown in the following immediate corollary of Theorem 3.9.

**Corollary 3.11.** *Under the same set of assumptions as Theorem 3.9, the matrix $\mathbf{G}_{\text{KFAC}}$, satisfies*

$$\left\|\mathbf{G}_0 + \alpha\eta\,\mathbf{f}_0\boldsymbol{\beta}_{\text{KFAC}}^\top - \mathbf{G}_{\text{KFAC}}\right\|_{\text{op}} \to 0$$

*with probability $1 - o(1)$, where $\alpha$ is defined in Theorem 3.9, and $\boldsymbol{\beta}_{\text{KFAC}} = (\mathbf{Q}_\mathbf{G} + \lambda_\mathbf{G}\mathbf{I}_{d_x})^{-1}\mathbf{X}^\top\boldsymbol{y}/n = \hat{\boldsymbol{\beta}}_{\lambda_\mathbf{G}}$.*

Because $\boldsymbol{\beta}_{\text{KFAC}}$ is equivalent to ridge regression, we expect it to align well with $\boldsymbol{\beta}_\star$ even for anisotropic $\boldsymbol{\Sigma}_\mathbf{x}$, given a proper choice of $\lambda_\mathbf{G}$. The following lemma formally characterizes the correlation between $\boldsymbol{\beta}_{\text{KFAC}}$ and $\boldsymbol{\beta}_\star$ for any $\lambda_\mathbf{G} \in \mathbb{R}$.

**Lemma 3.12.** *Under the assumptions of Theorem 3.9, the correlation between $\boldsymbol{\beta}_\star$ and $\boldsymbol{\beta}_{\text{KFAC}}$ satisfies*

$$\left|\frac{\boldsymbol{\beta}_{\text{KFAC}}^\top\boldsymbol{\beta}_\star}{\|\boldsymbol{\beta}_{\text{KFAC}}\|_2\|\boldsymbol{\beta}_\star\|_2} - \frac{c_{\star,1}\Psi_1}{\sqrt{c_{\star,1}^2\Psi_2 + \frac{d_x}{n}(c_{\star,>1}^2 + \sigma_\varepsilon^2)\Psi_3}}\right| \to 0$$

*with probability $1 - o(1)$, where $c_{\star,1}^2 = \mathsf{E}_z^2[\sigma'_\star(z)]$, $c_{\star,>1}^2 = \mathsf{E}_z[\sigma_{\star,\perp}^2(z)]$ with $z \sim \mathsf{N}(0, d_x^{-1}\,\mathrm{Tr}(\boldsymbol{\Sigma}_\mathbf{x}))$, and $\Psi_1, \Psi_2, \Psi_3$ are defined in (34) and depend on $\boldsymbol{\Sigma}_\mathbf{x}$, $d_x/n$, and $\lambda_\mathbf{G}$. In particular, as $\lambda_\mathbf{G} \to 0$ and $d_x/n \to 0$, we have*

$$\boldsymbol{\beta}_{\text{KFAC}}^\top\boldsymbol{\beta}_\star/(\|\boldsymbol{\beta}_{\text{KFAC}}\|_2\|\boldsymbol{\beta}_\star\|_2) \to 1.$$

This lemma shows that when $n \gg d_x$, and $\lambda_\mathbf{G} \to 0$, the one-step stylized KFAC update—unlike the one-step full-batch SGD—perfectly recovers the target direction $\boldsymbol{\beta}_\star$, fixing the issue with full batch SGD with anisotropic covariances.

*Remark* 3.13. It is well-known that, given features that align with $\boldsymbol{\beta}_\star$, applying least-squares on $\mathbf{Z} = \sigma(\mathbf{G}_{\text{KFAC}}\mathbf{X})$, which from Lemma 3.1 is equivalent to the KFAC $\mathbf{f}$-update with $\eta_\mathbf{f} = 1$, leverages the feature to obtain a solution with good generalization. See Appendix C.4 for more details.

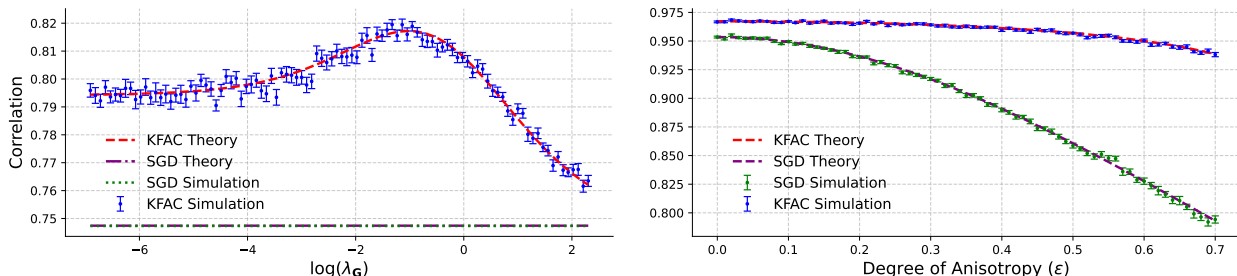

*Figure 2.* The correlation of the direction learned by SGD and KFAC with the the true direction by numerical simulations averaged over 30 trials, and theoretical predictions. **(Left)** For different values of $\lambda_\mathbf{G}$ the theoretical predictions match the simulations very well. **(Right)** The alignment of the feature learned by SGD deteriorates as anisotropy is increased (larger $\varepsilon$), whereas the KFAC update remains accurate.

## 4. Numerical Validation

### 4.1. Linear Representation Learning

We numerically study the behavior of different algorithms for a transfer learning setting (11), where the model is to be trained on data generated by $(\mathbf{F}_\star^{\mathrm{train}}, \mathbf{G}_\star)$, and the transfer task has data generated by $(\mathbf{F}_\star^{\mathrm{test}}, \mathbf{G}_\star)$, i.e. the embedding $\mathbf{G}_\star$ is shared, but the task heads $\mathbf{F}_\star^{\mathrm{train}}$ and $\mathbf{F}_\star^{\mathrm{test}}$ are different. The training and test covariates have anisotropic covariance matrices $\mathbf{\Sigma}_{\mathbf{x},\mathrm{train}}$ and $\mathbf{\Sigma}_{\mathbf{x},\mathrm{test}}$ respectively. Our data generation process for the training task and the transfer task are as follows:

$$\mathbf{y}_i^{\mathsf{s}} = \mathbf{F}_\star^{\mathsf{s}}\mathbf{G}_\star \mathbf{x}_i^{\mathsf{s}} + \boldsymbol{\varepsilon}_i^{\mathsf{s}}, \quad \mathbf{x}_i^{\mathsf{s}} \overset{\mathrm{i.i.d.}}{\sim} \mathbf{\Sigma}_{\mathbf{x},\mathsf{s}}^{1/2} \cdot \mathrm{Unif}(\{\pm 1\}^{\mathrm{d}_\mathbf{x}}),$$
$$\boldsymbol{\varepsilon}_i^{\mathsf{s}} \overset{\mathrm{i.i.d.}}{\sim} \mathsf{N}(0, \sigma_{\varepsilon,\mathsf{s}}^2 \mathbf{I}_{\mathrm{d}_\mathbf{Y}}), \quad \mathsf{s} \in \{\mathsf{test}, \mathsf{train}\}, \tag{13}$$

where $\sigma_{\varepsilon,\mathrm{train}} = 0.1$ and $\sigma_{\varepsilon,\mathrm{test}} = 1$. We use $\mathrm{d}_\mathbf{x} = 100$, $\mathrm{d}_\mathbf{Y} = 15$, $k = 8$, and batch size $n = 1024$. We present additional experiments and details in Appendix F, including discussions on the learning rates, and how $\mathbf{F}_\star^{\mathsf{s}}, \mathbf{G}_\star^{\mathsf{s}}, \mathbf{\Sigma}_{\mathbf{x},\mathsf{s}}$ are precisely generated.

**Head-to-head Evaluations.** We track the training loss, subspace distance, and transfer loss of different algorithms during the update (Figure 1). Alongside SGD, KFAC, Adam, and NGD, we also consider Alternating Min-SGD (AMGD) (Collins et al., 2021; Vaswani, 2024), and De-bias & Feature-Whiten (DFW) (Zhang et al., 2024b) (corresponding to (5) with $\mathbf{P}_\mathbf{F} = \mathbf{I}_{\mathrm{d}_\mathbf{Y}}$), two algorithms studied in linear representation learning. The transfer loss is the loss incurred by fitting a least-squares $\widehat{\mathbf{F}}_{\mathrm{ls}}^{\mathrm{test}}$ on the current $\mathbf{G}$ iterate (see Lemma 3.7). Although various algorithms converge on $\mathsf{s} = \mathsf{train}$, KFAC outperforms all others in terms of subspace distance and transfer loss, as suggested by the theory.

**Effect of Batch Normalization.** We track the subspace distance and the training loss of AMGD (with and without batch-norm) and KFAC, see Figure 3. As theoretically predicted in Section 3.1.1, since batch-norm approximately whitens $\mathbf{x}_i^{\mathrm{train}}$, AMGD+batch-norm converges in training loss. However, as predicted, it does not recover the correct representation, whereas KFAC does.

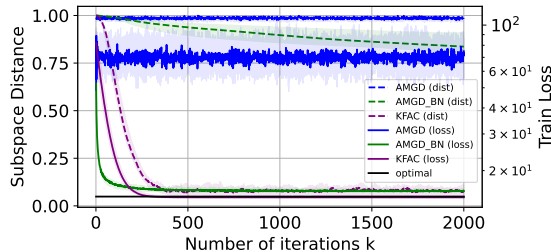

*Figure 3.* Subspace distance and the training loss of KFAC and AMGD (with and without batch-norm). Notably, batch-norm enables AMGD's train loss to converge, but not its subspace distance.

### 4.2. Single-Index Learning

Consider the single-index function learning setting of Section 3.2 with $\sigma_\star(z) = z + \frac{1}{\sqrt{2}}(z^2 - 1)$, and $\sigma_\varepsilon = 1$.

**Different Levels of Anisotropy.** In this experiment, we set $\mathrm{d}_\mathbf{x} = 200$, $n = 6000$, and $\mathrm{d}_\mathbf{h} = 1000$ and set $\lambda_\mathbf{G} \to 0$. For a parameter $\varepsilon \in \mathbb{R}$, we define $\mathbf{\Sigma}_\mathbf{x} = \mathbf{\Sigma}_\mathbf{x}^{(\varepsilon)}$ with

$$\mathbf{\Sigma}_\mathbf{x}^{(\varepsilon)} = \mathrm{diag}(\underbrace{1+\varepsilon, \ldots, 1+\varepsilon}_{\mathrm{d}_\mathbf{x}/2}, \underbrace{1-\varepsilon, \ldots, 1-\varepsilon}_{\mathrm{d}_\mathbf{x}/2}). \tag{14}$$

For different values of $\varepsilon$, we simulate the KFAC and SGD updates numerically and compute their correlation with the true direction. We also theoretically predict the correlation using Lemma 3.10 and 3.12; see Figure 2 (Right). The SGD update fails to recover the true direction in highly anisotropic settings (large $\varepsilon$), whereas the one-step KFAC update remains accurate.

**Theory vs. Simulations.** We set $\mathrm{d}_\mathbf{x} = 900$, $n = 5000$, $\mathrm{d}_\mathbf{h} = 1000$, and $\mathbf{\Sigma}_\mathbf{x} = \mathbf{\Sigma}_\mathbf{x}^{(0.5)}$. For different $\lambda_\mathbf{G}$, we simulate the correlation between the directions learned by KFAC and SGD with the true direction and compare it with predictions of Lemma 3.10 and 3.12; see Figure 2 (Left). We see that the theoretical results match very well with numerical simulations, even for moderately large $n, \mathrm{d}_\mathbf{x}$, and $\mathrm{d}_\mathbf{h}$. The direction learned by KFAC has a larger correlation with the true direction compared to that learned by SGD, as predicted.

## 5. Discussion

We study two models of feature learning in which we identify key issues of `SGD`-based feature learning approaches when departing from ideal settings. We then present Kronecker-Factored preconditioning—recovering variants of `KFAC`—to provably overcome these issues and derive improved guarantees. Our experiments on these simple models also confirm the suboptimality of full second-order methods, as well as the marginal benefit of `Adam` preconditioning and data normalization. We believe that analyzing properties of statistical learning problems can lead to fruitful insights into optimization and normalization schemes.

## Acknowledgments

Thomas Zhang and Behrad Moniri gratefully acknowledge gifts from AWS AI to Penn Engineering's ASSET Center for Trustworthy AI. The work of Behrad Moniri and Hamed Hassani is supported by The Institute for Learning-enabled Optimization at Scale (TILOS), under award number NSF-CCF-2112665, and the NSF CAREER award CIF-1943064. Thomas Zhang and Nikolai Matni are supported in part by NSF Award SLES-2331880, NSF CAREER award ECCS-2045834, NSF EECS-2231349, and AFOSR Award FA9550-24-1-0102.

## Impact Statement

The focus of this paper is on theoretical aspects of optimization and feature learning. We expect the results to be illuminating for the optimization and learning theory community. We do not anticipate any negative societal impact.

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

# A. Extended Background and Related Work

**Preconditioners for Neural Network Optimization.** A significant research effort in neural network optimization has been dedicated to understanding the role of preconditioning in convergence speed and generalization. Perhaps the most widespread paradigm falls under the category of *entry-wise* ("diagonal") preconditioners, whose notable members include `Adam` (Kingma & Ba, 2015), (diagonal) `AdaGrad` (Duchi et al., 2011), `RMSprop` (Tieleman & Hinton, 2012), and their innumerable relatives and descendants (see e.g. Schmidt et al. (2021); Dahl et al. (2023) for surveys). However, diagonal preconditioners inherently do not fully capture inter-parameter dependencies, which are better captured by stronger curvature estimates, e.g. Gauss-Newton approximations (Botev et al., 2017; Martens, 2020), L-BFGS (Byrd et al., 2016; Bollapragada et al., 2018; Goldfarb et al., 2020). Toward making non-diagonal preconditioners scalable to neural networks, many works (including the above) have made use of layer-wise *Kronecker-Factored* approximations, where each layer's curvature block is factored into a Kronecker product $\mathbf{Q} \otimes \mathbf{P}$. Perhaps the two most well-known examples are Kronecker-Factored Approximate Curvature (`KFAC`) (Martens & Grosse, 2015) and `Shampoo` (Gupta et al., 2018; Anil et al., 2020), where approximations are made to the Fisher Information and Gauss-Newton curvature, respectively. Many works have since expanded on these ideas, such as by improving practical efficiency (Ba et al., 2017; Shi et al., 2023; Jordan et al., 2024; Vyas et al., 2024) and exploring generalized constructions (Dangel et al., 2020; Amid et al., 2022; Benzing, 2022; Li, 2017; 2018; Pooladzandi & Li, 2024). An interesting alternate view subsumes certain preconditioners via steepest descent with respect to layer-wise ("modular") norms (Large et al., 2024; Bernstein & Newhouse, 2024a;b). We draw a connection therein by deriving the steepest descent norm that Kronecker-Factored preconditioners correspond to; see Appendix D.3.

**Multi-task Representation Learning (MTRL).** Toward a broader notion of generalization, the goal of MTRL is to characterize the benefits of learning a *shared* representation across distinct tasks. Various works focus on the generalization properties given access to an empirical risk minimizer (ERM) (Maurer et al., 2016; Du et al., 2021; Tripuraneni et al., 2020; Zhang et al., 2024a), with the latter work resolving the setting where distinct tasks may have different covariate distributions. Closely related formulations have been studied in the context of distribution shift (Kumar et al., 2022; Lee et al., 2023b). While these works consider general non-linear representations, access to an ERM obviates the (non-convex) optimization component. As such, multiple works have studied algorithms for linear representation learning (Tripuraneni et al., 2021b; Collins et al., 2021; Thekumparampil et al., 2021; Nayer & Vaswani, 2022) and specific non-linear variants (Collins et al., 2024; Nakhleh et al., 2024). In contrast to the ERM works, which are mostly agnostic to the covariate distribution, all the listed algorithmic works assume isotropic covariates $\mathsf{N}(\mathbf{0}, \mathbf{I})$. Zhang et al. (2024b) show that isotropy is in fact a key enabler, and propose an adjustment to handle general covariances. In this paper, we show that many prior linear representation learning algorithms belong to the same family of (preconditioned) optimizers. We then propose an algorithm coinciding with `KFAC` that achieves the first condition-number-free convergence rate.

**Nonlinear Feature Learning.** In the early phase of training, neural networks are shown to be essentially equivalent to the kernel methods, and can be described by the neural tangent kernel (NTK). See Jacot et al. (2018); Mei & Montanari (2022); Hu & Lu (2023). However, kernel methods are inherently limited and have a sample complexity superlinear in the input dimension $d$ for learning nonlinear functions (Ghorbani et al., 2021a;b). The main reason for this limitation is that kernel methods use a set of fixed features that are not task specific. There has been a lot of interest in studying the benefits of feature learning from a theoretical perspective (Bai & Lee (2020); Hanin & Nica (2020); Yang & Hu (2021a); Shi et al. (2022); Abbe et al. (2022), etc.). In a setting with isotropic covariates $\mathsf{N}(\mathbf{0}, \mathbf{I})$, it is shown that even a one-step of `SGD` update on the first layer of a two-layer neural networks can learn good enough features to provide a significant sample complexity improvement over kernel methods assuming that the target function has some low-dimensional structure (Damian et al., 2022; Ba et al., 2022; Moniri et al., 2024; Cui et al., 2024; Dandi et al., 2024a;b;c; Arnaboldi et al., 2024; Lee et al., 2024) and this has became a very popular model for studying feature learning. These results were later extended to three-layer neural networks in which the first layer is kept at random initialization and the second layer is updated using one step of `SGD` (Wang et al., 2024; Nichani et al., 2024a; Fu et al., 2024). Recently, Ba et al. (2024); Mousavi-Hosseini et al. (2023) considered an anisotropic case where the covariance contains a planted signal about the target function and showed that a single step of `SGD` can leverage this to better learn the target function. However, the general case of anisotropic covariate distributions remains largely unexplored. In this paper, we study feature learning with two-layer neural networks with general anisotropic covariates in single-index models and that one-step of `SGD` update has inherent limitations in this setting, and the natural fix will coincide with applying the `KFAC` layer-wise preconditioner.

# B. Proofs and Additional Details for Section 3.1

## B.1. Convergence Rate Lower Bound of SGD

Our goal is to establish the following lower bound construction.

**Proposition 3.5.** *Let* $\Sigma_{\mathbf{x}} = \mathbf{I}_{d_x}$, $n = \infty$. *Choose any* $d_x > k$, $d_y \geq k \geq 2$. *Let the learner be given knowledge of* $\mathbf{F}_\star, \mathbf{G}_\star$ *and* $\mathrm{dist}(\mathbf{G}_0, \mathbf{G}_\star)$. *However, assume the learner must fix* $\eta_{\mathbf{G}} > 0$ *before observing* $\mathbf{G}_0$. *Then, there exists* $\mathbf{F}_\star \in \mathbb{R}^{d_y \times k}$, $\mathbf{G}_\star, \mathbf{G}_0 \in \mathbb{R}^{k \times d_x}$, *such that* $\mathbf{G}_T = SGD(\mathbf{G}_0; \eta_{\mathbf{G}}, T)$ *satisfies:*

$$\mathrm{dist}(\mathbf{G}_T, \mathbf{G}_\star) \geq \left( 1 - 4 \frac{\lambda_{\min}(\mathbf{F}_\star^\top \mathbf{F}_\star)}{\lambda_{\max}(\mathbf{F}_\star^\top \mathbf{F}_\star)} \right)^T \mathrm{dist}(\mathbf{G}_0, \mathbf{G}_\star).$$

*Proof of Proposition 3.5.* We prove the lower bound by construction. First, we write out the one-step SGD update given step size $\eta_{\mathbf{G}}$.

$$\begin{aligned}
\overline{\mathbf{G}}_+ &= \mathbf{G} - \eta_{\mathbf{G}} \nabla_{\mathbf{G}} \widehat{\mathcal{L}}(\mathbf{F}, \mathbf{G}) \\
&= \mathbf{G} - \frac{1}{n} \mathbf{F}^\top \left( \mathbf{F} \mathbf{G} \mathbf{X}^\top \mathbf{X} - \mathbf{Y}^\top \mathbf{X} \right) \\
&= \mathbf{G} - \mathbf{F}^\top \left( \mathbf{F} \mathbf{G} - \mathbf{F}_\star \mathbf{G}_\star \right). \qquad (\mathbf{Y}^\top = \mathbf{F}_\star \mathbf{G}_\star \mathbf{X}^\top + \mathcal{E}^\top, n = \infty, \Sigma_{\mathbf{x}} = \mathbf{I}) \\
\mathbf{G}_+ &= \mathrm{Ortho}(\overline{\mathbf{G}}_+).
\end{aligned}$$

We recall $\mathbf{F}$ is given by the $\mathbf{F}$-update in (5) with $\eta_{\mathbf{F}} = 1$, which by Lemma 3.1 is equivalent to setting $\mathbf{F}$ to the least-squares solution conditional on $\mathbf{G}$:

$$\begin{aligned}
\mathbf{F} &= \mathbf{Y}^\top \mathbf{Z} \left( \mathbf{Z}^\top \mathbf{Z} \right)^{-1} = \mathbf{F}_\star \mathbf{G}_\star \Sigma_{\mathbf{x}} \mathbf{G}^\top \left( \mathbf{G} \Sigma_{\mathbf{x}} \mathbf{G}^\top \right)^{-1} \qquad (\mathbf{z} = \mathbf{G}\mathbf{x}, n = \infty) \\
&= \mathbf{F}_\star \mathbf{G}_\star \mathbf{G}^\top \left( \mathbf{G} \mathbf{G}^\top \right)^{-1} = \mathbf{F}_\star \mathbf{G}_\star \mathbf{G}^\top. \qquad (\mathbf{G} \text{ row-orthonormal})
\end{aligned}$$

Therefore, plugging in $\mathbf{F}$ into the SGD update yields:

$$\begin{aligned}
\overline{\mathbf{G}}_+ &= \mathbf{G} - \eta_{\mathbf{G}} \mathbf{F}^\top \left( \mathbf{F} \mathbf{G} - \mathbf{F}_\star \mathbf{G}_\star \right) \\
&= \mathbf{G} - \eta_{\mathbf{G}} \mathbf{G} \mathbf{G}_\star^\top \mathbf{F}_\star^\top \left( \mathbf{F}_\star \mathbf{G}_\star \mathbf{G}^\top \mathbf{G} - \mathbf{F}_\star \mathbf{G}_\star \right)
\end{aligned}$$

Before proceeding, let us present the construction of $\mathbf{F}_\star, \mathbf{G}_\star$. We focus on the case $d_x = 3$, $k = 2$, as it will be clear the construction is trivially embedded to arbitrary $d_x > k \geq 2$. We observe that $\mathbf{F}_\star$ only appears in the SGD update in the form $\mathbf{F}_\star^\top \mathbf{F}_\star \in \mathbb{R}^{k \times k}$, thus $d_y \geq k$ can be set arbitrarily as long as $\mathbf{F}_\star^\top \mathbf{F}_\star$ satisfies our specifications. Set $\mathbf{F}_\star, \mathbf{G}_\star$ such that

$$\mathbf{F}_\star^\top \mathbf{F}_\star = \begin{bmatrix} 1 - \lambda & 0 \\ 0 & \lambda \end{bmatrix}, \lambda \in (0, 1/2], \quad \mathbf{G}_\star = \begin{bmatrix} 1 & 0 & 0 \\ 0 & 1 & 0 \end{bmatrix}.$$

Accordingly, the initial representation $\mathbf{G}_0$ (which the learner is not initially given) will have form

$$\mathbf{G}_0 = \begin{bmatrix} 1 & 0 & 0 \\ 0 & \sqrt{1 - \varepsilon_0^2} & \varepsilon_0 \end{bmatrix}, \text{ or } \begin{bmatrix} \sqrt{1 - \varepsilon_0^2} & 0 & \varepsilon_0 \\ 0 & 1 & 0 \end{bmatrix}.$$

We prove all results with the first form of $\mathbf{G}_0$, as all results will hold for the second with the only change swapping $\lambda, 1 - \lambda$. It is clear that we may extend to arbitrary $d_x > k \geq 2$ by setting:

$$\mathbf{F}_\star^\top \mathbf{F}_\star = \begin{bmatrix} (1 - \lambda) \mathbf{I}_{k-1} & \mathbf{0} \\ \mathbf{0} & \lambda \end{bmatrix}, \quad \mathbf{G}_\star = \begin{bmatrix} \mathbf{I}_k & \mathbf{0}_{d_x - k} \end{bmatrix}, \quad \mathbf{G}_0 = \begin{bmatrix} \mathbf{I}_{k-1} & \mathbf{0} & \cdots & \\ 0 & \sqrt{1 - \varepsilon_0^2} & \varepsilon_0 & \mathbf{0} \end{bmatrix}.$$

Returning to the $d_x = 3, k = 2$ case, we first prove the following invariance result.

**Lemma B.1.** *Given* $\mathbf{G}_0 = \begin{bmatrix} 1 & 0 & 0 \\ 0 & \sqrt{1 - \varepsilon_0^2} & \varepsilon_0 \end{bmatrix}$, *then for any* $t \geq 0$, $\mathbf{G}_t = \begin{bmatrix} 1 & 0 & 0 \\ 0 & c_1 & c_2 \end{bmatrix}$ *for some* $c_1^2 + c_2^2 = 1$. *Furthermore, we have* $\mathrm{dist}(\mathbf{G}_t, \mathbf{G}_\star) = |c_2|$.

*Proof of Lemma B.1.* This follows by induction. The base case follows by definition of $\mathbf{G}_0$. Now given $\mathbf{G}_t = \begin{bmatrix} 1 & 0 & 0 \\ 0 & c_1 & c_2 \end{bmatrix}$ for some $c_1, c_2$, we observe that

$$\mathbf{G}_t\mathbf{G}_\star^\top = \begin{bmatrix} 1 & 0 \\ 0 & c_1 \end{bmatrix}, \quad \mathbf{F}_\star^\top\mathbf{F}_\star = \begin{bmatrix} 1-\lambda & 0 \\ 0 & \lambda \end{bmatrix}.$$

Notably, we may write

$$\begin{aligned}
\overline{\mathbf{G}}_{t+1} &= \mathbf{G}_t - \eta_{\mathbf{G}}\mathbf{G}_t\mathbf{G}_\star^\top\mathbf{F}_\star^\top(\mathbf{F}_\star\mathbf{G}_\star\mathbf{G}_t^\top\mathbf{G}_t - \mathbf{F}_\star\mathbf{G}_\star) \\
&= \left(\mathbf{I}_k - \eta_{\mathbf{G}}\mathbf{G}_t\mathbf{G}_\star^\top\mathbf{F}_\star^\top\mathbf{F}_\star\mathbf{G}_\star\mathbf{G}_t^\top\right)\mathbf{G}_t + \eta_{\mathbf{G}}\mathbf{G}_t\mathbf{G}_\star^\top\mathbf{F}_\star^\top\mathbf{F}_\star\mathbf{G}_\star \\
&= \left(\mathbf{I}_k - \eta_{\mathbf{G}}\begin{bmatrix} 1-\lambda & 0 \\ 0 & c_1^2\lambda \end{bmatrix}\right)\mathbf{G}_t + \eta_{\mathbf{G}}\begin{bmatrix} 1-\lambda & 0 \\ 0 & c_1\lambda \end{bmatrix}\mathbf{G}_\star \\
&= \begin{bmatrix} 1 & 0 & 0 \\ 0 & c_1(1+\eta_{\mathbf{G}}c_2^2\lambda) & c_2(1-\eta_{\mathbf{G}}c_1^2\lambda) \end{bmatrix}
\end{aligned} \tag{15}$$

$$\mathbf{G}_{t+1} = \mathrm{Ortho}(\overline{\mathbf{G}}_{t+1}).$$

Therefore, $\mathbf{G}_{t+1}$ shares the same support as $\mathbf{G}_t$, and by the orthonormalization step, the squared entries of the second row of $\mathbf{G}_{t+1}$ equal 1, completing the induction step.

To prove the second claim, we see that $\mathcal{P}_\star^\perp = \begin{bmatrix} \mathbf{0}_2 & \\ & 1 \end{bmatrix}$, and since $\mathbf{G}_t$ is by assumption row-orthonormal, we have

$$\mathrm{dist}(\mathbf{G}_t, \mathbf{G}_\star) = \|\mathbf{G}_t\mathcal{P}_\star^\perp\| = \left\| \begin{bmatrix} 0 & 0 & c_2 \end{bmatrix}^\top \right\|_{\mathrm{op}} = |c_2|,$$

completing the proof. $\qquad\square$

With these facts in hand, we prove the following stability limit of the step-size, and the consequences for the contraction rate.

**Lemma B.2.** *If $\eta_{\mathbf{G}} \geq \frac{4}{1-\lambda}$, then for any given $\mathrm{dist}(\mathbf{G}_0, \mathbf{G}_\star)$ we may find $\mathbf{G}_0$ such that $\limsup_t \mathrm{dist}(\mathbf{G}_t, \mathbf{G}_\star) \geq \frac{1}{2}$.*

*Proof of Lemma B.2.* By assumption $\lambda \leq 1/2$ and thus $\lambda \leq 1 - \lambda$. Evaluating Lemma B.1 instead on $\mathbf{G}_0 = \begin{bmatrix} \sqrt{1-\varepsilon_0^2} & 0 & \varepsilon_0 \\ 0 & 1 & 0 \end{bmatrix}$, writing out (15) yields symmetrically:

$$\mathrm{dist}(\mathbf{G}_t, \mathbf{G}_\star) = |c_2|$$

$$\overline{\mathbf{G}}_{t+1} = \begin{bmatrix} c_1(1+\eta_{\mathbf{G}}c_2^2(1-\lambda)) & 0 & c_2(1-\eta_{\mathbf{G}}c_1^2(1-\lambda)) \\ 0 & 1 & 0 \end{bmatrix}$$

$$\mathbf{G}_{t+1} = \mathrm{Ortho}(\overline{\mathbf{G}}_{t+1}).$$

We first observe that regardless of $\eta_{\mathbf{G}}$, the norm of the first row $\mathbf{G}_{t+1}$ is always greater than 1 pre-orthonormalization. Let us define $\omega = \eta_{\mathbf{G}}(1-\lambda)$. Then, the squared-norm of the first row satisfies:

$$\left(c_1(1+\omega c_2^2)\right)^2 + \left(c_2(1-\omega c_1^2)\right)^2 = 1 + \omega^2 c_1^2 c_2^2.$$

Therefore, the norm is strictly bounded away from 1 when $\omega > 0$ and either $c_1, c_2 \neq 0$ by the constraint $c_1^2 + c_2^2 = 1$. Importantly, this implies that regardless of the step-size taken, the resulting first-row norm of $\mathbf{G}_+$ must exceed 1 prior to orthonormalization. Given this property, we observe that for $\omega \geq 1/c_1^2$, we have:

$$\frac{|1-\omega c_1^2|}{|1+\omega c_2^2|} = \frac{\omega c_1^2 - 1}{1 + \omega c_2^2}.$$

When this ratio is greater than 1, we are guaranteed that the first-row coefficients $c_1', c_2'$ of $\mathbf{G}_{t+1}$ *post*-orthonormalization satisfy $c_2'/c_1' > c_2/c_1$, and recall from Lemma B.1 $\mathrm{dist}(\mathbf{G}_{t+1}, \mathbf{G}_\star) = c_2'$, and thus $\mathrm{dist}(\mathbf{G}_{t+1}, \mathbf{G}_\star) > \mathrm{dist}(\mathbf{G}_t, \mathbf{G}_\star)$. Rearranging the above ratio, this is equivalent to the condition $\omega = \eta_{\mathbf{G}}(1-\lambda) \geq \frac{2}{c_1^2-c_2^2}$, $\omega = \eta_{\mathbf{G}}(1-\lambda) \geq 1/c_1^2$. Setting $c_1^2 = 3/4, c_2^2 = 1/4$, this implies for $\eta_{\mathbf{G}} \geq \frac{4}{1-\lambda}$, the moment $\mathrm{dist}(\mathbf{G}_t, \mathbf{G}_\star) \leq c_2 = 1/2$, then we are guaranteed $\mathrm{dist}(\mathbf{G}_{t+1}, \mathbf{G}_\star) > \mathrm{dist}(\mathbf{G}_t, \mathbf{G}_\star)$, and thus $\limsup_{t\to\infty} \mathrm{dist}(\mathbf{G}_t, \mathbf{G}_\star) \geq \frac{1}{2}$, irregardless of $\mathrm{dist}(\mathbf{G}_0, \mathbf{G}_\star)$. $\qquad\square$

Now, to finish the construction of the lower bound, Lemma B.2 establishes that $\eta_{\mathbf{G}} \leq \frac{4}{1-\lambda}$ is necessary for convergence (though not sufficient!). This implies that when we plug back in $\mathbf{G}_0 = \begin{bmatrix} 1 & 0 & 0 \\ 0 & \sqrt{1-\varepsilon_0^2} & \varepsilon_0 \end{bmatrix}$, we have $\mathbf{G}_{t+1}$:

$$\overline{\mathbf{G}}_{t+1} = \begin{bmatrix} 1 & 0 & 0 \\ 0 & c_1(1 + \eta_{\mathbf{G}}c_2^2\lambda) & c_2(1 - \eta_{\mathbf{G}}c_1^2\lambda) \end{bmatrix}$$

$$\mathbf{G}_{t+1} = \mathrm{Ortho}(\overline{\mathbf{G}}_{t+1}) = \begin{bmatrix} 1 & 0 & 0 \\ 0 & c_1' & c_2' \end{bmatrix}.$$

We have trivially $1 - \eta_{\mathbf{G}}c_1^2\lambda \geq 1 - \eta_{\mathbf{G}}\lambda$. Therefore, for $\lambda \leq 1/5$ such that $\frac{\lambda}{1-\lambda} \leq 1/4$, we have $1 - \eta_{\mathbf{G}}\lambda \geq 1 - \frac{4\lambda}{1-\lambda} \geq 0$. As shown in the proof of Lemma B.2, the norm of the second row pre-orthonormalization is strictly greater than 1, and thus:

$$\mathrm{dist}(\mathbf{G}_{t+1}, \mathbf{G}_\star) = c_2' \geq c_2(1 - \eta_{\mathbf{G}}\lambda c_1^2) \geq (1 - \eta_{\mathbf{G}}\lambda)\mathrm{dist}(\mathbf{G}_t, \mathbf{G}_\star) \geq \left(1 - 4\frac{\lambda}{1-\lambda}\right)\mathrm{dist}(\mathbf{G}_t, \mathbf{G}_\star).$$

Applying this recursively to $\mathbf{G}_0$ yields the desired lower bound. $\qquad\square$

## B.2. Proof of Theorem 3.6

Recall that running an iteration of stylized KFAC (5) with $\lambda_{\mathbf{F}}, \lambda_{\mathbf{G}} = 0$, $\eta_{\mathbf{F}} = 1$ yields:

$$\begin{aligned} \overline{\mathbf{G}}_+ &= \mathbf{G} - \eta_{\mathbf{G}}\mathbf{P}_{\mathbf{G}}^{-1} \nabla_{\mathbf{G}}\widehat{\mathcal{L}}(\mathbf{F}, \mathbf{G})(\mathbf{Q}_{\mathbf{G}} + \lambda_{\mathbf{G}}\mathbf{I}_{d_{\mathsf{x}}})^{-1} \\ &= \mathbf{G} - \eta_{\mathbf{G}}(\mathbf{F}^\top\mathbf{F})^{-1}\mathbf{F}^\top(\mathbf{F}\mathbf{G}\widehat{\boldsymbol{\Sigma}}_{\mathbf{x}} - \mathbf{F}_\star\mathbf{G}_\star\widehat{\boldsymbol{\Sigma}}_{\mathbf{x}} - \frac{1}{n}\mathcal{E}^\top\mathbf{X})\widehat{\boldsymbol{\Sigma}}_{\mathbf{x}}^{-1} \\ &= \mathbf{G} - \eta_{\mathbf{G}}(\mathbf{F}^\top\mathbf{F})^{-1}\mathbf{F}^\top(\mathbf{F}\mathbf{G} - \mathbf{F}_\star\mathbf{G}_\star) + (\mathbf{F}^\top\mathbf{F})^{-1}\mathbf{F}^\top\mathcal{E}^\top\mathbf{X}(\mathbf{X}^\top\mathbf{X})^{-1}, \end{aligned} \tag{16}$$

where the matrix $\mathbf{F}$ is given by

$$\begin{aligned} \mathbf{F} &= \mathbf{F}_{\mathrm{prev}} - \eta_{\mathbf{F}}\mathbf{P}_{\mathbf{F}}^{-1} \nabla_{\mathbf{F}}\widehat{\mathcal{L}}(\mathbf{F}_{\mathrm{prev}}, \mathbf{G})(\mathbf{Q}_{\mathbf{F}} + \lambda_{\mathbf{F}}\mathbf{I}_{d_{\mathsf{h}}})^{-1} \\ &= \mathbf{Y}^\top\mathbf{Z}(\mathbf{Z}^\top\mathbf{Z})^{-1} \\ &= \mathbf{F}_\star\mathbf{G}_\star\mathbf{X}^\top\mathbf{Z}(\mathbf{Z}^\top\mathbf{Z})^{-1} + \mathcal{E}^\top\mathbf{Z}(\mathbf{Z}^\top\mathbf{Z})^{-1}, \end{aligned}$$

recalling that $\mathbf{z} \triangleq \mathbf{G}\mathbf{x}$. Focusing on the representation update, we have

$$\overline{\mathbf{G}}_+\mathcal{P}_\star^\perp = (1 - \eta_{\mathbf{G}})\mathbf{G}\mathcal{P}_\star^\perp + \eta_{\mathbf{G}}(\mathbf{F}^\top\mathbf{F})^{-1}\mathbf{F}^\top\mathcal{E}^\top\mathbf{X}(\mathbf{X}^\top\mathbf{X})^{-1}.$$

Therefore, to prove a one-step contraction of $\mathrm{rowsp}(\mathbf{G}_+)$ toward $\mathrm{rowsp}(\mathbf{G}_\star)$, we require two main components:

- Bounding the noise term $\eta_{\mathbf{G}}(\mathbf{F}^\top\mathbf{F})^{-1}\mathbf{F}^\top\mathcal{E}^\top\mathbf{X}^\top(\mathbf{X}^\top\mathbf{X})^{-1}$.

- Bounding the orthonormalization factor; the subspace distance measures distance between two orthonormalized bases (a.k.a. elements of the Stiefel manifold (Absil et al., 2008)), while a step of SGD or KFAC does not inherently conform to the Stiefel manifold, and thus the "off-manifold" shift must be considered when computing $\mathrm{dist}(\mathbf{G}_+, \mathbf{G}_\star)$. This amounts to bounding the "$\mathbf{R}$"-factor of the $\mathbf{QR}$-decomposition (Trefethen & Bau, 2022) of $\overline{\mathbf{G}}_+$.

Thanks to the left-preconditioning by $(\mathbf{F}^\top\mathbf{F})^{-1}$, the contraction factor is essentially determined by $(1 - \eta_{\mathbf{G}})$; however, the second point about the "off-manifold" shift is what prevents us from setting $\eta_{\mathbf{G}} = 1$.

### Bounding the noise term

We start by observing $\mathcal{E}^\top\mathbf{X}(\mathbf{X}^\top\mathbf{X})^{-1} = \sum_{i=1}^n \varepsilon_i\mathbf{x}_i^\top \left(\sum_{i=1}^n \mathbf{x}_i\mathbf{x}_i^\top\right)^{-1}$ (and thus $(\mathbf{F}^\top\mathbf{F})^{-1}\mathbf{F}^\top\mathcal{E}^\top\mathbf{X}(\mathbf{X}^\top\mathbf{X})^{-1}$) is a least-squares error-like term, and thus can be bounded by standard self-normalized martingale arguments. In particular, defining $\overline{\mathbf{F}} \triangleq (\mathbf{F}^\top\mathbf{F})^{-1}\mathbf{F}^\top$ we may decompose

$$\left\|\overline{\mathbf{F}}\mathcal{E}^\top\mathbf{X}(\mathbf{X}^\top\mathbf{X})^{-1}\right\|_{\mathrm{op}} \leq \left\|\overline{\mathbf{F}}\mathcal{E}^\top\mathbf{X}(\mathbf{X}^\top\mathbf{X})^{-1/2}\right\|_{\mathrm{op}} \lambda_{\min}(\mathbf{X}^\top\mathbf{X})^{-1/2},$$

where the first factor is the aforementioned self-normalized martingale (see e.g. Abbasi-Yadkori & Szepesvári (2011); Ziemann et al. (2023)), and the second can be bounded by standard covariance lower-tail bounds. Toward bounding the first factor, we invoke a high-probability self-normalized bound:

**Lemma B.3** (cf. Ziemann et al. (2023, Theorem 4.1)). *Let $\{\mathbf{v}_i, \mathbf{w}_i\}_{i \geq 1}$ be a $\mathbb{R}^{d_v} \times \mathbb{R}^{d_w}$-valued process and $\{\mathcal{F}_i\}_{i \geq 1}$ be a filtration such that $\{\mathbf{v}_i\}_{i \geq 1}$ is adapted to $\{\mathcal{F}_i\}_{i \geq 1}$, $\{\mathbf{w}_i\}_{i \geq 1}$ is adapted to $\{\mathcal{F}_i\}_{i \geq 2}$, and $\{\mathbf{w}_i\}_{i \geq 1}$ is a $\sigma^2$-subgaussian martingale difference sequence[6]. Fix (non-random) positive-definite matrix $\mathbf{Q}$. For $k \geq 1$, define $\widehat{\boldsymbol{\Sigma}}_k \triangleq \sum_{i=1}^{k} \mathbf{v}_i \mathbf{v}_i^\top$. Then, given any fixed $n \in \mathbb{N}_+$, with probability at least $1 - \delta$:*

$$\left\| \sum_{i=1}^{n} \mathbf{w}_i \mathbf{v}_i^\top \left( \mathbf{Q} + \widehat{\boldsymbol{\Sigma}}_n \right)^{-1/2} \right\|_{\text{op}}^2 \leq 4\sigma^2 \log \left( \frac{\det \left( \mathbf{Q} + \widehat{\boldsymbol{\Sigma}}_n \right)}{\det(\mathbf{Q})} \right) + 13 d_w \sigma^2 + 8\sigma^2 \log(1/\delta). \tag{17}$$

Instantiating this for Gaussian $\mathbf{w}_i \overset{\text{i.i.d.}}{\sim} \mathsf{N}(\mathbf{0}, \boldsymbol{\Sigma}_\mathbf{w})$, $\mathbf{v}_i \overset{\text{i.i.d.}}{\sim} \mathsf{N}(\mathbf{0}, \boldsymbol{\Sigma}_\mathbf{v})$, we may set $\mathbf{Q} \approx \boldsymbol{\Sigma}_\mathbf{v}$ to yield:

**Lemma B.4.** *Consider the quantities defined in Lemma B.3 and assume $\mathbf{w}_i \overset{\text{i.i.d.}}{\sim} \mathsf{N}(\mathbf{0}, \boldsymbol{\Sigma}_\mathbf{w})$, $\mathbf{v}_i \overset{\text{i.i.d.}}{\sim} \mathsf{N}(\mathbf{0}, \boldsymbol{\Sigma}_\mathbf{v})$, defining $\sigma_\mathbf{w}^2 \triangleq \lambda_{\max}(\boldsymbol{\Sigma}_\mathbf{w})$, $\sigma_\mathbf{v}^2 \triangleq \lambda_{\max}(\boldsymbol{\Sigma}_\mathbf{v})$. Then, as long as $n \gtrsim \frac{18.27}{c^2}(d_v + \log(1/\delta))$, with probability at least $1 - \delta$:*

$$\left\| \sum_{i=1}^{n} \mathbf{w}_i \mathbf{v}_i^\top \left( \widehat{\boldsymbol{\Sigma}}_n \right)^{-1/2} \right\|_{\text{op}}^2 \leq 8 d_v \log \left( \frac{1+c}{1-c} \right) \sigma_\mathbf{w}^2 + 26 d_w \sigma_\mathbf{w}^2 + 16\sigma_\mathbf{w}^2 \log(1/\delta)$$

$$\lambda_{\min}(\widehat{\boldsymbol{\Sigma}}_n) \geq (1 - c) \lambda_{\min}(\boldsymbol{\Sigma}_\mathbf{v}).$$

*Proof of Lemma B.4.* We observe that if $\widehat{\boldsymbol{\Sigma}}_n \succeq \mathbf{Q}$, then

$$2\widehat{\boldsymbol{\Sigma}}_n \succeq \mathbf{Q} + \widehat{\boldsymbol{\Sigma}}_n \implies \left( \widehat{\boldsymbol{\Sigma}}_n \right)^{-1} \preceq 2 \left( \mathbf{Q} + \widehat{\boldsymbol{\Sigma}}_n \right)^{-1}.$$

This implies

$$\mathbf{1}\left\{ \widehat{\boldsymbol{\Sigma}}_n \succeq \mathbf{Q} \right\} \left\| \sum_{i=1}^{n} \mathbf{w}_i \mathbf{v}_i^\top \left( \widehat{\boldsymbol{\Sigma}}_n \right)^{-1/2} \right\|^2 \leq 2 \mathbf{1}\left\{ \widehat{\boldsymbol{\Sigma}}_n \succeq \mathbf{Q} \right\} \left\| \sum_{i=1}^{n} \mathbf{w}_i \mathbf{v}_i^\top \left( \mathbf{Q} + \widehat{\boldsymbol{\Sigma}}_n \right)^{-1/2} \right\|^2. \tag{18}$$

Let us consider the event:

$$(1 - c)\boldsymbol{\Sigma}_\mathbf{v} \preceq \widehat{\boldsymbol{\Sigma}}_n \preceq (1 + c)\boldsymbol{\Sigma}_\mathbf{v},$$

which by Lemma E.2 occurs with probability at least $1 - \delta$ as long as $n \gtrsim \frac{18.27}{c^2}(d_v + \log(1/\delta))$. This immediately establishes the latter desired inequality. Setting $\mathbf{Q} = (1 - c)n\boldsymbol{\Sigma}_\mathbf{v}$ and conditioning on the above event, we observe that by definition $\widehat{\boldsymbol{\Sigma}}_n \succeq \mathbf{Q}$, and

$$\log \left( \frac{\det \left( \mathbf{Q} + \widehat{\boldsymbol{\Sigma}}_n \right)}{\det(\mathbf{Q})} \right) = \log \det \left( I_{d_v} + \widehat{\boldsymbol{\Sigma}}_n (\mathbf{Q})^{-1} \right)$$

$$\leq \log \det \left( \left( 1 + \frac{1+c}{1-c} \right) I_{d_v} \right)$$

$$\leq d_v \log \left( \frac{1+c}{1-c} \right).$$

Plugging this into Lemma B.3, applied to the RHS of (18), we get our desired result. $\qquad \square$

Therefore, instantiating $\mathbf{w} \to \overline{\mathbf{F}}\boldsymbol{\varepsilon}$, $\mathbf{v} \to \mathbf{x}$, $(\mathbf{x}_i, \boldsymbol{\varepsilon}_i)_{i \geq 1}$ is a $\mathbb{R}^{d_x} \times \mathbb{R}^k$-valued process. Furthermore, since we assumed out of convenience that $\mathbf{F}, \mathbf{G}_+$ are computed on independent batches of data, we have that $\overline{\mathbf{F}}\boldsymbol{\varepsilon} \sim \mathsf{N}(\mathbf{0}, \overline{\mathbf{F}}\boldsymbol{\Sigma}_\varepsilon \overline{\mathbf{F}}^\top)$. In order to complete the noise term bound, it suffices to provide a uniform bound on $\|\overline{\mathbf{F}}\| = 1/\sigma_{\min}(\mathbf{F})$ in terms of $\mathbf{F}_\star$.

---

[6]See Appendix E.3 for discussion of formalism. It suffices for our purposes to consider $\mathbf{w} \overset{\text{i.i.d.}}{\sim} \mathsf{N}(\mathbf{0}, \boldsymbol{\Sigma}_\mathbf{w})$.

**Lemma B.5.** *Assume the following conditions hold:*

$$n \gtrsim \max\left\{k + \log(1/\delta),\ \sigma_{\boldsymbol{\varepsilon}}^2 \frac{d_Y + k + \log(1/\delta)}{\sigma_{\min}(\mathbf{F}_\star)^2\,\lambda_{\min}(\boldsymbol{\Sigma}_{\mathbf{x}})}\right\}$$

$$\mathrm{dist}(\mathbf{G}, \mathbf{G}_\star) \leq \frac{2}{5}\kappa(\mathbf{F}_\star)^{-1}\kappa(\boldsymbol{\Sigma}_{\mathbf{x}})^{-1},$$

*then with probability at least $1 - \delta$, we have $\|\overline{\mathbf{F}}\| = 1/\sigma_{\min}(\mathbf{F}) \leq 2\,\sigma_{\min}(\mathbf{F}_\star)^{-1}$.*

*Proof of Lemma B.5.* Recall we may write $\mathbf{F}$ as

$$\begin{aligned}
\mathbf{F} &= \mathbf{F}_\star\mathbf{G}_\star\mathbf{X}^\top\mathbf{Z}(\mathbf{Z}^\top\mathbf{Z})^{-1} + \mathcal{E}^\top\mathbf{Z}(\mathbf{Z}^\top\mathbf{Z})^{-1} \\
&= \mathbf{F}_\star\mathbf{G}_\star\mathbf{G}^\top + \mathbf{F}_\star\mathbf{G}_\star(\mathbf{I}_{d_X} - \mathbf{G}^\top\mathbf{G})\mathbf{X}^\top\mathbf{Z}(\mathbf{Z}^\top\mathbf{Z})^{-1} + \mathcal{E}^\top\mathbf{Z}(\mathbf{Z}^\top\mathbf{Z})^{-1}
\end{aligned} \tag{19}$$

By Weyl's inequality for singular values (Horn & Johnson, 2012), we have

$$\sigma_{\min}(\mathbf{F}) \geq \sigma_{\min}(\mathbf{F}_\star\mathbf{G}_\star\mathbf{G}^\top) - \sigma_{\max}\left(\mathbf{F}_\star\mathbf{G}_\star(\mathbf{I}_{d_X} - \mathbf{G}^\top\mathbf{G})\mathbf{X}^\top\mathbf{Z}(\mathbf{Z}^\top\mathbf{Z})^{-1} + \mathcal{E}^\top\mathbf{Z}(\mathbf{Z}^\top\mathbf{Z})^{-1}\right)$$

Since $\mathbf{G}_\star\mathbf{G}^\top$ is an orthogonal matrix, the first term is equal to $\sigma_{\min}(\mathbf{F}_\star)$. On the other hand, applying triangle inequality on the second term, for $n \gtrsim k + \log(1/\delta)$ we have:

$$\begin{aligned}
\left\|\mathbf{F}_\star\mathbf{G}_\star(\mathbf{I}_{d_X} - \mathbf{G}^\top\mathbf{G})\mathbf{X}^\top\mathbf{Z}(\mathbf{Z}^\top\mathbf{Z})^{-1}\right\|_{\mathrm{op}} &\leq \|\mathbf{F}_\star\|_{\mathrm{op}}\left\|\mathbf{G}_\star(\mathbf{I}_{d_X} - \mathbf{G}^\top\mathbf{G})\right\|_{\mathrm{op}}\left\|\mathbf{X}^\top\mathbf{Z}(\mathbf{Z}^\top\mathbf{Z})^{-1}\right\|_{\mathrm{op}} \\
&\leq \|\mathbf{F}_\star\|_{\mathrm{op}}\,\mathrm{dist}(\mathbf{G}, \mathbf{G}_\star)\left(\frac{5}{4}\|\boldsymbol{\Sigma}_{\mathbf{x}}\|_{\mathrm{op}}\,\lambda_{\min}(\mathbf{G}\boldsymbol{\Sigma}_{\mathbf{x}}\mathbf{G}^\top)\right) \\
&\leq \frac{5}{4}\|\mathbf{F}_\star\|_{\mathrm{op}}\,\mathrm{dist}(\mathbf{G}, \mathbf{G}_\star)\kappa(\boldsymbol{\Sigma}_{\mathbf{x}}),
\end{aligned}$$

where we used covariance concentration for the second inequality Lemma E.2 and the trivial bound $\lambda_{\min}(\mathbf{A}\boldsymbol{\Sigma}\mathbf{A}^\top) \geq \lambda_{\min}(\boldsymbol{\Sigma})$ for the last inequality. In turn, we may bound:

$$\begin{aligned}
\left\|\mathcal{E}^\top\mathbf{Z}(\mathbf{Z}^\top\mathbf{Z})^{-1}\right\|_{\mathrm{op}} &\leq \left\|\mathcal{E}^\top\mathbf{Z}(\mathbf{Z}^\top\mathbf{Z})^{-1/2}\right\|_{\mathrm{op}}\lambda_{\min}(\mathbf{Z}^\top\mathbf{Z})^{-1/2} \\
&\lesssim \left\|\mathcal{E}^\top\mathbf{Z}(\mathbf{Z}^\top\mathbf{Z})^{-1/2}\right\|_{\mathrm{op}}n^{-1/2}\,\lambda_{\min}(\boldsymbol{\Sigma}_{\mathbf{x}})^{-1/2} &\text{(Lemma E.2)} \\
&\lesssim \sigma_{\boldsymbol{\varepsilon}}\sqrt{\frac{d_Y + k + \log(1/\delta)}{\lambda_{\min}(\boldsymbol{\Sigma}_{\mathbf{x}})n}}.
\end{aligned}$$

Therefore, setting $\mathrm{dist}(\mathbf{G}, \mathbf{G}_\star) \leq \frac{2}{5}\kappa(\mathbf{F}_\star)^{-1}\kappa(\boldsymbol{\Sigma}_{\mathbf{x}})^{-1}$, and $n \gtrsim \sigma_{\boldsymbol{\varepsilon}}^2\frac{d_Y+k+\log(1/\delta)}{\sigma_{\min}(\mathbf{F}_\star)^2\,\lambda_{\min}(\boldsymbol{\Sigma}_{\mathbf{x}})}$, we have $\sigma_{\min}(\mathbf{F}) \geq \frac{1}{2}\sigma_{\min}(\mathbf{F}_\star)$, which leads to our desired bound on $\|\overline{\mathbf{F}}\|$. $\qquad\square$

With a bound on $\|\overline{\mathbf{F}}\|_{\mathrm{op}}$, bounding the noise term is a straightforward application of Lemma B.4.

**Proposition B.6** (KFAC noise term bound)**.** *Let the conditions in Lemma B.5 hold. In addition, assume $n \gtrsim d_X + \log(1/\delta)$. Then, with probability at least $1 - \delta$:*

$$\left\|\overline{\mathbf{F}}\mathcal{E}^\top\mathbf{X}(\mathbf{X}^\top\mathbf{X})^{-1}\right\|_{\mathrm{op}} \lesssim \sigma_{\boldsymbol{\varepsilon}}\sqrt{\frac{d_X + k + \log(1/\delta)}{\sigma_{\min}(\mathbf{F}_\star)^2\,\lambda_{\min}(\boldsymbol{\Sigma}_{\mathbf{x}})n}}.$$

*Proof of Proposition B.6.* Condition on the event of Lemma B.5. Then, assuming $n \gtrsim d_X + \log(1/\delta)$, we may apply covariance concentration (Lemma E.2) on $\widehat{\boldsymbol{\Sigma}}_{\mathbf{x}}$ and Lemma B.4 to bound the noise term by:

$$\begin{aligned}
\left\|\overline{\mathbf{F}}\mathcal{E}^\top\mathbf{X}(\mathbf{X}^\top\mathbf{X})^{-1}\right\|_{\mathrm{op}} &\leq \left\|\overline{\mathbf{F}}\mathcal{E}^\top\mathbf{X}(\mathbf{X}^\top\mathbf{X})^{-1/2}\right\|_{\mathrm{op}}\lambda_{\min}(\mathbf{X}^\top\mathbf{X})^{-1/2} \\
&\leq \|\overline{\mathbf{F}}\|_{\mathrm{op}}\left\|\mathcal{E}^\top\mathbf{X}(\mathbf{X}^\top\mathbf{X})^{-1/2}\right\|_{\mathrm{op}}n^{-1/2}\,\lambda_{\min}(\boldsymbol{\Sigma}_{\mathbf{x}})^{-1/2} &\text{(Lemma E.2)} \\
&\lesssim \sigma_{\boldsymbol{\varepsilon}}\sqrt{\frac{d_X + k + \log(1/\delta)}{\sigma_{\min}(\mathbf{F}_\star)^2\,\lambda_{\min}(\boldsymbol{\Sigma}_{\mathbf{x}})n}}, &\text{(Lemma B.4)}
\end{aligned}$$

which completes the proof. $\qquad\square$

This completes the bound on the noise term. We proceed to the orthonormalization factor.

**Bounding the orthonormalization factor**

Toward bounding the orthonormalization factor from (8). Defining $\overline{\mathbf{G}}_+$ as the updated representation pre-orthonormalization, we write $\overline{\mathbf{G}}_+ = \mathbf{R}\mathbf{G}_+$, where $\mathbf{G}_+$ is the orthonormalized representation and $\mathbf{R} \in \mathbb{R}^{k \times k}$ is the corresponding orthonormalization factor. Therefore, defining the shorthand $\mathrm{Sym}(\mathbf{A}) = \mathbf{A} + \mathbf{A}^\top$, we have

$$
\begin{aligned}
\mathbf{R}\mathbf{R}^\top &= \mathbf{R}\mathbf{G}_+(\mathbf{R}\mathbf{G}_+)^\top \\
&= \left(\mathbf{G} - \eta_{\mathbf{G}}(\mathbf{F}^\top\mathbf{F})^{-1}\mathbf{F}^\top(\mathbf{F}\mathbf{G} - \mathbf{F}_\star\mathbf{G}_\star) + (\mathbf{F}^\top\mathbf{F})^{-1}\mathbf{F}^\top\mathcal{E}^\top\mathbf{X}(\mathbf{X}^\top\mathbf{X})^{-1}\right)(\cdots)^\top &&\text{(from (16))}\\
&\succ \mathbf{I}_k - \eta_{\mathbf{G}}\mathrm{Sym}\Big(\underbrace{\overline{\mathbf{F}}(\mathbf{F}\mathbf{G} - \mathbf{F}_\star\mathbf{G}_\star)\mathbf{G}^\top}_{\triangleq \Gamma_1}\Big) + \eta_{\mathbf{G}}\mathrm{Sym}\Big(\underbrace{\overline{\mathbf{F}}\mathcal{E}^\top\mathbf{X}(\mathbf{X}^\top\mathbf{X})^{-1}\mathbf{G}^\top}_{\triangleq \Gamma_2}\Big) \\
&\quad - \eta_{\mathbf{G}}^2\mathrm{Sym}\left(\overline{\mathbf{F}}(\mathbf{F}\mathbf{G} - \mathbf{F}_\star\mathbf{G}_\star)\left(\overline{\mathbf{F}}\mathcal{E}^\top\mathbf{X}(\mathbf{X}^\top\mathbf{X})^{-1}\right)^\top\right),
\end{aligned}
$$

where the strictly inequality comes from discarding the positive-definite "diagonal" terms of the expansion. Therefore, by Weyl's inequality for symmetric matrices (Horn & Johnson, 2012), we have:

$$
\lambda_{\min}(\mathbf{R}\mathbf{R}^\top) \geq 1 - 2\eta_{\mathbf{G}}\left(\|\Gamma_1\|_{\mathrm{op}} + \|\Gamma_2\|_{\mathrm{op}} + \eta_{\mathbf{G}}\|\Gamma_1\|_{\mathrm{op}}\|\Gamma_2\|_{\mathrm{op}}\right).
$$

Toward bounding $\|\Gamma_1\|_{\mathrm{op}}$, let the conditions of Lemma B.5 hold. Then,

$$
\begin{aligned}
\|\Gamma_1\|_{\mathrm{op}} &= \left\|\overline{\mathbf{F}}(\mathbf{F}\mathbf{G} - \mathbf{F}_\star\mathbf{G}_\star)\mathbf{G}^\top\right\|_{\mathrm{op}} \\
&= \left\|(\mathbf{F}^\top\mathbf{F})^{-1}\mathbf{F}^\top\mathbf{F}\mathbf{G}\mathbf{G}^\top - (\mathbf{F}^\top\mathbf{F})^{-1}\mathbf{F}^\top\mathbf{F}_\star\mathbf{G}_\star\mathbf{G}^\top\right\|_{\mathrm{op}} \\
&= \left\|\overline{\mathbf{F}}\left(\mathbf{F}_\star\mathbf{G}_\star(\mathbf{I}_{\mathrm{d_x}} - \mathbf{G}^\top\mathbf{G})\mathbf{X}^\top\mathbf{Z}(\mathbf{Z}^\top\mathbf{Z})^{-1} + \mathcal{E}^\top\mathbf{Z}(\mathbf{Z}^\top\mathbf{Z})^{-1}\right)\right\|_{\mathrm{op}} &&\text{(from (19))}\\
&\leq \frac{5}{4}\sigma_{\min}(\mathbf{F})\|\mathbf{F}_\star\|_{\mathrm{op}}\mathrm{dist}(\mathbf{G}, \mathbf{G}_\star)\kappa(\boldsymbol{\Sigma}_{\mathbf{x}}) + \sigma_{\min}(\mathbf{F})\sigma_{\boldsymbol{\varepsilon}}^2\sqrt{\frac{k + \log(1/\delta)}{\lambda_{\min}(\boldsymbol{\Sigma}_{\mathbf{x}})n}} \\
&\leq \underbrace{\frac{5}{2}\kappa(\mathbf{F}_\star)\mathrm{dist}(\mathbf{G}, \mathbf{G}_\star)\kappa(\boldsymbol{\Sigma}_{\mathbf{x}})}_{\triangleq \gamma_1} + \sigma_{\boldsymbol{\varepsilon}}^2\sqrt{\frac{k + \log(1/\delta)}{\sigma_{\min}(\mathbf{F}_\star)^2\,\lambda_{\min}(\boldsymbol{\Sigma}_{\mathbf{x}})n}}. &&\text{(Lemma B.5)}
\end{aligned}
$$

Similarly, letting the conditions of Proposition B.6 hold, we have

$$
\begin{aligned}
\|\Gamma_2\|_{\mathrm{op}} &= \left\|\overline{\mathbf{F}}\mathcal{E}^\top\mathbf{X}(\mathbf{X}^\top\mathbf{X})^{-1}\mathbf{G}^\top\right\|_{\mathrm{op}} \\
&\leq \sigma_{\boldsymbol{\varepsilon}}\sqrt{\frac{\mathrm{d_x} + k + \log(1/\delta)}{\sigma_{\min}(\mathbf{F}_\star)^2\,\lambda_{\min}(\boldsymbol{\Sigma}_{\mathbf{x}})n}} &&\text{(Proposition B.6)}\\
&\triangleq \gamma_2.
\end{aligned}
$$

We observe that $\gamma_2$ will always dominate the second term of the bound on $\|\Gamma_1\|_{\mathrm{op}}$, and therefore:

$$
\lambda_{\min}(\mathbf{R}\mathbf{R}^\top) \geq 1 - 2\eta_{\mathbf{G}}(\gamma_1 + 2\gamma_2 + \eta_{\mathbf{G}}(\gamma_1 + \gamma_2)\gamma_2).
$$

Therefore, we have the following bound on the orthonormalization factor.

**Proposition B.7.** *Let the following conditions hold:*

$$
n \gtrsim \max\left\{\mathrm{d_x} + \log(1/\delta),\; \frac{\sigma_{\boldsymbol{\varepsilon}}^2}{\gamma_2^2}\frac{\mathrm{d_x} + \log(1/\delta)}{\sigma_{\min}(\mathbf{F}_\star)^2\,\lambda_{\min}(\boldsymbol{\Sigma}_{\mathbf{x}})}\right\}
$$

$$
\mathrm{dist}(\mathbf{G}, \mathbf{G}_\star) \leq \frac{2}{5\gamma_1}\kappa(\mathbf{F}_\star)^{-1}\kappa(\boldsymbol{\Sigma}_{\mathbf{x}})^{-1}.
$$

*Then, with probability at least $1 - \delta$, we have the following bound on the orthonormalization factor:*

$$
\sigma_{\min}(\mathbf{R}) \geq \sqrt{1 - 2\eta_{\mathbf{G}}(\gamma_1 + 2\gamma_2 + \eta_{\mathbf{G}}(\gamma_1 + \gamma_2)\gamma_2)}.
$$

The constants $\gamma_1, \gamma_2$ will be instantiated to control the deflation of the contraction factor $1 - \eta_{\mathbf{G}} \implies 1 - c\eta_{\mathbf{G}}$ due to the orthonormalization factor.

**Completing the bound**

We are almost ready to complete the proof. By instantiating the noise bound Proposition B.6 and the orthonormalization factor bound Proposition B.7, we have:

$$\left\|\mathbf{G}_+\mathcal{P}_\star^\perp\right\|_{\mathrm{op}} = \left\|\mathbf{R}^{-1}\left((1-\eta_{\mathbf{G}})\mathbf{G}\mathcal{P}_\star^\perp + \eta_{\mathbf{G}}(\mathbf{F}^\top\mathbf{F})^{-1}\mathbf{F}^\top\mathcal{E}^\top\mathbf{X}(\mathbf{X}^\top\mathbf{X})^{-1}\right)\right\|_{\mathrm{op}}$$

$$\leq \frac{1-\eta_{\mathbf{G}}}{\sigma_{\min}(\mathbf{R})}\left\|\mathbf{G}\mathcal{P}_\star^\perp\right\|_{\mathrm{op}} + \eta_{\mathbf{G}}\left\|(\mathbf{F}^\top\mathbf{F})^{-1}\mathbf{F}^\top\mathcal{E}^\top\mathbf{X}(\mathbf{X}^\top\mathbf{X})^{-1}\right\|_{\mathrm{op}}$$

$$\leq \frac{1-\eta_{\mathbf{G}}}{\sqrt{1-2\eta_{\mathbf{G}}(\gamma_1 + 2\gamma_2 + \eta_{\mathbf{G}}(\gamma_1+\gamma_2)\gamma_2)}}\left\|\mathbf{G}\mathcal{P}_\star^\perp\right\|_{\mathrm{op}} + \eta_{\mathbf{G}}\sigma_{\boldsymbol{\varepsilon}}\sqrt{\frac{d_{\mathsf{x}} + k + \log(1/\delta)}{\sigma_{\min}(\mathbf{F}_\star)^2\,\lambda_{\min}(\mathbf{\Sigma}_{\mathbf{x}})n}}.$$

To understand the effective deflation of the convergence rate, we prove the following numerical helper lemma.

**Lemma B.8.** *Given $c, d \in (0,1)$ and $\varepsilon \in (0, 1/2)$, if $\varepsilon \geq c$, then the following holds:*

$$\frac{1-d}{\sqrt{1-cd}} < 1 - (1-\varepsilon)d.$$

*Additionally, as long as $\varepsilon \leq 1 - \frac{1-\sqrt{1-d}}{d}$, then $1 - (1-\varepsilon)d \leq \sqrt{1-d}$.*

*Proof of Lemma B.8:* squaring both sides of the desired inequality and re-arranging some terms, we arrive at

$$c \leq \frac{1}{d}\left(1 - \frac{(1-d)^2}{(1-(1-\varepsilon)d)^2}\right)$$

$$= \frac{1}{d}\left(1 - \underbrace{\frac{1-d}{1-(1-\varepsilon)d}}_{<1}\right)\underbrace{\left(1 + \frac{1-d}{1-(1-\varepsilon)d}\right)}_{>1}.$$

To certify the above inequality, it suffices to lower-bound the RHS. Since $c, d \in (0,1)$, the last factor is at least 1, such that we have

$$\frac{1}{d}\left(1 - \frac{1-d}{1-(1-\varepsilon)d}\right)\left(1 + \frac{1-d}{1-(1-\varepsilon)d}\right) > \frac{1}{d}\left(1 - \frac{1-d}{1-(1-\varepsilon)d}\right)$$

$$= \frac{1}{d}\frac{(1-\varepsilon)d}{1-(1-\varepsilon)d}$$

$$> \varepsilon.$$

Therefore, $c \leq \varepsilon$ is sufficient for certifying the desired inequality. The latter claim follows by squaring and rearranging terms to yield the quadratic inequality:

$$(1-\varepsilon)^2 d - 2(1-\varepsilon) + 1 \leq 0,$$

Setting $\lambda := 1 - \varepsilon$, the solution interval is $\lambda \in \left(\frac{1-\sqrt{1-d}}{d}, \frac{1+\sqrt{1-d}}{d}\right)$. The upper limit is redundant as it exceeds 1 and $\varepsilon \in (0,1)$, leaving the lower limit as the condition on $\varepsilon$ proposed in the lemma.

Plugging in $\eta_{\mathbf{G}} = d \in (0,1]$ and $2(\gamma_1 + 2\gamma_2 + \eta_{\mathbf{G}}(\gamma_1+\gamma_2)\gamma_2) \leq (\gamma_1 + 2\gamma_2 + (\gamma_1+\gamma_2)\gamma_2 = c$, we try candidate values $\gamma_1 = 1/40$, $\gamma_2 = 1/100$ and set $\varepsilon = c$ to get:

$$\frac{1-\eta_{\mathbf{G}}}{\sqrt{1-2\eta_{\mathbf{G}}(\gamma_1 + 2\gamma_2 + \eta_{\mathbf{G}}(\gamma_1+\gamma_2)\gamma_2)}} < (1 - 0.9\eta_{\mathbf{G}}).$$

Plugging in our candidate values of $\gamma_1, \gamma_2$ into the burn-in conditions of Proposition B.7 finishes the proof of Theorem 3.6.

**Theorem 3.6.** *Consider running* (8) *with* $\lambda_{\mathbf{G}} = 0$, $\eta_{\mathbf{G}} \in [0, 1]$, *and* $\eta_{\mathbf{F}} = 1$. *Define* $\bar{\sigma}^2 \triangleq \frac{\sigma_{\varepsilon}^2}{\sigma_{\min}(\mathbf{F}_{\star})^2 \lambda_{\min}(\Sigma_{\mathbf{x}})}$. *As long as* $\mathrm{dist}(\mathbf{G}, \mathbf{G}_{\star}) \leq 0.01\kappa^{-1}(\Sigma_{\mathbf{x}})\kappa^{-1}(\mathbf{F}_{\star})$ *and* $n \gtrsim \overline{\max}\{1, \sigma^2\}(\mathrm{d_x} + \log(1/\delta))$, *we have:*

$$\mathrm{dist}(\mathbf{G}_+, \mathbf{G}_{\star}) \leq (1 - 0.9\eta_{\mathbf{G}})\mathrm{dist}(\mathbf{G}, \mathbf{G}_{\star}) + \eta_{\mathbf{G}}\Delta,$$

*with probability* $\geq 1 - \delta$, *where* $\Delta \triangleq \mathcal{O}\left(\bar{\sigma}\sqrt{\frac{\mathrm{d_x} + \log(1/\delta)}{n}}\right)$.

## B.3. Multi-Task and Transfer Learning

We first discuss how the ideas in our "single-task" setting directly translate to multi-task learning. For example, taking our proposed algorithm template in (8), an immediate idea is, given the current task heads and shared representation $(\{\mathbf{F}^{(t)}\}, \mathbf{G})$, to form task-specific preconditioners formed locally on each task's batch data:

$$\mathbf{P}_{\mathbf{F}}^{(t)} = \widehat{\mathsf{E}}^{(t)}[\mathbf{z}\mathbf{z}^{\top}], \ \mathbf{P}_{\mathbf{G}}^{(t)} = \mathbf{F}^{(t)\top}\mathbf{F}^{(t)}, \ \mathbf{Q}_{\mathbf{G}}^{(t)} = \widehat{\mathsf{E}}^{(t)}[\mathbf{x}\mathbf{x}^{\top}],$$

and perform a local update on $\mathbf{F}^{(t)}, \mathbf{G}$ before a central agent averages the resulting updated $\mathbf{G}$:

$$\mathbf{F}_+^{(t)} = \mathbf{F}^{(t)} - \eta_{\mathbf{F}}\nabla_{\mathbf{F}}\widehat{\mathcal{L}}(\mathbf{F}^{(t)}, \mathbf{G}^{(t)})\mathbf{Q}_{\mathbf{F}}^{(t)-1}$$
$$\mathbf{G}_+^{(t)} = \mathbf{G} - \eta_{\mathbf{G}}\mathbf{P}_{\mathbf{G}}^{(t)-1}\nabla_{\mathbf{G}}\widehat{\mathcal{L}}^{(t)}(\mathbf{F}_+^{(t)}, \mathbf{G})\mathbf{Q}_{\mathbf{G}}^{(t)-1}, \quad t \in [T]$$
$$\mathbf{G}_+ = \frac{1}{T}\sum_{t=1}^{T}\mathbf{G}_+^{(t)}.$$

However, this presumes $\mathbf{F}^{(t)}$ are invertible, i.e. the task-specific dimension $\mathrm{d_Y} > k$. As opposed to the single-task setting, where as stated in Remark 3.4 we are really viewing $\mathbf{F}$ as the concatentation of of $\mathbf{F}^{(t)}$ to make recovering the representation a well-posed problem, in multi-task settings $\mathrm{d_Y}$ may often be small, e.g. $\mathrm{d_Y} = 1$ (Tripuraneni et al., 2021b; Du et al., 2021; Collins et al., 2021; Thekumparampil et al., 2021). Therefore, (pseudo)-inverting away $(\mathbf{F}^{(t)\top}\mathbf{F}^{(t)})^{\dagger}$ may be highly suboptimal. However, we observe that writing out the representation gradient (9), *as long as we invert away* $\mathbf{Q}_{\mathbf{G}}^{(t)-1}$ *first*, then we have:

$$\mathbf{G}_+^{(t)} = \mathbf{G} - \eta_{\mathbf{G}}\nabla_{\mathbf{G}}\widehat{\mathcal{L}}^{(t)}(\mathbf{F}_+^{(t)}, \mathbf{G})\mathbf{Q}_{\mathbf{G}}^{(t)-1}$$
$$\mathbf{G}_+\mathcal{P}_{\star}^{\perp} = \frac{1}{T}\sum_{t=1}^{T}\mathbf{G}_+^{(t)}\mathcal{P}_{\star}^{\perp}$$
$$= \left(\mathbf{I}_k - \eta_{\mathbf{G}}\frac{1}{T}\sum_{t=1}^{T}\mathbf{F}^{(t)\top}\mathbf{F}^{(t)}\right)\mathbf{G}\mathcal{P}_{\star}^{\perp} + \text{ (task-averaged) noise term.}$$

Since by assumption $\frac{1}{T}\sum_{t=1}^{T}\mathbf{F}^{(t)\top}\mathbf{F}^{(t)}$ is full-rank (otherwise recovering the rank $k$ representation is impossible), then suggestively, we may instead invert away the *task-averaged* preconditioner $\mathbf{P}_{\mathbf{G}} = \frac{1}{T}\sum_{t=1}^{T}\mathbf{F}^{(t)\top}\mathbf{F}^{(t)}$ on the task-averaged $\mathbf{G}^{(t)}$ descent direction before taking a representation step $\mathbf{G}_+$. To summarize, we propose the following two-stage preconditioning:

$$\mathbf{F}_+^{(t)} = \mathbf{F}^{(t)} - \eta_{\mathbf{F}}\nabla_{\mathbf{F}}\widehat{\mathcal{L}}(\mathbf{F}^{(t)}, \mathbf{G}^{(t)})\mathbf{Q}_{\mathbf{F}}^{(t)-1} \tag{20}$$

$$\mathbf{D}^{(t)} = \nabla_{\mathbf{G}}\widehat{\mathcal{L}}^{(t)}(\mathbf{F}_+^{(t)}, \mathbf{G})\mathbf{Q}_{\mathbf{G}}^{(t)-1}, \quad t \in [T] \tag{21}$$

$$\mathbf{G}_+ = \mathbf{G} - \eta_{\mathbf{G}}\mathbf{P}_{\mathbf{G}}^{-1}\left(\frac{1}{T}\sum_{t=1}^{T}\mathbf{D}^{(t)}\right) \tag{22}$$

such that $\mathbf{G}_+\mathcal{P}_{\star}^{\perp} = (1 - \eta_{\mathbf{G}})\mathbf{G}\mathcal{P}_{\star}^{\perp} + \text{ (task-averaged) noise term.} \tag{23}$

The exact same tools used in the proof of Theorem 3.6 apply here, with the requirement of a few additional standard tools to study the "task-averaged" noise term(s). As an example, we refer to Zhang et al. (2024b) for some candidates. However,

we note the qualitative behavior is unchanged. As such, since we are using $n$ data points per each of $T$ tasks to update the gradient, the scaling of the noise term goes from $\mathcal{O}(\sigma_{\boldsymbol{\varepsilon}}\sqrt{d_{\mathsf{x}}/n})$ in our bounds to $\mathcal{O}(\sigma_{\boldsymbol{\varepsilon}}\sqrt{d_{\mathsf{x}}/nT})$.

We remark that in the multi-task setting, where each task may have differing covariances and task-heads $\mathbf{F}_{\star}^{(t)}$, the equivalence of our stylized KFAC variant and the alternating min-min algorithm proposed in Jain et al. (2013); Thekumparampil et al. (2021) breaks down. In particular, the alternating min-min algorithm no longer in general admits $\mathbf{G}$ iterates that can be expressed as a product of matrices as in (5) or (20), and rather can only be stated in vectorized space $\mathrm{vec}(\mathbf{G})$. This means that whereas (20) can be solved as $T$ parallel small matrix multiplication problems, the alternating min-min algorithm nominally requires operating in the vectorized-space $d_{\mathsf{x}}k$.

### Transfer Learning

We first prove the proposed fine-tuning generalization bound.

**Lemma 3.7.** *Let $\widehat{\mathbf{F}}_{\mathrm{ls}}^{(t)} = \mathrm{argmin}_{\widehat{\mathbf{F}}}\ \widehat{\mathsf{E}}^{(t)}[\|\mathbf{y}^{(t)} - \widehat{\mathbf{F}}\mathbf{z}^{(t)}\|_2^2]$, $\mathbf{z}^{(t)} \triangleq \widehat{\mathbf{G}}\mathbf{x}^{(t)}$ be the optimal $\mathbf{F}$ on the batch of $n^{(t)}$ target data (11) given $\widehat{\mathbf{G}}$. Defining $\nu = \mathrm{dist}(\widehat{\mathbf{G}}, \mathbf{G}_{\star})$, given $n^{(t)} \gtrsim k + \log(1/\delta)$, we have with probability $\geq 1 - \delta$:*

$$\mathcal{L}^{(t)}(\widehat{\mathbf{F}}_{\mathrm{ls}}^{(t)}, \widehat{\mathbf{G}}) \triangleq \mathsf{E}\left[\|\mathbf{y}^{(t)} - \widehat{\mathbf{F}}_{\mathrm{ls}}^{(t)}\widehat{\mathbf{G}}\mathbf{x}^{(t)}\|_2^2\right]$$

$$\lesssim \|\mathbf{F}_{\star}^{(t)}\|_F^2\,\lambda_{\max}(\boldsymbol{\Sigma}_{\mathbf{x}}^{(t)})\nu^2 + \frac{\sigma_{\boldsymbol{\varepsilon}}^2(d_{\mathsf{y}}k + \log(1/\delta))}{n^{(t)}}.$$

*Proof of Lemma 3.7.* We observe that we may write:

$$\mathcal{L}^{(t)}(\widehat{\mathbf{F}}_{\mathrm{ls}}^{(t)}, \widehat{\mathbf{G}}) = \mathsf{E}\left[\|\mathbf{y}^{(t)} - \widehat{\mathbf{F}}_{\mathrm{ls}}^{(t)}\widehat{\mathbf{G}}\mathbf{x}^{(t)}\|_2^2\right]$$

$$= \mathsf{E}\left[\|(\widehat{\mathbf{F}}_{\mathrm{ls}}^{(t)}\widehat{\mathbf{G}} - \mathbf{F}_{\star}^{(t)}\mathbf{G}_{\star})\mathbf{x}^{(t)}\|_2^2\right]$$

$$= \left\|(\widehat{\mathbf{F}}_{\mathrm{ls}}^{(t)}\widehat{\mathbf{G}} - \mathbf{F}_{\star}^{(t)}\mathbf{G}_{\star})\boldsymbol{\Sigma}_{\mathbf{x}}^{(t)1/2}\right\|_F^2.$$

Now writing out the definition of $\widehat{\mathbf{F}}_{\mathrm{ls}}^{(t)}$, defining $\mathbf{z}^{(t)} = \widehat{\mathbf{G}}\mathbf{x}^{(t)}$, we have

$$\widehat{\mathbf{F}}_{\mathrm{ls}}^{(t)} = \underset{\widehat{\mathbf{F}}}{\mathrm{argmin}}\ \widehat{\mathsf{E}}^{(t)}[\|\mathbf{y}^{(t)} - \widehat{\mathbf{F}}\widehat{\mathbf{G}}\mathbf{x}^{(t)}\|_2^2]$$

$$= \mathbf{Y}^{(t)\top}\mathbf{Z}^{(t)}(\mathbf{Z}^{(t)\top}\mathbf{Z}^{(t)})^{-1}$$

$$= \mathbf{F}_{\star}^{(t)}\mathbf{G}_{\star}\mathbf{X}^{(t)\top}\mathbf{Z}^{(t)}(\mathbf{Z}^{(t)\top}\mathbf{Z}^{(t)})^{-1} + \mathcal{E}^{\top}\mathbf{Z}^{(t)}(\mathbf{Z}^{(t)\top}\mathbf{Z}^{(t)})^{-1}$$

$$= \mathbf{F}_{\star}^{(t)}\mathbf{G}_{\star}\widehat{\mathbf{G}}^{\top} + \mathbf{F}_{\star}^{(t)}\mathbf{G}_{\star}\mathcal{P}_{\widehat{\mathbf{G}}}^{\perp}\mathbf{X}^{(t)\top}\mathbf{Z}^{(t)}(\mathbf{Z}^{(t)\top}\mathbf{Z}^{(t)})^{-1} + \mathcal{E}^{\top}\mathbf{Z}^{(t)}(\mathbf{Z}^{(t)\top}\mathbf{Z}^{(t)})^{-1},$$

where $\mathcal{P}_{\widehat{\mathbf{G}}}^{\perp} = \mathbf{I}_{d_{\mathsf{x}}} - \widehat{\mathbf{G}}^{\top}\widehat{\mathbf{G}}$ is the projection matrix onto the rowspace of $\widehat{\mathbf{G}}$, using the fact that $\widehat{\mathbf{G}}$ is row-orthonormal (8). Therefore, plugging in the last line into error expression, we have

$$\mathcal{L}^{(t)}(\widehat{\mathbf{F}}_{\mathrm{ls}}^{(t)}, \widehat{\mathbf{G}}) = \left\|(\widehat{\mathbf{F}}_{\mathrm{ls}}^{(t)}\widehat{\mathbf{G}} - \mathbf{F}_{\star}^{(t)}\mathbf{G}_{\star})\boldsymbol{\Sigma}_{\mathbf{x}}^{(t)1/2}\right\|_F^2$$

$$\leq 2\left\|\left(\mathbf{F}_{\star}^{(t)}\mathbf{G}_{\star}\widehat{\mathbf{G}}^{\top}\widehat{\mathbf{G}} + \mathbf{F}_{\star}^{(t)}\mathbf{G}_{\star}\mathcal{P}_{\widehat{\mathbf{G}}}^{\perp}\mathbf{X}^{(t)\top}\mathbf{Z}^{(t)}(\mathbf{Z}^{(t)\top}\mathbf{Z}^{(t)})^{-1}\widehat{\mathbf{G}} - \mathbf{F}_{\star}^{(t)}\mathbf{G}_{\star}\right)\boldsymbol{\Sigma}_{\mathbf{x}}^{(t)1/2}\right\|_F^2$$

$$+ 2\left\|\mathcal{E}^{\top}\mathbf{Z}^{(t)}(\mathbf{Z}^{(t)\top}\mathbf{Z}^{(t)})^{-1}\widehat{\mathbf{G}}\boldsymbol{\Sigma}_{\mathbf{x}}^{(t)1/2}\right\|_F^2. \qquad ((a+b)^2 \leq 2a^2 + 2b^2)$$

Focusing on the first term, we have:

$$\mathbf{F}_{\star}^{(t)}\mathbf{G}_{\star}\widehat{\mathbf{G}}^{\top}\widehat{\mathbf{G}} + \mathbf{F}_{\star}^{(t)}\mathbf{G}_{\star}\mathcal{P}_{\widehat{\mathbf{G}}}^{\perp}\mathbf{X}^{(t)\top}\mathbf{Z}^{(t)}(\mathbf{Z}^{(t)\top}\mathbf{Z}^{(t)})^{-1}\widehat{\mathbf{G}} - \mathbf{F}_{\star}^{(t)}\mathbf{G}_{\star}$$

$$= \mathbf{F}_{\star}^{(t)}\mathbf{G}_{\star}\mathcal{P}_{\widehat{\mathbf{G}}}^{\perp}\left(\mathbf{X}^{(t)\top}\mathbf{Z}^{(t)}(\mathbf{Z}^{(t)\top}\mathbf{Z}^{(t)})^{-1}\widehat{\mathbf{G}} - \mathbf{I}_{d_{\mathsf{x}}}\right).$$

By a covariance concentration argument Lemma E.2, since $\mathbf{X}^{(t)\top}\mathbf{Z}^{(t)}$ and $\mathbf{Z}^{(t)\top}\mathbf{Z}^{(t)}$ are rank-$k$ matrices, as long as $n^{(t)} \gtrsim k + \log(1/\delta)$, we have with probability at least $1 - \delta$:

$$\mathbf{X}^{(t)\top}\mathbf{Z}^{(t)} \approx n^{(t)}\boldsymbol{\Sigma}_{\mathbf{x}}^{(t)}\widehat{\mathbf{G}}^{\top}, \quad (\mathbf{Z}^{(t)\top}\mathbf{Z}^{(t)})^{-1} \approx n^{(t)}\widehat{\mathbf{G}}\boldsymbol{\Sigma}_{\mathbf{x}}^{(t)}\widehat{\mathbf{G}}^{\top},$$

and thus

$$\|\mathbf{F}_\star^{(\mathrm{t})}\mathbf{G}_\star\mathcal{P}_{\widehat{\mathbf{G}}}^\perp\left(\mathbf{X}^{(\mathrm{t})\top}\mathbf{Z}^{(\mathrm{t})}(\mathbf{Z}^{(\mathrm{t})\top}\mathbf{Z}^{(\mathrm{t})})^{-1}\widehat{\mathbf{G}} - \mathbf{I}_{\mathrm{d_x}}\right)\boldsymbol{\Sigma}_\mathbf{x}^{(\mathrm{t})1/2}\|_F$$

$$\approx \|\mathbf{F}_\star^{(\mathrm{t})}\mathbf{G}_\star\mathcal{P}_{\widehat{\mathbf{G}}}^\perp\boldsymbol{\Sigma}_\mathbf{x}^{(\mathrm{t})1/2}\left(\boldsymbol{\Sigma}_\mathbf{x}^{(\mathrm{t})1/2}\widehat{\mathbf{G}}^\top(\widehat{\mathbf{G}}\boldsymbol{\Sigma}_\mathbf{x}^{(\mathrm{t})}\widehat{\mathbf{G}}^\top)^{-1}\widehat{\mathbf{G}}\boldsymbol{\Sigma}_\mathbf{x}^{(\mathrm{t})1/2} - \mathbf{I}_{\mathrm{d_x}}\right)\|_F$$

$$\lesssim \|\mathbf{F}_\star^{(\mathrm{t})}\|_F\left\|\mathbf{G}_\star\mathcal{P}_{\widehat{\mathbf{G}}}^\perp\right\|_{\mathrm{op}}\left\|\boldsymbol{\Sigma}_\mathbf{x}^{(\mathrm{t})1/2}\right\|_{\mathrm{op}}\left\|\boldsymbol{\Sigma}_\mathbf{x}^{(\mathrm{t})1/2}\widehat{\mathbf{G}}^\top(\widehat{\mathbf{G}}\boldsymbol{\Sigma}_\mathbf{x}^{(\mathrm{t})}\widehat{\mathbf{G}}^\top)^{-1}\widehat{\mathbf{G}}\boldsymbol{\Sigma}_\mathbf{x}^{(\mathrm{t})1/2} - \mathbf{I}_{\mathrm{d_x}}\right\|_{\mathrm{op}}$$

$$\leq \|\mathbf{F}_\star^{(\mathrm{t})}\|_F\,\mathrm{dist}(\widehat{\mathbf{G}}, \mathbf{G}_\star)\,\lambda_{\max}(\boldsymbol{\Sigma}_\mathbf{x}^{(\mathrm{t})})^{1/2},$$

where in the last line we applied the definition $\mathrm{dist}(\widehat{\mathbf{G}}, \mathbf{G}_\star) = \left\|\mathbf{G}_\star\mathcal{P}_{\widehat{\mathbf{G}}}^\perp\right\|_{\mathrm{op}} = \left\|\widehat{\mathbf{G}}\mathcal{P}_\star^\perp\right\|_{\mathrm{op}}$, and the fact that the matrix $\mathcal{P} \triangleq \boldsymbol{\Sigma}_\mathbf{x}^{(\mathrm{t})1/2}\widehat{\mathbf{G}}^\top(\widehat{\mathbf{G}}\boldsymbol{\Sigma}_\mathbf{x}^{(\mathrm{t})}\widehat{\mathbf{G}}^\top)^{-1}\widehat{\mathbf{G}}\boldsymbol{\Sigma}_\mathbf{x}^{(\mathrm{t})1/2}$ can be verified to be a projection matrix $\mathcal{P}^2 = \mathcal{P}$, $\mathcal{P}^\top = \mathcal{P}$, such that $\mathcal{P} - \mathbf{I} = \mathcal{P}^\perp$ is also an orthogonal projection and $\left\|\mathcal{P}^\perp\right\|_{\mathrm{op}} = 1$. Now, we analyze the noise term:

$$\mathcal{E}^\top\mathbf{Z}^{(\mathrm{t})}(\mathbf{Z}^{(\mathrm{t})\top}\mathbf{Z}^{(\mathrm{t})})^{-1}\widehat{\mathbf{G}}\boldsymbol{\Sigma}_\mathbf{x}^{(\mathrm{t})1/2} \approx \mathcal{E}^\top\mathbf{Z}^{(\mathrm{t})}(\mathbf{Z}^{(\mathrm{t})\top}\mathbf{Z}^{(\mathrm{t})})^{-1/2}(n^{(\mathrm{t})})^{-1/2}\left(\widehat{\mathbf{G}}\boldsymbol{\Sigma}_\mathbf{x}^{(\mathrm{t})}\widehat{\mathbf{G}}^\top\right)^{-1/2}\widehat{\mathbf{G}}\boldsymbol{\Sigma}_\mathbf{x}^{(\mathrm{t})1/2},$$

where we observed $\mathbf{Z}^{(\mathrm{t})\top}\mathbf{Z}^{(\mathrm{t})} = n\mathbf{G}\widehat{\boldsymbol{\Sigma}}_\mathbf{x}\mathbf{G}^\top$ and applied covariance concentration. Now, defining the (compact) SVD of $\widehat{\mathbf{G}}\boldsymbol{\Sigma}_\mathbf{x}^{(\mathrm{t})1/2} = \mathbf{U}_\mathbf{z}\mathbf{D}_\mathbf{z}\mathbf{V}_\mathbf{z}^\top$, we find

$$\|\mathcal{E}^\top\mathbf{Z}^{(\mathrm{t})}(\mathbf{Z}^{(\mathrm{t})\top}\mathbf{Z}^{(\mathrm{t})})^{-1}\widehat{\mathbf{G}}\boldsymbol{\Sigma}_\mathbf{x}^{(\mathrm{t})1/2}\|_F \lesssim \frac{1}{\sqrt{n^{(\mathrm{t})}}}\|\mathcal{E}^\top\mathbf{Z}^{(\mathrm{t})}(\mathbf{Z}^{(\mathrm{t})\top}\mathbf{Z}^{(\mathrm{t})})^{-1/2}\mathbf{U}_\mathbf{z}\mathbf{V}_\mathbf{z}^\top\|_F$$

$$\lesssim \frac{1}{\sqrt{n^{(\mathrm{t})}}}\|\mathcal{E}^\top\mathbf{Z}^{(\mathrm{t})}(\mathbf{Z}^{(\mathrm{t})\top}\mathbf{Z}^{(\mathrm{t})})^{-1/2}\|_F$$

$$\lesssim \frac{1}{\sqrt{n^{(\mathrm{t})}}}\sigma_{\boldsymbol{\varepsilon}}\sqrt{\mathrm{d_Y}k + \log(1/\delta)},$$

for $n^{(\mathrm{t})} \gtrsim k + \log(1/\delta)$. The last line comes from the Frobenius norm variants of Lemma B.3 and Lemma B.4 (see Ziemann et al. (2023, Theorem 4.1) or Zhang et al. (2024b, Lemma A.3) for details). Putting the two bounds together yields the desired result. $\qquad\square$

## C. Proofs and Additional Details for Section 3.2

### C.1. Proof of Theorem 3.9

**Theorem 3.9.** *Assume that the activation function $\sigma$ is $\mathcal{O}(1)$-Lipschitz and that Assumption 3.8 holds. In the limit where $n, \mathrm{d_x}, \mathrm{d_h}$ tend to infinity proportionally, the matrix $\mathbf{G}_{\mathrm{SGD}}$, with probability $1 - o(1)$, satisfies*

$$\left\|\mathbf{G}_0 + \alpha\eta\,\mathbf{f}_0\boldsymbol{\beta}_{\mathrm{SGD}}^\top - \mathbf{G}_{\mathrm{SGD}}\right\|_{\mathrm{op}} \to 0,$$

*in which $\alpha = \mathsf{E}_z[\sigma'(z)]$ with $z \sim \mathsf{N}(0, \mathrm{d_x}^{-1}\mathrm{Tr}(\boldsymbol{\Sigma}_\mathbf{x}))$, and the vector $\boldsymbol{\beta}_{\mathrm{SGD}}$ is given by $\boldsymbol{\beta}_{\mathrm{SGD}} = n^{-1}\mathbf{X}^\top\boldsymbol{y}$.*

*Proof.* To prove this theorem, we first note that

$$\nabla_\mathbf{G}\widehat{\mathcal{L}}(\mathbf{f}_0, \mathbf{G}_0) = -\frac{1}{n}\sum_{i=1}^n\left(y_i - \frac{1}{\sqrt{\mathrm{d_h}}}\mathbf{f}_0^\top\sigma(\mathbf{G}_0\mathbf{x}_i)\right)\left(\frac{1}{\sqrt{\mathrm{d_h}}}\mathbf{f}_0 \odot \sigma'(\mathbf{G}_0\mathbf{x}_i)\right)\mathbf{x}_i^\top.$$

Adopting the matrix notation $\mathbf{X} = [\mathbf{x}_1|\ldots|\mathbf{x}_n]^\top \in \mathbb{R}^{n\times\mathrm{d_x}}$ and $\boldsymbol{y} = [y_1, \ldots, y_n]^\top \in \mathbb{R}^n$, we can write

$$\nabla_\mathbf{G}\widehat{\mathcal{L}}(\mathbf{f}_0, \mathbf{G}_0) = -\frac{1}{n}\left[\left(\mathrm{d_h}^{-1/2}\mathbf{f}_0\boldsymbol{y}^\top - \mathrm{d_h}^{-1}\mathbf{f}_0\mathbf{f}_0^\top\sigma(\mathbf{G}_0\mathbf{X}^\top)\right) \odot \sigma'(\mathbf{G}_0\mathbf{X}^\top)\right]\mathbf{X}. \tag{24}$$

Let $z \sim \mathsf{N}(0, \mathrm{d_x^{-1}} \operatorname{Tr}(\boldsymbol{\Sigma_x}))$ and define $\alpha = \mathsf{E}_z \left[ \sigma'(z) \right]$, and $\sigma_\perp : \mathbb{R} \to \mathbb{R}$ as $\sigma_\perp(z) = \sigma(z) - \alpha z$. This function satisfies $\mathsf{E}_z \left[ \sigma'_\perp(z) \right] = 0$. With this, we decompose the gradient into three components as $\nabla_{\mathbf{G}} \widehat{\mathcal{L}}(\mathbf{f}_0, \mathbf{G}_0) = \mathbf{T}_1 + \mathbf{T}_2 + \mathbf{T}_3$ with

$$\mathbf{T}_1 = -\alpha \, \mathrm{d_h^{-1/2}} \, \mathbf{f}_0 \left( \frac{\mathbf{X}^\top \boldsymbol{y}}{n} \right)^\top, \quad \mathbf{T}_2 = -n^{-1} \mathrm{d_h^{-1/2}} \left[ \mathbf{f}_0 \, \boldsymbol{y}^\top \odot \sigma'_\perp(\mathbf{G}_0 \mathbf{X}^\top) \right] \mathbf{X},$$

$$\mathbf{T}_3 = n^{-1} \mathrm{d_h^{-1}} \left[ \left( \mathbf{f}_0 \mathbf{f}_0^\top \sigma(\mathbf{G}_0 \mathbf{X}^\top) \right) \odot \sigma'(\mathbf{G}_0 \mathbf{X}^\top) \right] \mathbf{X}.$$

We will analyze each of these components separately.

- **Term 1:** For this term, using the facts that $\|\mathbf{f}_0\|_2 = \mathcal{O}(1)$ and $\|\mathbf{X}^\top \boldsymbol{y}/n\|_2 = \mathcal{O}(1)$, we have

$$\|\mathbf{T}_1\|_{\mathrm{op}} = \mathcal{O}\left( \mathrm{d_h^{-1/2}} \right).$$

- **Term 2:** To analyze this term, note that

$$\mathbf{f}_0 \, \boldsymbol{y}^\top \odot \sigma'_\perp(\mathbf{G}_0 \mathbf{X}^\top) = \operatorname{diag}(\mathbf{f}_0) \, \sigma'_\perp(\mathbf{G}_0 \mathbf{X}^\top) \, \operatorname{diag}(\boldsymbol{y}),$$

which gives

$$\left\| \mathbf{f}_0 \, \boldsymbol{y}^\top \odot \sigma'_\perp(\mathbf{G}_0 \mathbf{X}^\top) \right\|_{\mathrm{op}} \leq \|\mathbf{f}_0\|_\infty \|\boldsymbol{y}\|_\infty \|\sigma'_\perp(\mathbf{G}_0 \mathbf{X}^\top)\|_{\mathrm{op}}.$$

Using basic concentration arguments, we have $\|\mathbf{f}_0\|_\infty = \tilde{\mathcal{O}}(\mathrm{d_h^{-1/2}})$, and $\|\boldsymbol{y}\|_\infty = \tilde{\mathcal{O}}(1)$, with probability $1 - o(1)$. By construction of $\sigma_\perp(\cdot)$, the matrix $\sigma'_\perp(\mathbf{G}_0 \mathbf{X}^\top)$ has mean zero entries, thus using (Vershynin, 2012, Theorem 5.44), we have $\|\sigma'_\perp(\mathbf{G}_0 \mathbf{X}^\top)\|_{\mathrm{op}} = \tilde{O}(\mathrm{d_h^{1/2}} + n^{1/2})$ with probability $1 - o(1)$ Thus, the norm of $\mathbf{T}_2$ can be upper bounded as

$$\|\mathbf{T}_2\|_{\mathrm{op}} = \tilde{\mathcal{O}}\left( \frac{1}{\mathrm{d_h}} \left( 1 + \sqrt{\frac{\mathrm{d_h}}{n}} \right) \left( 1 + \sqrt{\frac{\mathrm{d_x}}{n}} \right) \right) = \tilde{\mathcal{O}}\left( \mathrm{d_h^{-1}} \right).$$

- **Term 3:** Similar to the second term, note that

$$\left( \mathbf{f}_0 \, \mathbf{f}_0^\top \sigma(\mathbf{G}_0 \mathbf{X}^\top) \right) \odot \sigma'(\mathbf{G}_0 \mathbf{X}^\top) = \operatorname{diag}(\mathbf{f}_0) \, \sigma'(\mathbf{G}_0 \mathbf{X}^\top) \, \operatorname{diag}\left( \mathbf{f}_0^\top \sigma(\mathbf{G}_0 \mathbf{X}^\top) \right).$$

Thus, the norm of the third term can be upper bounded as

$$\|\mathbf{T}_3\|_{\mathrm{op}} = \left\| \frac{1}{n \, \mathrm{d_h}} \left[ \left( \mathbf{f}_0 \mathbf{f}_0^\top \sigma(\mathbf{G}_0 \mathbf{X}^\top) \right) \odot \sigma'(\mathbf{G}_0 \mathbf{X}^\top) \right] \mathbf{X} \right\|_{\mathrm{op}} \leq n^{-1} \mathrm{d_h^{-1}} \|\mathbf{X}\|_{\mathrm{op}} \|\mathbf{f}_0\|_\infty \|\mathbf{f}_0^\top \sigma(\mathbf{G}_0 \mathbf{X}^\top)\|_\infty \left\| \sigma'(\mathbf{G}_0 \mathbf{X}^\top) \right\|_{\mathrm{op}}.$$

To analyze the right hand side, note that assuming that $\sigma$ is $\mathcal{O}(1)$-Lipschitz, the entries of $\sigma'(\mathbf{G}_0 \mathbf{X}^\top)$ are bounded by the Lipschitz constant, and we have $\|\sigma'(\mathbf{G}_0 \mathbf{X}^\top)\|_{\mathrm{op}} = \mathcal{O}(\sqrt{n \, \mathrm{d_h}})$. Also, using a simple orderwise analysis we have $\|\mathbf{f}_0^\top \sigma(\mathbf{G}_0 \mathbf{X}^\top)\|_\infty = \tilde{\mathcal{O}}(1)$, which gives

$$\|\mathbf{T}_3\|_{\mathrm{op}} = \tilde{\mathcal{O}}\left( \frac{1}{\mathrm{d_h}} \left( 1 + \sqrt{\frac{\mathrm{d_x}}{n}} \right) \right) = \tilde{\mathcal{O}}(\mathrm{d_h^{-1}}).$$

To wrap up, note that $\mathrm{d_h^{1/2}} \|\mathbf{T}_1\|_{\mathrm{op}} = \mathcal{O}(1)$, whereas $\mathrm{d_h^{1/2}} \|\mathbf{T}_2\|_{\mathrm{op}}$ and $\mathrm{d_h^{1/2}} \|\mathbf{T}_3\|_{\mathrm{op}} = o(1)$. Thus, with probability $1 - o(1)$ we have

$$\mathbf{G}_{\mathrm{SGD}} = \mathbf{G}_0 + \eta \, \mathrm{d_h^{1/2}} \, \nabla_{\mathbf{G}} \widehat{\mathcal{L}}(\mathbf{f}_0, \mathbf{G}_0) = \mathbf{G}_0 + \alpha \eta \, \mathbf{f}_0 \left( n^{-1} \mathbf{X}^\top \boldsymbol{y} \right)^\top + \boldsymbol{\Delta}$$

with $\|\boldsymbol{\Delta}\|_{\mathrm{op}} = o(1)$, finishing the proof. $\qquad \square$

## C.2. Proof of Lemma 3.10

**Lemma 3.10.** *Under the assumptions of Theorem 3.9, the correlation between $\boldsymbol{\beta}_\star$ and $\boldsymbol{\beta}_{\text{SGD}}$ satisfies*

$$\left| \frac{\boldsymbol{\beta}_\star^\top \boldsymbol{\beta}_{\text{SGD}}}{\|\boldsymbol{\beta}_{\text{SGD}}\|_2 \|\boldsymbol{\beta}_\star\|_2} - \frac{\frac{c_{\star,1}}{d_{\text{x}}} \operatorname{Tr}(\boldsymbol{\Sigma}_{\mathbf{x}})}{\sqrt{\frac{c_\star^2 + \sigma_\varepsilon^2}{n} \operatorname{Tr}(\boldsymbol{\Sigma}_{\mathbf{x}}) + \frac{c_{\star,1}^2}{d_{\text{x}}} \operatorname{Tr}(\boldsymbol{\Sigma}_{\mathbf{x}}^2)}} \right| \to 0$$

*with probability $1 - o(1)$, in which $c_{\star,1} = \mathsf{E}_z[\sigma_\star'(z)]$ and $c_\star^2 = \mathsf{E}_z[\sigma_\star^2(z)]$ with $z \sim \mathsf{N}(0, d_{\text{x}}^{-1} \operatorname{Tr}(\boldsymbol{\Sigma}_{\mathbf{x}}))$.*

*Proof.* Recall that $\boldsymbol{\beta}_{\text{SGD}} = \frac{1}{n} \mathbf{X}^\top \boldsymbol{y}$ and $\boldsymbol{y} = \sigma_\star(\mathbf{X}\boldsymbol{\beta}_\star) + \boldsymbol{\varepsilon}$ where $\boldsymbol{\varepsilon} = [\varepsilon_1, \dots, \varepsilon_n]^\top$. Therefore, with probability $1 - o(1)$ we have

$$\boldsymbol{\beta}_{\text{SGD}}^\top \boldsymbol{\beta}_\star = \frac{1}{n}(\mathbf{X}\boldsymbol{\beta}_\star)^\top \boldsymbol{y} = \frac{1}{n}(\mathbf{X}\boldsymbol{\beta}_\star)^\top \sigma_\star(\mathbf{X}\boldsymbol{\beta}_\star) + o(1)$$

where we have used the fact that $\boldsymbol{\varepsilon}$ is mean zero. Thus, using the weak law of large numbers,

$$\boldsymbol{\beta}_{\text{SGD}}^\top \boldsymbol{\beta}_\star \to \mathsf{E}_z\left[z\sigma_\star(z)\right] = d_{\text{x}}^{-1} \operatorname{Tr}(\boldsymbol{\Sigma}_{\mathbf{x}}) \mathsf{E}_z\left[\sigma_\star'(z)\right] = c_{\star,1} d_{\text{x}}^{-1} \operatorname{Tr}(\boldsymbol{\Sigma}_{\mathbf{x}}) \tag{25}$$

in probability, where $z \sim \mathsf{N}(0, d_{\text{x}}^{-1} \operatorname{Tr}(\boldsymbol{\Sigma}_{\mathbf{x}}))$. Similarly, $\|\boldsymbol{\beta}_{\text{SGD}}\|_2^2$ can be written as

$$\|\boldsymbol{\beta}_{\text{SGD}}\|_2^2 = n^{-2} \boldsymbol{y}^\top \mathbf{X}\mathbf{X}^\top \boldsymbol{y} = n^{-2}\boldsymbol{\varepsilon}^\top \mathbf{X}\mathbf{X}^\top \boldsymbol{\varepsilon} + n^{-2}\sigma_\star(\mathbf{X}\boldsymbol{\beta}_\star)^\top \mathbf{X}\mathbf{X}^\top \sigma_\star(\mathbf{X}\boldsymbol{\beta}_\star) + o(1).$$

We will analyze each of the two remaining term separately. For the first term, recall that $\boldsymbol{\varepsilon}$ is independent of $\mathbf{X}$. Using the Hanson-Wright inequality (Theorem E.6) we have

$$n^{-2}\boldsymbol{\varepsilon}^\top \mathbf{X}\mathbf{X}^\top \boldsymbol{\varepsilon} = \sigma_\varepsilon^2 n^{-1} \operatorname{Tr}(\mathbf{X}\mathbf{X}^\top/n) + o(1) = \sigma_\varepsilon^2 n^{-1} \operatorname{Tr}(\boldsymbol{\Sigma}_{\mathbf{x}}) + o(1).$$

For the second term, note that $\mathbf{X}\boldsymbol{\beta}_\star$ is a vector with i.i.d. elements $\mathbf{x}_i^\top \boldsymbol{\beta}_\star$, each of them distributed according to $\mathsf{N}(0, \boldsymbol{\beta}_\star^\top \boldsymbol{\Sigma}_{\mathbf{x}} \boldsymbol{\beta}_\star)$. Let $z$ be a random variable distributed as $z \sim \mathsf{N}(0, \boldsymbol{\beta}_\star^\top \boldsymbol{\Sigma}_{\mathbf{x}} \boldsymbol{\beta}_\star)$. We decompose the function $\sigma_\star$ into a linear and a nonlinear part as

$$\sigma_\star(z) = c_{\star,1} z + \sigma_{\star,\perp}(z). \tag{26}$$

This decomposition satisfies

$$\mathsf{E}_z[\sigma_{\star,\perp}(z)] = \mathsf{E}_z[\sigma_\star(z)] = 0$$
$$\mathsf{E}_z[z\,\sigma_{\star,\perp}(z)] = \mathsf{E}_z\, z\,\sigma_\star(z) - c_{\star,1} \mathsf{E}_z[z^2] = \mathsf{E}_z[z\,\sigma_\star(z)] - c_{\star,1} \boldsymbol{\beta}_\star^\top \boldsymbol{\Sigma}_{\mathbf{x}} \boldsymbol{\beta}_\star = 0,$$

where the last equality is due to Stein's lemma (Lemma E.7). This shows that the random variables $z$ and $\sigma_{\star,\perp}(z)$ are uncorrelated. With this, we have

$$n^{-2}\sigma_\star(\mathbf{X}\boldsymbol{\beta}_\star)^\top \mathbf{X}\mathbf{X}^\top \sigma_\star(\mathbf{X}\boldsymbol{\beta}_\star) = n^{-2}(c_{\star,1}\mathbf{X}\boldsymbol{\beta}_\star + \sigma_{\star,\perp}(\mathbf{X}\boldsymbol{\beta}_\star))^\top \mathbf{X}\mathbf{X}^\top (c_{\star,1}\mathbf{X}\boldsymbol{\beta}_\star + \sigma_{\star,\perp}(\mathbf{X}\boldsymbol{\beta}_\star))$$

$$= c_{\star,1}^2 n^{-2}\boldsymbol{\beta}_\star^\top (\mathbf{X}^\top \mathbf{X})^2 \boldsymbol{\beta}_\star + 2c_{\star,1}n^{-2}(\mathbf{X}\boldsymbol{\beta}_\star)^\top \mathbf{X}\mathbf{X}^\top \sigma_{\star,\perp}(\mathbf{X}\boldsymbol{\beta}_\star) + n^{-2}\sigma_{\star,\perp}(\mathbf{X}\boldsymbol{\beta}_\star)^\top \mathbf{X}\mathbf{X}^\top \sigma_{\star,\perp}(\mathbf{X}\boldsymbol{\beta}_\star). \tag{27}$$

For the first term in this sum, by assumption 3.8 and the Hanson-Wright inequality (Theorem E.6), we can write

$$c_{\star,1}^2 n^{-2}\boldsymbol{\beta}_\star^\top (\mathbf{X}^\top \mathbf{X})^2 \boldsymbol{\beta}_\star = c_{\star,1}^2\, d_{\text{x}}^{-1} \operatorname{Tr}(n^{-2}(\mathbf{X}^\top \mathbf{X})^2) + o(1) = c_{\star,1}^2 d_{\text{x}}^{-1} \operatorname{Tr}(\boldsymbol{\Sigma}_{\mathbf{x}}^2) + c_{\star,1}^2 n^{-1} d_{\text{x}}^{-1} \operatorname{Tr}^2(\boldsymbol{\Sigma}_{\mathbf{x}}),$$

where in the last we plugged in the second Wishart moment. For the second term in (27), although by construction $\mathbf{X}\boldsymbol{\beta}_\star$ and $\sigma_{\star,\perp}(\mathbf{X}\boldsymbol{\beta}_\star)$ are uncorrelated, the vector $\mathbf{X}\boldsymbol{\beta}_\star$ and the matrix $\mathbf{X}\mathbf{X}^\top$ are dependent, which complicates the analysis. To resolve this issue, we define $\tilde{\mathbf{X}} = \mathbf{X} - \mathbf{X}\boldsymbol{\beta}_\star \boldsymbol{\beta}_\star^\top$ which satisfies $\mathbf{X}\boldsymbol{\beta}_\star \perp\!\!\!\perp \tilde{\mathbf{X}}$ and write

$$\mathbf{X}\mathbf{X}^\top = \tilde{\mathbf{X}}\tilde{\mathbf{X}}^\top + \tilde{\mathbf{X}}\boldsymbol{\beta}_\star \boldsymbol{\beta}_\star^\top \mathbf{X}^\top + \mathbf{X}\boldsymbol{\beta}_\star \boldsymbol{\beta}_\star^\top \tilde{\mathbf{X}}^\top + \mathbf{X}\boldsymbol{\beta}_\star \boldsymbol{\beta}_\star^\top \mathbf{X}^\top.$$

Thus, the second term in (27) can be written as

$$
n^{-2}(\mathbf{X}\boldsymbol{\beta}_\star)^\top \mathbf{X}\mathbf{X}^\top \sigma_{\star,\perp}(\mathbf{X}\boldsymbol{\beta}_\star) = n^{-2}(\mathbf{X}\boldsymbol{\beta}_\star)^\top \left[ \tilde{\mathbf{X}}\tilde{\mathbf{X}}^\top + \tilde{\mathbf{X}}\boldsymbol{\beta}_\star \boldsymbol{\beta}_\star^\top \mathbf{X}^\top + \mathbf{X}\boldsymbol{\beta}_\star \boldsymbol{\beta}_\star^\top \tilde{\mathbf{X}}^\top + \mathbf{X}\boldsymbol{\beta}_\star \boldsymbol{\beta}_\star^\top \mathbf{X}^\top \right] \sigma_{\star,\perp}(\mathbf{X}\boldsymbol{\beta}_\star)
$$

$$
= \underbrace{n^{-2}(\mathbf{X}\boldsymbol{\beta}_\star)^\top \tilde{\mathbf{X}}\tilde{\mathbf{X}}^\top \sigma_{\star,\perp}(\mathbf{X}\boldsymbol{\beta}_\star)}_{T_1} + \underbrace{n^{-2}(\mathbf{X}\boldsymbol{\beta}_\star)^\top \left[ \tilde{\mathbf{X}}\boldsymbol{\beta}_\star \boldsymbol{\beta}_\star^\top \mathbf{X}^\top + \mathbf{X}\boldsymbol{\beta}_\star \boldsymbol{\beta}_\star^\top \tilde{\mathbf{X}}^\top + \mathbf{X}\boldsymbol{\beta}_\star \boldsymbol{\beta}_\star^\top \mathbf{X}^\top \right] \sigma_{\star,\perp}(\mathbf{X}\boldsymbol{\beta}_\star)}_{T_2}.
$$

The term $T_1$ can be shown to be $o(1)$ by using the Hanson-Wright inequality (Theorem E.6) and noting that $\tilde{\mathbf{X}}\tilde{\mathbf{X}}^\top$ is independent of $\mathbf{X}\boldsymbol{\beta}_\star$, and also the fact that $\mathbf{X}\boldsymbol{\beta}_\star$ and $\sigma_{\star,\perp}(\mathbf{X}\boldsymbol{\beta}_\star)$ are orthogonal, by construction. Similarly, $T_2$ can also be shown to be $o(1)$ using a similar argument. For this, we also use that fact that by construction we have $\tilde{\mathbf{X}}\boldsymbol{\beta}_\star \perp\!\!\!\perp \mathbf{X}\boldsymbol{\beta}_\star$ which alongside $\mathsf{E}\left[\tilde{\mathbf{X}}\boldsymbol{\beta}_\star\right] = \mathbf{0}_n$ proves that $n^{-1}(\mathbf{X}\boldsymbol{\beta}_\star)^\top \tilde{\mathbf{X}}\boldsymbol{\beta}_\star = o(1)$ and $n^{-1}(\tilde{\mathbf{X}}\boldsymbol{\beta}_\star)^\top \sigma_{\star,\perp}(\tilde{\mathbf{X}}\boldsymbol{\beta}_\star) = o(1)$. Hence,

$$
2c_{\star,1}n^{-2}(\mathbf{X}\boldsymbol{\beta}_\star)^\top \mathbf{X}\mathbf{X}^\top \sigma_{\star,\perp}(\mathbf{X}\boldsymbol{\beta}_\star) \to 0.
$$

For the third term in (27), we can use a similar argument and replace $\mathbf{X}$ with $\tilde{\mathbf{X}}$ to show that

$$
n^{-2}\sigma_{\star,\perp}(\mathbf{X}\boldsymbol{\beta}_\star)^\top \mathbf{X}\mathbf{X}^\top \sigma_{\star,\perp}(\mathbf{X}\boldsymbol{\beta}_\star) \to \mathsf{E}_z[\sigma_{\star,\perp}(z)]^2 \, n^{-1} \operatorname{Tr}(\boldsymbol{\Sigma}_\mathbf{x}).
$$

Putting everything together, we have

$$
\|\boldsymbol{\beta}_{\text{SGD}}\|_2^2 = c_{\star,1}^2 \mathrm{d}_\mathsf{x}^{-1} \operatorname{Tr}(\boldsymbol{\Sigma}_\mathbf{x}^2) + \sigma_\varepsilon^2 \, n^{-1} \operatorname{Tr}(\boldsymbol{\Sigma}_\mathbf{x}) + \mathsf{E}_z[\sigma_{\star,\perp}(z)]^2 \, n^{-1} \operatorname{Tr}(\boldsymbol{\Sigma}_\mathbf{x}) + c_{\star,1}^2 n^{-1}\mathrm{d}_\mathsf{x}^{-1} \operatorname{Tr}^2(\boldsymbol{\Sigma}_\mathbf{x}) + o(1)
$$

$$
= c_{\star,1}^2 \mathrm{d}_\mathsf{x}^{-1} \operatorname{Tr}(\boldsymbol{\Sigma}_\mathbf{x}^2) + n^{-1} \operatorname{Tr}(\boldsymbol{\Sigma}_\mathbf{x}) \left( \sigma_\varepsilon^2 + \mathsf{E}_z[\sigma_{\star,\perp}(z)]^2 + c_{\star,1}^2 \mathrm{d}_\mathsf{x}^{-1} \operatorname{Tr}(\boldsymbol{\Sigma}_\mathbf{x}) \right) + o(1).
$$

Note that given the decomposition (26), we have

$$
\mathsf{E}_z\left[\sigma_\star^2(z)\right] = \mathsf{E}_z\left[\sigma_{\star,\perp}^2(z)\right] + c_{\star,1}^2 \mathrm{d}_\mathsf{x}^{-1} \operatorname{Tr}(\boldsymbol{\Sigma}_\mathbf{x})
$$

given the orthogonality of the linear and nonlinear terms. Hence,

$$
\|\boldsymbol{\beta}_{\text{SGD}}\|_2^2 = c_{\star,1}^2 \mathrm{d}_\mathsf{x}^{-1} \operatorname{Tr}(\boldsymbol{\Sigma}_\mathbf{x}^2) + n^{-1} \operatorname{Tr}(\boldsymbol{\Sigma}_\mathbf{x}) \left( \sigma_\varepsilon^2 + c_\star^2 \right) + o(1),
$$

which alongside (25) proves the lemma. $\qquad\square$

### C.3. Proof of Lemma 3.12

**Lemma 3.12.** *Under the assumptions of Theorem 3.9, the correlation between $\boldsymbol{\beta}_\star$ and $\boldsymbol{\beta}_{\text{KFAC}}$ satisfies*

$$
\left| \frac{\boldsymbol{\beta}_{\text{KFAC}}^\top \boldsymbol{\beta}_\star}{\|\boldsymbol{\beta}_{\text{KFAC}}\|_2 \|\boldsymbol{\beta}_\star\|_2} - \frac{c_{\star,1}\Psi_1}{\sqrt{c_{\star,1}^2 \Psi_2 + \frac{\mathrm{d}_\mathsf{x}}{n}(c_{\star,>1}^2 + \sigma_\varepsilon^2)\Psi_3}} \right| \to 0
$$

*with probability $1 - o(1)$, where $c_{\star,1}^2 = \mathsf{E}_z^2[\sigma_\star'(z)]$, $c_{\star,>1}^2 = \mathsf{E}_z[\sigma_{\star,\perp}^2(z)]$ with $z \sim \mathsf{N}(0, \mathrm{d}_\mathsf{x}^{-1} \operatorname{Tr}(\boldsymbol{\Sigma}_\mathbf{x}))$, and $\Psi_1, \Psi_2, \Psi_3$ are defined in (34) and depend on $\boldsymbol{\Sigma}_\mathbf{x}$, $\mathrm{d}_\mathsf{x}/n$, and $\lambda_\mathbf{G}$. In particular, as $\lambda_\mathbf{G} \to 0$ and $\mathrm{d}_\mathsf{x}/n \to 0$, we have*

$$
\boldsymbol{\beta}_{\text{KFAC}}^\top \boldsymbol{\beta}_\star / (\|\boldsymbol{\beta}_{\text{KFAC}}\|_2 \|\boldsymbol{\beta}_\star\|_2) \to 1.
$$

*Proof.* Recall that $\mathbf{Q}_\mathbf{G} = n^{-1}\mathbf{X}^\top \mathbf{X}$ and let $\mathbf{R} = (\mathbf{Q}_\mathbf{G} + \lambda_\mathbf{G}\mathbf{I}_{\mathrm{d}_\mathsf{x}})^{-1}$. The inner product of $\boldsymbol{\beta}_\star$ and $\boldsymbol{\beta}_{\text{KFAC}}$ is given by

$$
\boldsymbol{\beta}_\star^\top \boldsymbol{\beta}_{\text{KFAC}} = n^{-1}\boldsymbol{\beta}_\star^\top \mathbf{R}\mathbf{X}^\top \sigma_\star(\mathbf{X}\boldsymbol{\beta}_\star) + o(1),
$$

where we have used the fact that $\boldsymbol{\varepsilon}$ is mean zero and is independent of all other randomness in the problem. Defining $\bar{\mathbf{R}} = \left(\mathbf{X}\mathbf{X}^\top/n + \lambda_\mathbf{G}\mathbf{I}_n\right)^{-1}$, we can use the push-through identity to rewrite the inner product as

$$
\boldsymbol{\beta}_\star^\top \boldsymbol{\beta}_{\text{KFAC}} = n^{-1}(\mathbf{X}\boldsymbol{\beta}_\star)^\top \bar{\mathbf{R}} \, \sigma_\star(\mathbf{X}\boldsymbol{\beta}_\star) + o(1).
$$

Note that $\mathbf{X}\boldsymbol{\beta}_\star$ is a vector with i.i.d. elements $\mathbf{x}_i^\top \boldsymbol{\beta}_\star$, each of them distributed according to $\mathsf{N}(0, \boldsymbol{\beta}_\star^\top \boldsymbol{\Sigma}_\mathbf{x} \boldsymbol{\beta}_\star)$. Using the same decomposition for $\sigma_\star$ as the one used in the proof of Lemma 3.10 in (26), we have

$$\boldsymbol{\beta}_\star^\top \boldsymbol{\beta}_{\mathrm{KFAC}} = \frac{1}{n}(\mathbf{X}\boldsymbol{\beta}_\star)^\top \bar{\mathbf{R}}\left(c_{\star,1}\mathbf{X}\boldsymbol{\beta}_\star + \sigma_{\star,\perp}(\mathbf{X}\boldsymbol{\beta}_\star)\right) + o(1)$$

$$= c_{\star,1}\mathrm{d}_\mathbf{x}^{-1}\mathrm{Tr}\left(\mathbf{X}^\top(\mathbf{X}\mathbf{X}^\top + \lambda_\mathbf{G} n\mathbf{I}_n)^{-1}\mathbf{X}\right) + n^{-1}(\mathbf{X}\boldsymbol{\beta}_\star)^\top \bar{\mathbf{R}}\,\sigma_{\star,\perp}(\mathbf{X}\boldsymbol{\beta}_\star) + o(1), \qquad (28)$$

where for the first term we have used Assumption 3.8 and the Hanson-Wright inequality (Theorem E.6). To analyze the second term, note that although by construction $\mathbf{X}\boldsymbol{\beta}_\star$ and $\sigma_{\star,\perp}(\mathbf{X}\boldsymbol{\beta}_\star)$ are uncorrelated, the vectors $\mathbf{X}\boldsymbol{\beta}_\star$ and $\bar{\mathbf{R}}$ are dependent, which complicates the analysis. To resolve this issue, we use the same trick used in the proof of Lemma 3.10 and set $\tilde{\mathbf{X}} = \mathbf{X} - \mathbf{X}\boldsymbol{\beta}_\star\boldsymbol{\beta}_\star^\top$ which satisfies $\mathbf{X}\boldsymbol{\beta}_\star \perp\!\!\!\perp \tilde{\mathbf{X}}$, and use it to write

$$\bar{\mathbf{R}}^{-1} = n^{-1}\mathbf{X}\mathbf{X}^\top + \lambda_\mathbf{G}\mathbf{I}_n = n^{-1}\left(\tilde{\mathbf{X}} + \mathbf{X}\boldsymbol{\beta}_\star\boldsymbol{\beta}_\star^\top\right)\left(\tilde{\mathbf{X}} + \mathbf{X}\boldsymbol{\beta}_\star\boldsymbol{\beta}_\star^\top\right)^\top + \lambda_\mathbf{G}\mathbf{I}_n$$

$$= \left(n^{-1}\tilde{\mathbf{X}}\tilde{\mathbf{X}}^\top + \lambda_\mathbf{G}\mathbf{I}_n\right) + n^{-1}(\mathbf{X}\boldsymbol{\beta}_\star)(\mathbf{X}\boldsymbol{\beta}_\star)^\top + n^{-1}(\tilde{\mathbf{X}}\boldsymbol{\beta}_\star)(\mathbf{X}\boldsymbol{\beta}_\star)^\top + n^{-1}(\mathbf{X}\boldsymbol{\beta}_\star)(\tilde{\mathbf{X}}\boldsymbol{\beta}_\star)^\top.$$

Defining $\tilde{\mathbf{R}} = \left(\tilde{\mathbf{X}}\tilde{\mathbf{X}}^\top/n + \lambda_\mathbf{G}\mathbf{I}_n\right)^{-1} \in \mathbb{R}^{n\times n}$, $\mathbf{V} = [\,n^{-1/2}\mathbf{X}\boldsymbol{\beta}_\star \,|\, n^{-1/2}\tilde{\mathbf{X}}\boldsymbol{\beta}_\star\,] \in \mathbb{R}^{n\times 2}$, and

$$\mathbf{D} = \begin{bmatrix} 1 & 1 \\ 1 & 0 \end{bmatrix},$$

we have $\bar{\mathbf{R}} = \tilde{\mathbf{R}} + \mathbf{V}\mathbf{D}\mathbf{V}^\top$. Using Woodbury matrix identity (Theorem E.8), $\bar{\mathbf{R}}$ is given by

$$\bar{\mathbf{R}} = \tilde{\mathbf{R}} - \tilde{\mathbf{R}}\,\mathbf{V}(\mathbf{D}^{-1} + \mathbf{V}^\top\tilde{\mathbf{R}}\mathbf{V})^{-1}\mathbf{V}^\top\tilde{\mathbf{R}} \qquad (29)$$

and plugging this expression into the second term in (28) gives

$$n^{-1}(\mathbf{X}\boldsymbol{\beta}_\star)^\top \bar{\mathbf{R}}\,\sigma_{\star,\perp}(\mathbf{X}\boldsymbol{\beta}_\star) = n^{-1}(\mathbf{X}\boldsymbol{\beta}_\star)^\top\left(\tilde{\mathbf{R}} - \tilde{\mathbf{R}}\,\mathbf{V}(\mathbf{D}^{-1} + \mathbf{V}^\top\tilde{\mathbf{R}}\mathbf{V})^{-1}\mathbf{V}^\top\tilde{\mathbf{R}}\right)\sigma_{\star,\perp}(\mathbf{X}\boldsymbol{\beta}_\star) + o(1)$$

$$= n^{-1}(\mathbf{X}\boldsymbol{\beta}_\star)^\top\tilde{\mathbf{R}}\,\sigma_{\star,\perp}(\mathbf{X}\boldsymbol{\beta}_\star) - n^{-1}(\mathbf{X}\boldsymbol{\beta}_\star)^\top\tilde{\mathbf{R}}\left(\mathbf{V}(\mathbf{D}^{-1} + \mathbf{V}^\top\tilde{\mathbf{R}}\mathbf{V})^{-1}\mathbf{V}^\top\right)\tilde{\mathbf{R}}\,\sigma_{\star,\perp}(\mathbf{X}\boldsymbol{\beta}_\star) + o(1).$$

The first term can be shown to be $o(1)$ in probability by using the fact that $\tilde{\mathbf{R}}$ is independent of $\mathbf{X}\boldsymbol{\beta}_\star$ and the orthogonality of $\mathbf{X}\boldsymbol{\beta}_\star$ and $\sigma_{\star,\perp}(\mathbf{X}\boldsymbol{\beta}_\star)$. To analyze the second term, first note that the elements of the matrix $\boldsymbol{\Omega} = (\mathbf{D}^{-1} + \mathbf{V}^\top\tilde{\mathbf{R}}\mathbf{V})^{-1} \in \mathbb{R}^{2\times 2}$ can all be shown to be $\mathcal{O}(1)$ by a simple norm argument. Moreover,

$$n^{-1}(\mathbf{X}\boldsymbol{\beta}_\star)^\top\tilde{\mathbf{R}}\mathbf{V}(\mathbf{D}^{-1} + \mathbf{V}^\top\tilde{\mathbf{R}}\mathbf{V})^{-1}\mathbf{V}^\top\tilde{\mathbf{R}}\,\sigma_{\star,\perp}(\mathbf{X}\boldsymbol{\beta}_\star)$$

$$= n^{-2}(\mathbf{X}\boldsymbol{\beta}_\star)^\top\tilde{\mathbf{R}}\left(\Omega_{11}(\mathbf{X}\boldsymbol{\beta}_\star)(\mathbf{X}\boldsymbol{\beta}_\star)^\top + \Omega_{12}(\mathbf{X}\boldsymbol{\beta}_\star)(\tilde{\mathbf{X}}\boldsymbol{\beta}_\star)^\top + \Omega_{21}(\tilde{\mathbf{X}}\boldsymbol{\beta}_\star)(\mathbf{X}\boldsymbol{\beta}_\star)^\top + \Omega_{22}(\tilde{\mathbf{X}}\boldsymbol{\beta}_\star)(\tilde{\mathbf{X}}\boldsymbol{\beta}_\star)^\top\right)\tilde{\mathbf{R}}\,\sigma_{\star,\perp}(\mathbf{X}\boldsymbol{\beta}_\star)$$

where $\Omega_{ij}$ are the elements of the matrix $\boldsymbol{\Omega}$. We analyze each term in this sum separately and show that all of them are $o(1)$.

- **First Term.** Using a simple norm argument, $n^{-1}(\mathbf{X}\boldsymbol{\beta}_\star)\tilde{\mathbf{R}}(\mathbf{X}\boldsymbol{\beta}_\star) = \mathcal{O}(1)$. Also, by construction of $\sigma_{\star,\perp}$, we have

$$n^{-1}(\mathbf{X}\boldsymbol{\beta}_\star)^\top\tilde{\mathbf{R}}\sigma_{\star,\perp}(\mathbf{X}\boldsymbol{\beta}_\star) \to 0.$$

  Thus, the whole term is $o(1)$.

- **Second Term.** Similar to the first term, we have $n^{-1}(\mathbf{X}\boldsymbol{\beta}_\star)\tilde{\mathbf{R}}(\mathbf{X}\boldsymbol{\beta}_\star) = \mathcal{O}(1)$. Also, $n^{-1}(\tilde{\mathbf{X}}\boldsymbol{\beta}_\star)^\top\tilde{\mathbf{R}}\,\sigma_{\star,\perp}(\mathbf{X}\boldsymbol{\beta}_\star) \to 0$ in probability, using the weak law of large numbers by noting that $\sigma_{\star,\perp}(\mathbf{X}\boldsymbol{\beta}_\star)$ is independent of $n^{-1}(\tilde{\mathbf{X}}\boldsymbol{\beta}_\star)^\top\tilde{\mathbf{R}}$ by construction and that it has mean zero. Hence, the whole term is $o(1)$.

- **Third Term.** By construction, the vector $\mathbf{X}\boldsymbol{\beta}_\star$ is independent of $\tilde{\mathbf{R}}(\tilde{\mathbf{X}}\boldsymbol{\beta}_\star)$ and has mean zero, which gives $n^{-1}(\mathbf{X}\boldsymbol{\beta}_\star)^\top\tilde{\mathbf{R}}(\tilde{\mathbf{X}}\boldsymbol{\beta}_\star) \to 0$. Also, using a simple norm argument, we have $n^{-1}(\mathbf{X}\boldsymbol{\beta}_\star)^\top\tilde{\mathbf{R}}\sigma_{\star,\perp}(\mathbf{X}\boldsymbol{\beta}_\star) = \mathcal{O}(1)$ which proves that the third term is also $o(1)$.

- **Fourth Term.** This term can be shown to be $o(1)$ with an argument very similar to the argument for the third term.

Putting these all together and using (28), we have

$$\boldsymbol{\beta}_\star^\top \boldsymbol{\beta}_{\mathrm{KFAC}} = c_{\star,1} \mathrm{d}_{\mathrm{x}}^{-1} \mathrm{Tr}\left((\mathbf{X}^\top \mathbf{X}/n)\,\mathbf{R}\right) + o(1). \tag{30}$$

Next we move to the analysis of the squared $\ell_2$-norm of the vector $\boldsymbol{\beta}_{\mathrm{KFAC}}$. By decomposing the function $\sigma_\star$ into a linear and an orthogonal nonlinear component similar to the one used for the analysis of the inner product term above, we write

$$\|\boldsymbol{\beta}_{\mathrm{KFAC}}\|_2^2 = n^{-2}\,\boldsymbol{y}^\top \mathbf{X}\mathbf{R}^2\mathbf{X}^\top \boldsymbol{y} = n^{-2}\boldsymbol{\varepsilon}^\top \mathbf{X}\mathbf{R}^2\mathbf{X}^\top \boldsymbol{\varepsilon} + c_{\star,1}^2 n^{-2}\boldsymbol{\beta}_\star^\top \mathbf{X}^\top \mathbf{X}\mathbf{R}^2\mathbf{X}^\top \mathbf{X}\boldsymbol{\beta}_\star$$

$$+ n^{-2}\sigma_{\star,\perp}(\mathbf{X}\boldsymbol{\beta}_\star)^\top \mathbf{X}\mathbf{R}^2\mathbf{X}^\top \sigma_{\star,\perp}(\mathbf{X}\boldsymbol{\beta}_\star) + 2\,c_{\star,1}\,n^{-2}\boldsymbol{\beta}_\star^\top \mathbf{X}^\top \mathbf{X}\mathbf{R}^2\mathbf{X}^\top \sigma_{\star,\perp}(\mathbf{X}\boldsymbol{\beta}_\star).$$

We will analyze each of these terms separately.

- **First Term.** Recalling that $\boldsymbol{\varepsilon} \sim \mathsf{N}(0, \sigma_\varepsilon^2 \mathbf{I}_n)$ independent of all randomness in the problem, using the Hanson-Wright inequality (Theorem E.6) we have

$$n^{-2}\boldsymbol{\varepsilon}^\top \mathbf{X}\mathbf{R}^2\mathbf{X}^\top \boldsymbol{\varepsilon} = \sigma_\varepsilon^2 n^{-1}\,\mathrm{Tr}((\mathbf{X}^\top \mathbf{X}/n)\mathbf{R}^2) + o(1).$$

- **Second Term.** Using Assumption 3.8, and by the Hanson-Wright inequality we have

$$n^{-2}c_{\star,1}^2\,\boldsymbol{\beta}_\star^\top \mathbf{X}^\top \mathbf{X}\mathbf{R}^2\mathbf{X}^\top \mathbf{X}\boldsymbol{\beta}_\star = c_{\star,1}^2 \mathrm{d}_{\mathrm{x}}^{-1}\,\mathrm{Tr}\left[(\mathbf{X}^\top \mathbf{X}/n)^2\mathbf{R}^2\right] + o(1)$$

- **Third Term.** Note that $\bar{\mathbf{R}}$ and $\mathbf{X}\boldsymbol{\beta}_\star$ are dependent. Note that $\mathbf{X}\mathbf{R}^2\mathbf{X}^\top = \mathbf{X}\mathbf{X}^\top \bar{\mathbf{R}} = n\bar{\mathbf{R}} - \lambda_{\mathbf{G}} n\bar{\mathbf{R}}^2$. Thus, an almost identical argument to the argument used above for the analysis of $\boldsymbol{\beta}_\star^\top \boldsymbol{\beta}_{\mathrm{KFAC}}$ using $\tilde{\mathbf{X}} = \mathbf{X} - \mathbf{X}\boldsymbol{\beta}_\star \boldsymbol{\beta}_\star^\top$ gives

$$n^{-2}\sigma_{\star,\perp}(\mathbf{X}\boldsymbol{\beta}_\star)^\top \mathbf{X}\mathbf{R}^2\mathbf{X}^\top \sigma_{\star,\perp}(\mathbf{X}\boldsymbol{\beta}_\star) = \mathsf{E}_z[\sigma_{\star,\perp}^2(z)] \cdot n^{-1}\,\mathrm{Tr}\left((\mathbf{X}^\top \mathbf{X}/n)\,\mathbf{R}^2\right) + o(1)$$

- **Fourth Term.** This term can readily be shown to be $o(1)$ in the analysis of $\boldsymbol{\beta}_\star^\top \boldsymbol{\beta}_{\mathrm{KFAC}}$; i.e.,

$$2\,c_{\star,1}\,n^{-2}\boldsymbol{\beta}_\star^\top \mathbf{X}^\top \mathbf{X}\mathbf{R}^2\mathbf{X}^\top \sigma_{\star,\perp}(\mathbf{X}\boldsymbol{\beta}_\star) = o(1).$$

Putting everything together, we find

$$\|\boldsymbol{\beta}_{\mathrm{KFAC}}\|_2^2 = c_{\star,1}^2 \mathrm{d}_{\mathrm{x}}^{-1}\,\mathrm{Tr}\left[(\mathbf{X}^\top \mathbf{X}/n)^2\,\mathbf{R}^2\right] + (\sigma_\varepsilon^2 + \mathsf{E}_z[\sigma_{\star,\perp}^2(z)])\,n^{-1}\,\mathrm{Tr}\left((\mathbf{X}^\top \mathbf{X}/n)\,\mathbf{R}^2\right) \tag{31}$$

Now, given (30) and (31), defining $c_{\star,>1} = \mathsf{E}_z[\sigma_{\star,\perp}^2(z)]$, we have

$$\frac{\boldsymbol{\beta}_\star^\top \boldsymbol{\beta}_{\mathrm{KFAC}}}{\|\boldsymbol{\beta}_{\mathrm{KFAC}}\|\|\boldsymbol{\beta}_\star\|} = \frac{c_{\star,1}\mathrm{d}_{\mathrm{x}}^{-1}\,\mathrm{Tr}\left((\mathbf{X}^\top \mathbf{X}/n)\,\mathbf{R}\right)}{\sqrt{c_{\star,1}^2 \mathrm{d}_{\mathrm{x}}^{-1}\,\mathrm{Tr}\left[(\mathbf{X}^\top \mathbf{X}/n)^2\,\mathbf{R}^2\right] + (\sigma_\varepsilon^2 + c_{\star,>1}^2)\,n^{-1}\,\mathrm{Tr}\left((\mathbf{X}^\top \mathbf{X}/n)\,\mathbf{R}^2\right)}}. \tag{32}$$

Thus, noting that if $\mathrm{d}_{\mathrm{x}}/n \to 0$ and $\lambda_{\mathbf{G}} \to 0$, we have $\mathbf{R} \to \boldsymbol{\Sigma}_{\mathbf{x}}^{-1}$, we find

$$\lim_{\lambda \to 0}\lim_{\mathrm{d}_{\mathrm{x}}/n \to \infty} \frac{\boldsymbol{\beta}_\star^\top \boldsymbol{\beta}_{\mathrm{KFAC}}}{\|\boldsymbol{\beta}_{\mathrm{KFAC}}\|\|\boldsymbol{\beta}_\star\|} = 1,$$

proving the second part of the lemma. For the first part, we define $m(z) : \mathbb{R} \to \mathbb{R}$ as the limiting Stieltjes transform of the the empirical eigenvalue distribution of $n^{-1}\mathbf{X}^\top \mathbf{X}$; i.e.,

$$m(z) = \lim_{\mathrm{d}_{\mathrm{x}},n \to \infty} \mathrm{d}_{\mathrm{x}}^{-1}\,\mathrm{Tr}\left[\left(\mathbf{X}^\top \mathbf{X}/n - z\,\mathbf{I}_{\mathrm{d}_{\mathrm{x}}}\right)^{-1}\right] \tag{33}$$

where the limit is taken under the assumption that $\mathrm{d}_{\mathrm{x}}/n \to \phi > 0$. For a general covariance matrix $\boldsymbol{\Sigma}_{\mathbf{X}}$, $m(z)$ does not have a closed form except for very special cases; however, it it can be efficiently computed. See Section E.8 for more details. The derivative of the function $m$ is given by

$$m'(z) = -\lim_{\mathrm{d}_{\mathrm{x}},n \to \infty} \mathrm{d}_{\mathrm{x}}^{-1}\,\mathrm{Tr}\left[\left(\mathbf{X}^\top \mathbf{X}/n - z\,\mathbf{I}_{\mathrm{d}_{\mathrm{x}}}\right)^{-2}\right].$$

We can write all the traces appearing in (32) in terms of the function $m$ and its derivative:

$$d_x^{-1} \operatorname{Tr} \left( (\mathbf{X}^\top \mathbf{X}/n) \, \mathbf{R} \right) = d_x^{-1} \operatorname{Tr} \left( (\mathbf{X}^\top \mathbf{X}/n + \lambda_\mathbf{G} \mathbf{I}_{d_x} - \lambda_\mathbf{G} \mathbf{I}_{d_x}) \, \mathbf{R} \right) = d_x^{-1} \operatorname{Tr} \left( \mathbf{I}_{d_x} - \lambda_\mathbf{G} \mathbf{R} \right) = 1 - \lambda_\mathbf{G} \, m(-\lambda_\mathbf{G}),$$

$$d_x^{-1} \operatorname{Tr} \left( (\mathbf{X}^\top \mathbf{X}/n)^2 \, \mathbf{R}^2 \right) = d_x^{-1} \operatorname{Tr} \left( (\mathbf{X}^\top \mathbf{X}/n + \lambda_\mathbf{G} \mathbf{I}_{d_x} - \lambda_\mathbf{G} \mathbf{I}_{d_x})^2 \, \mathbf{R}^2 \right) = 1 - \lambda_\mathbf{G}^2 \, m'(-\lambda_\mathbf{G}) - 2\lambda_\mathbf{G} \, m(-\lambda_\mathbf{G}),$$

$$n^{-1} \operatorname{Tr} \left( (\mathbf{X}^\top \mathbf{X}/n) \, \mathbf{R}^2 \right) = n^{-1} \operatorname{Tr} \left( \mathbf{R} - \lambda_\mathbf{G} \mathbf{R}^2 \right) = \phi \, m(-\lambda_\mathbf{G}) + \phi \, \lambda_\mathbf{G} \, m'(-\lambda_\mathbf{G}).$$

With these, the correlation is given by

$$\frac{\boldsymbol{\beta}_{\text{KFAC}}^\top \boldsymbol{\beta}_\star}{\|\boldsymbol{\beta}_{\text{KFAC}}\|_2 \|\boldsymbol{\beta}_\star\|_2} = \frac{c_{\star,1} [\, 1 - \lambda_\mathbf{G} \, m(-\lambda_\mathbf{G}) \,]}{\sqrt{c_{\star,1}^2 [\, 1 - \lambda_\mathbf{G}^2 \, m'(-\lambda_\mathbf{G}) - 2\lambda_\mathbf{G} \, m(-\lambda_\mathbf{G}) \,] + \phi(c_{\star,>1}^2 + \sigma_\varepsilon^2)[\, m(-\lambda_\mathbf{G}) + \lambda_\mathbf{G} m'(-\lambda_\mathbf{G}) \,]}},$$

which defining

$$\Psi_1 = 1 - \lambda_\mathbf{G} \, m(-\lambda_\mathbf{G})$$
$$\Psi_2 = 1 - \lambda_\mathbf{G}^2 \, m'(-\lambda_\mathbf{G}) - 2\lambda_\mathbf{G} \, m(-\lambda_\mathbf{G})$$
$$\Psi_3 = m(-\lambda_\mathbf{G}) + \lambda_\mathbf{G} m'(-\lambda_\mathbf{G}) \tag{34}$$

concludes the proof. $\qquad\square$

## C.4. From Feature Learning to Generalization

In Section 3.2, we showed that after one step of SGD and KFAC, the first layer weights will become approximately equal to

$$\widehat{\mathbf{G}}_a \approx \widehat{\mathbf{G}}_0 + \alpha\eta \, \mathbf{f}_0 \hat{\boldsymbol{\beta}}_a^\top, \quad a \in \{\text{SGD}, \text{KFAC}\}. \tag{35}$$

Given Lemma 3.10 and Lemma 3.12, we argued that compared to SGD, the weights obtained by the KFAC algorithm are more aligned to the true direction $\boldsymbol{\beta}_\star$. Given a nontrivial alignment between the weights and the target direction, the second layer $\mathbf{f}$ can be trained using least squares (or based on Lemma 3.1, equivalently using one step of the KFAC update on $\mathbf{f}$ with $\eta_\mathbf{f} = 1$) with $\Theta(d)$ samples to achieve good generalization performance (See e.g., Ba et al. (2022, Theorem 11) and Dandi et al. (2024c, Section 3.4)). The existence of nontrivial alignment of the learned weights and the true direction in a single index model is often called *weak recovery* and has been subject to extensive investigation (see e.g., Ben Arous et al. (2021); Dandi et al. (2024c); Troiani et al. (2024); Arnaboldi et al. (2024), etc.).

To see this, consider the feature matrix $\mathbf{Z}_a \in \mathbb{R}^{n \times d_h}$ as $\mathbf{Z}_a = \sigma(\mathbf{X}\widehat{\mathbf{G}}_a^\top)$, where the activation function is applied element-wise. Based on equation (35), this matrix can be written as

$$\mathbf{Z}_a \approx \sigma \left( \mathbf{X}\mathbf{G}_0^\top + \alpha\eta \, (\mathbf{X}\hat{\boldsymbol{\beta}}_a)\mathbf{f}_0^\top \right).$$

This is an example of a random matrix in which a nonlinear function is applied element-wise to a random component plus a rank-one signal component which has been studied in the literature (Guionnet et al., 2023; Moniri et al., 2024; Moniri & Hassani, 2024b). In particular, by Taylor expanding the activation function, the feature matrix $\mathbf{Z}_a$ can be written as

$$\mathbf{Z}_a \approx \sigma(\mathbf{X}\mathbf{G}_0^\top) + \sum_{k=1}^{\ell} \frac{\alpha^k \eta^k}{k!} \left( \sigma^{(k)}(\mathbf{X}\mathbf{G}_0^\top) \right) \odot \left( (\mathbf{X}\hat{\boldsymbol{\beta}}_a)^{\circ k} \, \mathbf{f}_0^{\circ k \top} \right) + \mathcal{E}_\ell,$$

where $\circ$ denotes element-wise power and $\mathcal{E}_\ell$ is the reminder term. Let $\eta = n^\alpha$ for some $\alpha \in [0, 0.5)$. Given $\alpha$, the integer $\ell$ is chosen to be large enough so that the operator norm of $\mathcal{E}_\ell$ is $o(d_x^{1/2})$ and the reminder term is negligible compared to $\sigma(\mathbf{X}\mathbf{G}_0^\top)$. By a simple concentration argument, the matrix $\sigma^{(k)}(\mathbf{X}\mathbf{G}_0^\top)$ can be replaced with its mean $\mathsf{E} \left( \sigma^{(k)}(\mathbf{X}\mathbf{G}_0^\top) \right) = \mu \mathbf{1}\mathbf{1}^\top$ to get

$$\mathbf{Z}_a \approx \sigma(\mathbf{X}\mathbf{G}_0^\top) + \sum_{k=1}^{\ell} \frac{\alpha^k \eta^k \mu}{k!} (\mathbf{X}\hat{\boldsymbol{\beta}}_a)^{\circ k} \, \mathbf{f}_0^{\circ k \top}.$$

The first term $\sigma(\mathbf{X}\mathbf{G}_0^\top)$ is the feature matrix of a random feature model and based on the Gaussian Equivalence Theorem (GET) (see e.g., Goldt et al. (2022); Hu & Lu (2023); Dandi et al. (2024a); Moniri et al. (2024)), we can linearize it; i.e., we can replace it with $\alpha \mathbf{X}\mathbf{G}_0^\top + \mathbf{N}$ where $\mathbf{N}$ is a properly scaled independent Gaussian noise. The vectors $(\mathbf{X}\hat{\beta}_a)^{\circ k}$ are nonlinear functions of the covariates with different degrees. The least squares estimator $\hat{\mathbf{f}}_a$ is then fit on the features $\mathbf{Z}_a$ in a way that $\widehat{\mathcal{L}}(\hat{\mathbf{f}}_a, \widehat{\mathbf{G}}_a)$ is minimized; i.e.

$$\boldsymbol{y} \approx \sigma(\mathbf{X}\mathbf{G}_0^\top)\hat{\mathbf{f}}_a + \sum_{k=1}^{\ell} \frac{\alpha^k \eta^k \mu\,(\mathbf{f}_0^{\circ k\top}\hat{\mathbf{f}}_a)}{k!}(\mathbf{X}\hat{\beta}_a)^{\circ k}. \tag{36}$$

Based on the GET, the random feature component can only learn linear functions with sample complexity of learning $n = \Theta(\mathrm{d_h}) = \Theta(\mathrm{d_x})$. When $\eta$ is large enough and $\beta_a$ is aligned to $\beta_\star$, with the finite dimensional correction to the random features model, the model can also represent nonlinear functions $(\boldsymbol{x}^\top\hat{\beta}_a)^k$ of degree $k \leq \ell$ by matching the coefficients $\alpha^k \eta^k \mu\,(\mathbf{f}_0^{\circ k\top}\hat{\mathbf{f}}_a)/k!$ with the Taylor coefficients of the teacher function $\sigma_\star(\boldsymbol{x}^\top\hat{\beta}_\star)$.

Although we have provided a complete proof sketch for providing generalization guarantees given weight alignment, a complete analysis require tedious computations and is beyond the scope of this work as we mainly focus on feature learning properties of different optimization algorithms.

## D. Additional Information Regarding Kronecker-Factored Preconditioners

Here, we provide some additional background information regarding key Kronecker-Factored preconditioning methods, including their derivation and relations to various methods in the literature. We recall the running example of a fully-connected net omitting biases, introducing layer-wise dimensionality and a final non-linear layer (e.g. softmax) for completeness:

$$f_{\boldsymbol{\theta}}(\mathbf{x}) = \phi(\mathbf{W}_L \sigma(\mathbf{W}_{L-1} \cdots \sigma(\mathbf{W}_1 \mathbf{x}) \cdots)), \quad \mathbf{W}_\ell \in \mathbb{R}^{d_\ell \times d_{\ell-1}}, d_0 = \mathrm{d_x}. \tag{37}$$

As before, we define $\boldsymbol{\theta}$ as the concatenation of $\boldsymbol{\theta}_\ell = \mathrm{vec}(\mathbf{W}_\ell)$, $\ell \in [L]$. We define an expected loss induced by the neural network $\mathcal{L}(\boldsymbol{\theta}) = \mathsf{E}_{(\mathbf{x},\mathbf{y})}[\ell(f_{\boldsymbol{\theta}}(\mathbf{x}), \mathbf{y})]$, and its batch counterpart $\widehat{\mathcal{L}}(\boldsymbol{\theta})$. Here, we define the family of Kronecker-Factored preconditioned optimizers as those that update weights in the following fashion:

$$\mathbf{W}_{\ell+} = \mathbf{W}_\ell - \eta\,\mathbf{P}_\ell^{-1}\nabla_{\mathbf{W}_\ell}\widehat{\mathcal{L}}(\boldsymbol{\theta})\mathbf{Q}_\ell^{-1}, \quad \ell \in [L],$$

where $\mathbf{P}_\ell \in \mathbb{R}^{d_\ell \times d_\ell}$, $\mathbf{Q}_\ell \in \mathbb{R}^{d_{\ell-1} \times d_{\ell-1}}$, $\ell \in [L]$ are square matrices. For simplicity, we ignore moving parts such as momentum, damping exponents, adaptive learning rate schedules/regularization etc. We now demonstrate the basic principles and derivation of certain notable members of these preconditioning methods on the feedforward network (37).

### D.1. Kronecker-Factored Approximate Curvature `KFAC`

As described in the main paper, `KFAC` (Martens & Grosse, 2015) is at its core an approximation to natural gradient descent. Given that we are approximating `NGD`, a crucial presumption on $f_{\boldsymbol{\theta}}(\mathbf{x})$ and $\mathcal{L}(\boldsymbol{\theta})$ is that the network output $f_{\boldsymbol{\theta}}(\mathbf{x})$ parameterizes a conditional distribution $p(\mathbf{y}|\mathbf{x}; \boldsymbol{\theta})$, and $\mathcal{L}(\boldsymbol{\theta}) \propto \mathsf{E}_{(\mathbf{x},\mathbf{y})}[-\log p(\mathbf{y}|\mathbf{x}; \boldsymbol{\theta})]$ is the corresponding negative log-likelihood. As such, `KFAC` is technically only applicable to settings where such an interpretation exists. However, this notably subsumes cases $\mathcal{L}(\boldsymbol{\theta}) = \mathsf{E}_{(\mathbf{x},\mathbf{y})}[\ell(f_{\boldsymbol{\theta}}(\mathbf{x}), \mathbf{y})]$, where $\ell(\cdot)$ is a strictly convex function in $f_{\boldsymbol{\theta}}(\mathbf{x})$, as this admits an interpretation as $f_{\boldsymbol{\theta}}(\mathbf{x})$ parameterizing an exponential family distribution. In particular, the square-loss regression case $\ell(\hat{\mathbf{y}}, \mathbf{y}) = \|\hat{\mathbf{y}} - \mathbf{y}\|^2$ corresponds to a conditionally-Gaussian predictive distribution with fixed variance $\hat{\mathbf{y}}(\mathbf{x}) \sim \mathsf{N}(f_{\boldsymbol{\theta}}(\mathbf{x}), \sigma^2\mathbf{I})$, and if $\phi(\cdot)$ is a softmax layer and $\ell(\hat{\mathbf{y}}, \mathbf{y}) = \mathrm{CrossEnt}(\hat{\mathbf{y}}, \mathbf{y})$, the multi-class classification case corresponds to a conditionally-multinomial predictive distribution.

Defining $\mathbf{h}_\ell = \mathbf{W}_\ell\mathbf{z}_{\ell-1}$, $\mathbf{z}_\ell = \sigma(\mathbf{h})$, $\mathbf{z}_0 = \mathbf{x}$, the Fisher Information of the predictive distribution $p(\mathbf{y}|\mathbf{x}; \boldsymbol{\theta})$ at $\boldsymbol{\theta}$ can be

expressed in block form:

$$\mathbf{FI}(\boldsymbol{\theta}) \triangleq \mathsf{E}_{\mathbf{x}} \left[ \frac{\partial p(\mathbf{y}|\mathbf{x}; \boldsymbol{\theta})}{\partial \boldsymbol{\theta}} \left( \frac{\partial p(\mathbf{y}|\mathbf{x}; \boldsymbol{\theta})}{\partial \boldsymbol{\theta}} \right)^{\top} \right] \qquad \text{(recall } \boldsymbol{\theta} \text{ is vec-ed parameters)}$$

$$= \begin{bmatrix} \mathsf{E}_{\mathbf{x}} \left[ \frac{\partial p(\mathbf{y}|\mathbf{x};\boldsymbol{\theta})}{\partial \boldsymbol{\theta}_1} \left( \frac{\partial p(\mathbf{y}|\mathbf{x};\boldsymbol{\theta})}{\partial \boldsymbol{\theta}_1} \right)^{\top} \right] & \cdots & \mathsf{E}_{\mathbf{x}} \left[ \frac{\partial p(\mathbf{y}|\mathbf{x};\boldsymbol{\theta})}{\partial \boldsymbol{\theta}_1} \left( \frac{\partial p(\mathbf{y}|\mathbf{x};\boldsymbol{\theta})}{\partial \boldsymbol{\theta}_L} \right)^{\top} \right] \\ \vdots & \ddots & \vdots \\ \mathsf{E}_{\mathbf{x}} \left[ \frac{\partial p(\mathbf{y}|\mathbf{x};\boldsymbol{\theta})}{\partial \boldsymbol{\theta}_L} \left( \frac{\partial p(\mathbf{y}|\mathbf{x};\boldsymbol{\theta})}{\partial \boldsymbol{\theta}_1} \right)^{\top} \right] & \cdots & \mathsf{E}_{\mathbf{x}} \left[ \frac{\partial p(\mathbf{y}|\mathbf{x};\boldsymbol{\theta})}{\partial \boldsymbol{\theta}_L} \left( \frac{\partial p(\mathbf{y}|\mathbf{x};\boldsymbol{\theta})}{\partial \boldsymbol{\theta}_L} \right)^{\top} \right] \end{bmatrix}$$

Looking at the $(i, j)$th block, we have

$$\mathsf{E}_{\mathbf{x}} \left[ \frac{\partial p(\mathbf{y}|\mathbf{x}; \boldsymbol{\theta})}{\partial \boldsymbol{\theta}_i} \left( \frac{\partial p(\mathbf{y}|\mathbf{x}; \boldsymbol{\theta})}{\partial \boldsymbol{\theta}_j} \right)^{\top} \right] = \mathsf{E}_{\mathbf{x}} \left[ \mathrm{vec} \left( \frac{\partial p(\mathbf{y}|\mathbf{x}; \boldsymbol{\theta})}{\partial \mathbf{W}_i} \right) \mathrm{vec} \left( \frac{\partial p(\mathbf{y}|\mathbf{x}; \boldsymbol{\theta})}{\partial \mathbf{W}_j} \right)^{\top} \right]$$

$$= \mathsf{E}_{\mathbf{x}} \left[ (\mathbf{z}_{i-1} \otimes \mathbf{g}_i)(\mathbf{z}_{j-1} \otimes \mathbf{g}_j)^{\top} \right] \qquad (\mathbf{g}_{\ell} \triangleq -\frac{\partial p(\mathbf{y}|\mathbf{x};\boldsymbol{\theta})}{\partial \mathbf{h}_{\ell}})$$

$$= \mathsf{E}_{\mathbf{x}} \left[ (\mathbf{z}_{i-1}\mathbf{z}_{j-1}^{\top}) \otimes (\mathbf{g}_i \mathbf{g}_j^{\top}) \right], \qquad \text{(Lemma E.1, item 2)}$$

where the second line comes from writing out the backpropagation formula. KFAC makes two key approximations:

1. The matrix $\mathbf{FI}(\boldsymbol{\theta})^{-1}$ is approximated by a block-diagonal, and hence so is $\mathbf{FI}(\boldsymbol{\theta})$. We note the original formulation of KFAC in Martens & Grosse (2015) also supports a tridiagonal inverse approximation.

2. The vectors $\mathbf{z}_{\ell-1}$ and $\mathbf{g}_{\ell}$ are independent for all $\ell \in [L]$, such that

$$\mathsf{E}_{\mathbf{x}} \left[ (\mathbf{z}_{\ell-1} \mathbf{z}_{\ell-1}^{\top}) \otimes (\mathbf{g}_{\ell} \mathbf{g}_{\ell}^{\top}) \right] = \mathsf{E}[\mathbf{z}_{\ell-1} \mathbf{z}_{\ell-1}^{\top}] \otimes \mathsf{E}[\mathbf{g}_{\ell} \mathbf{g}_{\ell}^{\top}].$$

Now replacing the true expectation with the empirical estimate, and defining $\mathbf{P}_{\ell} = \widehat{\mathsf{E}}[\mathbf{g}_{\ell} \mathbf{g}_{\ell}^{\top}]$, $\mathbf{Q}_{\ell} = \widehat{\mathsf{E}}[\mathbf{z}_{\ell-1} \mathbf{z}_{\ell-1}^{\top}]$ completes the Kronecker-Factored approximation to the Fisher Information. It is clear to see from the derivation that, as we previewed in the introduction and expressed emphatically in Martens & Grosse (2015), this approximation is *never* expected to be tight.

**Some related preconditioners**

Having introduced KFAC, we introduce some related preconditioners. Notably, it has been noted that computing $\mathbf{g}_{\ell}$ requires a backwards gradient computation, whereas $\mathbf{z}_{\ell}$ only requires a forward pass. In particular, various works have recovered the *right* preconditioner $\mathbf{Q}_{\ell}$ of KFAC via various notions of "local" (layer-wise) losses. Notably, these alternative views allow KFAC-like preconditioning to extend beyond the negative-log-likelihood interpretation.

- LocoProp, square-loss case (Amid et al., 2022):

$$\text{Update rule}: \mathbf{W}_{\ell+} = \underset{\mathbf{W}}{\mathrm{argmin}} \frac{1}{2} \widehat{\mathsf{E}} \left[ \|\mathbf{W}\mathbf{z}_{\ell-1} - \mathbf{h}_{\ell}\|^2 \right] + \frac{1}{2\eta} \|\mathbf{W} - \mathbf{W}_{\ell}\|_F^2$$

$$= \mathbf{W}_{\ell} - \eta \nabla_{\mathbf{W}_{\ell}} \widehat{\mathcal{L}}(\boldsymbol{\theta}) \left( \mathbf{I}_{d_{\ell-1}} + \eta \widehat{\mathsf{E}}[\mathbf{z}_{\ell-1}\mathbf{z}_{\ell-1}^{\top}] \right)^{-1}.$$

As noted in Amid et al. (2022), this update is also closely related to ProxProp (Frerix et al., 2018).

- FOOF (Benzing, 2022):

$$\text{Update rule}: \Delta\mathbf{W}_{\ell} = \underset{\Delta\mathbf{W}}{\mathrm{argmin}} \ \widehat{\mathsf{E}} \left[ \|\Delta\mathbf{W}\mathbf{z}_{\ell-1} - \eta\mathbf{g}_{\ell}\|^2 \right] + \frac{\lambda}{2} \|\Delta\mathbf{W}\|_F^2 \qquad \left( \mathbf{g}_{\ell} = \frac{\partial \ell(f_{\boldsymbol{\theta}}(\mathbf{x}), \mathbf{y})}{\partial \mathbf{h}_{\ell}} \right)$$

$$= \eta \nabla_{\mathbf{W}_{\ell}} \widehat{\mathcal{L}}(\boldsymbol{\theta}) \left( \widehat{\mathsf{E}}[\mathbf{z}_{\ell-1}\mathbf{z}_{\ell-1}^{\top}] + \lambda \mathbf{I}_{d_{\ell-1}} \right)^{-1},$$

$$\mathbf{W}_{\ell+} = \mathbf{W}_{\ell} - \Delta\mathbf{W}_{\ell}.$$

Interestingly, we note that these right-preconditioner-only variants subsume the DFW algorithm for two-layer linear representation learning proposed in Zhang et al. (2024b); thus we may see the guarantee therein as support of the above algorithms from a feature learning perspective, albeit weaker than Theorem 3.6.

### D.2. Shampoo

Shampoo is designed to be a Kronecker-Factored approximation of the full AdaGrad preconditioner, which we recall is the running sum of the outer-product of loss gradients. Turning off the AdaGrad accumulator and instead considering the empirical batch estimate $\widehat{\mathsf{E}}[\nabla_{\boldsymbol{\theta}}\ell(f_{\boldsymbol{\theta}}(\mathbf{x}),\mathbf{y})\,\nabla_{\boldsymbol{\theta}}\ell(f_{\boldsymbol{\theta}}(\mathbf{x}),\mathbf{y})^{\top}]$, the curvature matrix being estimated can also be viewed as the Gauss-Newton matrix $\mathsf{E}_{(\mathbf{x},\mathbf{y})}[\nabla_{\boldsymbol{\theta}}\ell(f_{\boldsymbol{\theta}}(\mathbf{x}),\mathbf{y})\,\nabla_{\boldsymbol{\theta}}\ell(f_{\boldsymbol{\theta}}(\mathbf{x}),\mathbf{y})^{\top}]$. As documented in various works (see e.g. Martens (2020)), the (generalized) Gauss-Newton matrix in many cases is related or equal to the Fisher Information, establishing a link between the target curvatures of KFAC and Shampoo.

However, the Shampoo preconditioners differ from KFAC's. Let us define the $\mathbf{z}_{\ell}, \mathbf{h}_{\ell}$ as before, and $\mathbf{g}_{\ell} = \frac{\partial\ell(f_{\boldsymbol{\theta}}(\mathbf{x}),\mathbf{y})}{\partial\mathbf{h}_{\ell}}$. Then, the Shampoo preconditioners are given by

$$\mathbf{P}_{\ell} = \widehat{\mathsf{E}}\left[\mathbf{g}_{\ell}\mathbf{z}_{\ell-1}^{\top}(\mathbf{g}_{\ell}\mathbf{z}_{\ell-1}^{\top})^{\top}\right]^{1/4}, \quad \mathbf{Q}_{\ell} = \widehat{\mathsf{E}}\left[\mathbf{z}_{\ell-1}\mathbf{g}_{\ell}^{\top}(\mathbf{z}_{\ell-1}\mathbf{g}_{\ell}^{\top})^{\top}\right]^{1/4}.$$

Notably, Shampoo takes the fourth root in the preconditioners, as its target is the AdaGrad preconditioner which is (modulo scaling) the square-root of the empirical Gauss-Newton matrix–analogous to the square-root of the second moment in Adam. Whether the target curvature should be the square-root or not of the Gauss-Newton matrix is the topic of recent discussion (Morwani et al., 2024; Lin et al., 2024).

### D.3. Kronecker-Factored Preconditioners and the Modular Norm

The "modular norm" (Large et al., 2024; Bernstein & Newhouse, 2024a;b) is a recently introduced notion that provides a general recipe for producing different optimization algorithms that act layer-wise. By specifying different norms customized for different kinds of layers (e.g. feed-forward, residual, convolutional etc.), one in principle has the flexibility to customize an optimizer to handle the different kinds of curvature induced by different parameter spaces. Given a choice of norm on the weight tensor $\mathbf{W}_{\ell}$, the descent direction is returned by *steepest descent* with respect to that norm. To introduce steepest descent, we require a few definitions (cf. Bernstein & Newhouse (2024a)):

**Definition D.1** (Dual norms, steepest direction). Given a norm $\|\cdot\|$ defined over a finite-dimensional real vector space $\mathcal{V}$. The dual norm $\|\cdot\|_{\dagger}$ is defined by

$$\|\mathbf{v}\|_{\dagger} = \max_{\|\mathbf{u}\|=1} \langle \mathbf{u}, \mathbf{v} \rangle.$$

With $\mathbf{g} \in \mathcal{V}$ and a "sharpness" parameter $\eta > 0$, the steepest direction(s) are given by the following variational representation:

$$\operatorname*{argmin}_{\mathbf{d}} \left[ \langle \mathbf{g}, \mathbf{d} \rangle + \frac{1}{2\eta}\|\mathbf{d}\|^{2} \right] = -\eta\|\mathbf{g}\|_{\dagger} \cdot \operatorname*{argmax}_{\|\mathbf{u}\|=1} \langle \mathbf{g}, \mathbf{u} \rangle.$$

Here we focus on finite-dimensional normed spaces, but note that these concepts extend *mutatis mutandis* to general Banach spaces. The aforementioned works derive various standard optimizers by choosing different norms, including *induced matrix norms* $\|\mathbf{W}_{\ell}\|_{\alpha\to\beta} = \max_{\mathbf{x}} \frac{\|\mathbf{W}_{\ell}\mathbf{x}\|_{\beta}}{\|\mathbf{x}\|_{\alpha}}$, applied to a given layer's weight space, for example (Bernstein & Newhouse, 2024b):

- SGD: induced by Frobenius (Euclidean) norm $\|\cdot\| = \|\cdot\|_{F}$. Note the Frobenius norm is *not* an induced matrix norm.

- Sign-descent ("ideal" Adam with EMA on moments turned off): induced by $\|\cdot\| = \|\cdot\|_{\ell_1\to\ell_\infty}$.

- Shampoo ("ideal" variant with moment accumulator turned off): induced by $\|\cdot\| = \|\cdot\|_{\ell_2\to\ell_2} = \|\cdot\|_{\mathrm{op}}$.

Therefore, in light of this characterization, a natural question to ask is *what norm induces a given Kronecker-Factored preconditioner* (which includes Shampoo). We provide a simple derivation that determines the norm.

**Proposition D.2** (Kronecker-Factored matrix norm)**.** *Recall the fully-connected network* (37)*. Given preconditioners* $\{(\mathbf{P}_\ell, \mathbf{Q}_\ell)\}_{\ell=1}^L$*, where* $\mathbf{P}_\ell \in \mathbb{R}^{d_\ell \times d_\ell}$*,* $\mathbf{Q}_\ell \in \mathbb{R}^{d_{\ell-1} \times d_{\ell-1}}$*,* $\ell \in [L]$ *are invertible square matrices. Then, the layer-wise Kronecker-Factored update:*

$$\mathbf{W}_{\ell+} = \mathbf{W}_\ell - \eta\,\mathbf{P}_\ell^{-1} \nabla_{\mathbf{W}_\ell} \mathcal{L}(\boldsymbol{\theta})\,\mathbf{Q}_\ell^{-1}, \quad \ell \in [L]$$

*is equivalent to layer-wise steepest descent with norm* $\|\mathbf{M}_\ell\| \triangleq \|\mathbf{P}_\ell^\top \mathbf{M}_\ell \mathbf{Q}_\ell^\top\|_F$*:*

$$\operatorname*{argmin}_{\mathbf{M}} \left[ \langle \nabla_{\mathbf{W}_\ell} \mathcal{L}(\boldsymbol{\theta}), \mathbf{M} \rangle + \frac{1}{2\eta} \|\mathbf{M}\|^2 \right] = -\eta\,\mathbf{P}_\ell^{-1} \nabla_{\mathbf{W}_\ell} \mathcal{L}(\boldsymbol{\theta})\,\mathbf{Q}_\ell^{-1}.$$

*Proof of Proposition D.2.* It is straightforward to verify $\|\mathbf{M}\| \triangleq \|\mathbf{P}^\top \mathbf{M} \mathbf{Q}^\top\|_F$ for invertible $\mathbf{P}, \mathbf{Q}$ satisfies the axioms of a norm. It remains to verify the steepest descent direction:

$$\operatorname*{argmin}_{\mathbf{M}} \left[ \langle \nabla_{\mathbf{W}_\ell} \mathcal{L}(\boldsymbol{\theta}), \mathbf{M} \rangle + \frac{1}{2\eta} \|\mathbf{M}\|^2 \right] = -\eta\, \|\nabla_{\mathbf{W}_\ell} \mathcal{L}(\boldsymbol{\theta})\|_\dagger \cdot \operatorname*{argmax}_{\|\mathbf{M}\|=1} \langle \nabla_{\mathbf{W}_\ell} \mathcal{L}(\boldsymbol{\theta}), \mathbf{M} \rangle = \mathbf{P}_\ell^{-1} \nabla_{\mathbf{W}_\ell} \mathcal{L}(\boldsymbol{\theta})\,\mathbf{Q}_\ell^{-1}.$$

We start by writing:

$$
\begin{aligned}
\|\nabla_{\mathbf{W}_\ell} \mathcal{L}(\boldsymbol{\theta})\|_\dagger &\triangleq \max_{\|\mathbf{M}\|=1} \langle \nabla_{\mathbf{W}_\ell} \mathcal{L}(\boldsymbol{\theta}), \mathbf{M} \rangle \\
&= \max_{\|\mathbf{P}_\ell^\top \mathbf{M} \mathbf{Q}_\ell^\top\|_F = 1} \operatorname{Tr}(\mathbf{M}^\top \nabla_{\mathbf{W}_\ell} \mathcal{L}(\boldsymbol{\theta})) \\
&= \max_{\|\mathbf{D}\|_F = 1} \operatorname{Tr}(\mathbf{Q}_\ell^{-1} \mathbf{D}^\top \mathbf{P}_\ell^{-1} \nabla_{\mathbf{W}_\ell} \mathcal{L}(\boldsymbol{\theta})) && (\mathbf{P}_\ell^\top \mathbf{M} \mathbf{Q}_\ell^\top \to \mathbf{D}) \\
&= \|\mathbf{P}_\ell^{-1} \nabla_{\mathbf{W}_\ell} \mathcal{L}(\boldsymbol{\theta})\,\mathbf{Q}_\ell^{-1}\|_F. && (\text{trace cyclic property, } \|\cdot\|_F \text{ is self-dual})
\end{aligned}
$$

Similarly, it is straightforward to verify that the maximizing matrix is:

$$\operatorname*{argmax}_{\|\mathbf{M}\|=1} \langle \nabla_{\mathbf{W}_\ell} \mathcal{L}(\boldsymbol{\theta}), \mathbf{M} \rangle = \frac{\mathbf{P}_\ell^{-1} \nabla_{\mathbf{W}_\ell} \mathcal{L}(\boldsymbol{\theta})\,\mathbf{Q}_\ell^{-1}}{\|\mathbf{P}_\ell^{-1} \nabla_{\mathbf{W}_\ell} \mathcal{L}(\boldsymbol{\theta})\,\mathbf{Q}_\ell^{-1}\|_F},$$

such that plugging it into the steepest descent expression yields:

$$
\begin{aligned}
\operatorname*{argmin}_{\mathbf{M}} \left[ \langle \nabla_{\mathbf{W}_\ell} \mathcal{L}(\boldsymbol{\theta}), \mathbf{M} \rangle + \frac{1}{2\eta} \|\mathbf{M}\|^2 \right] &= -\eta\, \|\mathbf{P}_\ell^{-1} \nabla_{\mathbf{W}_\ell} \mathcal{L}(\boldsymbol{\theta})\,\mathbf{Q}_\ell^{-1}\|_F \cdot \frac{\mathbf{P}_\ell^{-1} \nabla_{\mathbf{W}_\ell} \mathcal{L}(\boldsymbol{\theta})\,\mathbf{Q}_\ell^{-1}}{\|\mathbf{P}_\ell^{-1} \nabla_{\mathbf{W}_\ell} \mathcal{L}(\boldsymbol{\theta})\,\mathbf{Q}_\ell^{-1}\|_F} \\
&= -\eta\,\mathbf{P}_\ell^{-1} \nabla_{\mathbf{W}_\ell} \mathcal{L}(\boldsymbol{\theta})\,\mathbf{Q}_\ell^{-1},
\end{aligned}
$$

as required. $\qquad\square$

We remark that for complex-valued matrices, the above holds without modification for the Hermitian transpose $\mathbf{A}^{\mathrm{H}}$. Notably, the layer-wise norm corresponding to Kronecker-Factored preconditioning is not an induced matrix norm, though modified optimizers can certainly be derived via induced-norm variants, such as a "Mahalonobis-to-Mahalonobis" induced norm:

$$
\begin{aligned}
\|\mathbf{M}\|_{\mathbf{Q}^{-1} \to \mathbf{P}} &\triangleq \max_{\mathbf{x}} \frac{\sqrt{(\mathbf{M}\mathbf{x})^\top \mathbf{P}(\mathbf{M}\mathbf{x})}}{\sqrt{\mathbf{x}^\top \mathbf{Q}^{-1} \mathbf{x}}} && (\mathbf{P}, \mathbf{Q} \succ 0) \\
&= \max_{\|\mathbf{x}\|=1} \|\mathbf{P}^{1/2} \mathbf{M} \mathbf{Q}^{1/2} \mathbf{x}\| \\
&= \left\| \mathbf{P}^{1/2} \mathbf{M} \mathbf{Q}^{1/2} \right\|_{\mathrm{op}}.
\end{aligned}
$$

# E. Auxiliary Results

## E.1. Properties of Kronecker Product

Recall the definition of the Kronecker Product: given $\mathbf{A} \in \mathbb{R}^{m \times n}$, $\mathbf{B} \in \mathbb{R}^{p \times q}$

$$\mathbf{A} \otimes \mathbf{B} = \begin{bmatrix} A_{11}\mathbf{B} & \cdots & A_{1n}\mathbf{B} \\ \vdots & \ddots & \vdots \\ A_{m1}\mathbf{B} & \cdots & A_{mn}\mathbf{B} \end{bmatrix} \in \mathbb{R}^{mp \times nq}.$$

Complementarily, the vectorization operator $\mathrm{vec}(\mathbf{A})$ is defined by stacking the columns of $\mathbf{A}$ on top of each other (i.e. column-major order)

$$\mathrm{vec}(\mathbf{A}) = \begin{bmatrix} A_{11} & \cdots & A_{m1} & \cdots & A_{1n} & \cdots & A_{mn} \end{bmatrix}^\top \in \mathbb{R}^{mn}.$$

We now introduce some fundamental facts about the Kronecker Product.

**Lemma E.1** (Kronecker-Product Properties). *The following properties hold:*

1. $(\mathbf{A} \otimes \mathbf{B})^{-1} = \mathbf{A}^{-1} \otimes \mathbf{B}^{-1}$. *Holds for Moore-Penrose pseudoinverse* $^\dagger$ *as well.*

2. *For size-compliant* $\mathbf{A}, \mathbf{B}, \mathbf{C}, \mathbf{D}$, *we have* $(\mathbf{A} \otimes \mathbf{B})(\mathbf{C} \otimes \mathbf{D}) = (\mathbf{A}\mathbf{C}) \otimes (\mathbf{B}\mathbf{D})$.

3. $\mathrm{vec}(\mathbf{A}\mathbf{X}\mathbf{B}) = (\mathbf{B}^\top \otimes \mathbf{A})\mathrm{vec}(\mathbf{X})$.

## E.2. Covariance Concentration

We often use the following Gaussian covariance concentration result.

**Lemma E.2** (Gaussian covariance concentration). *Let* $\mathbf{x}_i \overset{\text{i.i.d.}}{\sim} \mathsf{N}(\mathbf{0}, \boldsymbol{\Sigma}_\mathbf{x})$ *for* $i = 1, \ldots, n$, *where* $\mathbf{x}_i \in \mathbb{R}^d$. *Defining the empirical covariance matrix* $\widehat{\boldsymbol{\Sigma}}_\mathbf{x} \triangleq \frac{1}{n}\sum_{i=1}^n \mathbf{x}_i\mathbf{x}_i^\top$, *as long as* $n \geq \frac{18.27}{c^2}(d + \log(1/\delta))$, *we have with probability at least* $1 - \delta$,

$$(1 - c)\boldsymbol{\Sigma}_\mathbf{x} \preceq \widehat{\boldsymbol{\Sigma}}_\mathbf{x} \preceq (1 + c)\boldsymbol{\Sigma}_\mathbf{x}.$$

*Proof of Lemma E.2.* The result follows essentially from combining a by-now standard concentration inequality for Gaussian quadratic forms and a covering number argument. To be precise, we observe that

$$\left\|\widehat{\boldsymbol{\Sigma}}_\mathbf{x} - \boldsymbol{\Sigma}_\mathbf{x}\right\|_{\mathrm{op}} \leq c\|\boldsymbol{\Sigma}_\mathbf{x}\| \implies (1 - c)\boldsymbol{\Sigma}_\mathbf{x} \preceq \widehat{\boldsymbol{\Sigma}}_\mathbf{x} \preceq (1 + c)\boldsymbol{\Sigma}_\mathbf{x}.$$

Therefore, it suffices to establish a concentration bound on $\|\widehat{\boldsymbol{\Sigma}}_\mathbf{x} - \boldsymbol{\Sigma}_\mathbf{x}\|$ and invert for $c\|\boldsymbol{\Sigma}_\mathbf{x}\|$. To do so, we recall a standard covering argument (see e.g. Vershynin (2018, Chapter 4)) yields: given an $\varepsilon$-covering of $\mathbb{S}^{d-1}$, $\mathcal{N} \triangleq \mathcal{N}(\mathbb{S}^{d-1}, \|\cdot\|_2, \varepsilon)$, the operator norm of a symmetric matrix $\boldsymbol{\Sigma}$ is bounded by

$$\|\boldsymbol{\Sigma}\| \leq \frac{1}{1 - 2\varepsilon} \max_{\mathbf{u} \in \mathcal{N}} \mathbf{u}^\top \boldsymbol{\Sigma}\mathbf{u},$$

where the corresponding covering number is bounded by:

$$|\mathcal{N}(\mathbb{S}^{d-1}, \|\cdot\|_2, \varepsilon)| \leq \left(1 + \frac{2}{\varepsilon}\right)^d.$$

As such it suffices to provide a concentration bound on $\mathbf{u}^\top \boldsymbol{\Sigma}\mathbf{u}$ for each $\mathbf{u} \in \mathcal{N}$ and then union-bound. Toward establishing this, we first state the Gaussian quadratic form concentration bound due to Hsu et al. (2012), which is in turn an instantiation of a chi-squared concentration bound from Laurent & Massart (2000).

**Proposition E.3** (Prop. 1 in (Hsu et al., 2012)). *Let* $\mathbf{A} \in \mathbb{R}^{m \times d}$ *be a fixed matrix. Let* $\mathbf{g} \sim \mathsf{N}(\mathbf{0}, \mathbf{I}_d)$ *be a mean-zero, isotropic Gaussian random vector. For any* $\delta \in (0, 1)$, *we have*

$$\mathsf{P}[\|\mathbf{A}\mathbf{g}\|^2 > \mathrm{Tr}(\mathbf{A}^\top\mathbf{A}) + 2\sqrt{\mathrm{Tr}((\mathbf{A}^\top\mathbf{A})^2)\log(1/\delta)} + 2\left\|\mathbf{A}^\top\mathbf{A}\right\|_{\mathrm{op}}\log(1/\delta)] \leq \delta.$$

Now, given $\mathbf{u} \in \mathbb{S}^{d-1}$, setting $\mathbf{A} = \mathbf{u}^\top \mathbf{\Sigma}^{1/2}$ such that $\mathbf{u}^\top \mathbf{\Sigma}^{1/2} \mathbf{g} \overset{d}{=} \mathbf{u}^\top \mathbf{x}$, instantiating Proposition E.3 yields:

$$\mathsf{P}\left[\mathbf{u}^\top \widehat{\mathbf{\Sigma}}_\mathbf{x} \mathbf{u} > \mathbf{u}^\top \mathbf{\Sigma}_\mathbf{x} \mathbf{u} + 2\mathbf{u}^\top \mathbf{\Sigma}_\mathbf{x} \mathbf{u}\sqrt{\frac{\log(1/\delta)}{n}} + 2\mathbf{u}^\top \mathbf{\Sigma}_\mathbf{x} \mathbf{u}\frac{\log(1/\delta)}{n}\right] \leq \delta.$$

Put another way, this says with probability at least $1 - \delta$:

$$\mathbf{u}^\top (\widehat{\mathbf{\Sigma}}_\mathbf{x} - \mathbf{\Sigma})\mathbf{u} \leq 2\mathbf{u}^\top \mathbf{\Sigma}_\mathbf{x} \mathbf{u}\left(\sqrt{\frac{\log(1/\delta)}{n}} + \frac{\log(1/\delta)}{n}\right).$$

Taking a union bound over $\mathbf{u} \in \mathcal{N}$, we get with probability at least $1 - \delta$:

$$\max_{\mathbf{u}\in\mathcal{N}} \mathbf{u}^\top (\widehat{\mathbf{\Sigma}}_\mathbf{x} - \mathbf{\Sigma})\mathbf{u} \leq \max_{\mathbf{u}\in\mathcal{N}} 2\mathbf{u}^\top \mathbf{\Sigma}_\mathbf{x} \mathbf{u}\left(\sqrt{\frac{\log(|\mathcal{N}|/\delta)}{n}} + \frac{\log(|\mathcal{N}|/\delta)}{n}\right)$$

$$\leq 2\left\|\mathbf{\Sigma}_\mathbf{x}\right\|_{\mathrm{op}}\left(\sqrt{\frac{d\log\left(1 + \frac{2}{\varepsilon}\right) + \log(1/\delta)}{n}} + \frac{d\log\left(1 + \frac{2}{\varepsilon}\right) + \log(1/\delta)}{n}\right)$$

$$\leq 4\sqrt{\log\left(1 + \frac{2}{\varepsilon}\right)}\left\|\mathbf{\Sigma}_\mathbf{x}\right\|_{\mathrm{op}}\sqrt{\frac{d + \log(1/\delta)}{n}},$$

as long as $n \geq d\log\left(1 + \frac{2}{\varepsilon}\right) + \log(1/\delta)$. Chaining together inequalities, this yields with probability at least $1 - \delta$ under the same condition on $n$:

$$\left\|\widehat{\mathbf{\Sigma}}_\mathbf{x} - \mathbf{\Sigma}_\mathbf{x}\right\|_{\mathrm{op}} \leq \left\|\mathbf{\Sigma}_\mathbf{x}\right\|_{\mathrm{op}}\frac{2}{1 - \varepsilon}\sqrt{\log\left(1 + \frac{2}{\varepsilon}\right)}\sqrt{\frac{d + \log(1/\delta)}{n}}.$$

Minimizing the RHS for $\varepsilon \approx 0.0605$ yields the result. $\qquad\square$

### E.3. Extensions to subgaussianity

As previewed, many results can be extended from the Gaussian setting to subgaussian random vectors.

**Definition E.4.** A (scalar) random variable $X$ is *subgaussian* with variance proxy $\sigma^2$ if the following holds on its moment-generating function:

$$\mathsf{E}[\exp(\lambda X)] \leq \exp\left(\frac{\lambda^2 \sigma^2}{2}\right).$$

A mean-zero random vector $\mathbf{x} \in \mathbb{R}^d$, $\mathsf{E}[\mathbf{x}] = \mathbf{0}$, is *subgaussian* with variance proxy $\sigma^2$ if every linear projection is a $\sigma^2$-subgaussian random variable:

$$\mathsf{E}[\exp(\lambda\mathbf{v}^\top\mathbf{x})] \leq \exp\left(\frac{\lambda^2\|\mathbf{v}\|^2\sigma^2}{2}\right), \quad \text{for all } \mathbf{v} \in \mathbb{R}^d.$$

With this in hand, we may introduce the subgaussian variant of covariance concentration and the Hanson-Wright inequality.

### E.4. Subgaussian Covariance Concentration

We state the subgaussian variant of Lemma E.2, whose proof is structurally the same, replacing the $\chi^2$ random variables with a generic subexponential (c.f. Vershynin (2018, Chapter 2)) random variable, and using a generic Bernstein's inequality rather than the specific $\chi^2$ concentration inequality. The result is qualitatively identical, sacrificing tight/explicit universal numerical constants. The result is relatively standard, and can be found in e.g., Vershynin (2018, Chapter 5) or Du et al. (2021, Lemma A.6).

**Lemma E.5** (Subgaussian covariance concentration). *Let $\mathbf{x}_i$ be i.i.d. zero-mean $\sigma^2$-subgaussian random vectors for $i = 1, \ldots, n$, where $\mathbf{x}_i \in \mathbb{R}^d$, and $\mathsf{E}[\mathbf{x}\mathbf{x}^\top] = \boldsymbol{\Sigma}_\mathbf{x}$. Defining the empirical covariance matrix $\widehat{\boldsymbol{\Sigma}}_\mathbf{x} \triangleq \frac{1}{n} \sum_{i=1}^n \mathbf{x}_i \mathbf{x}_i^\top$, there exists a universal constant $C_1 > 0$ such that with probability at least $1 - 2\delta$:*

$$\left\| \widehat{\boldsymbol{\Sigma}}_\mathbf{x} - \boldsymbol{\Sigma}_\mathbf{x} \right\|_{\mathrm{op}} \leq C\sigma^2 \left\| \boldsymbol{\Sigma}_\mathbf{x} \right\|_{\mathrm{op}} \left( \sqrt{\frac{d + \log(1/\delta)}{n}} + \frac{d + \log(1/\delta)}{n} \right).$$

*Therefore, as long as $n \geq C_2 \frac{\sigma^2}{c^2}(d + \log(1/\delta))$, we have with probability at least $1 - \delta$,*

$$(1 - c)\boldsymbol{\Sigma}_\mathbf{x} \preceq \widehat{\boldsymbol{\Sigma}}_\mathbf{x} \preceq (1 + c)\boldsymbol{\Sigma}_\mathbf{x}.$$

### E.5. Hanson-Wright Inequality

We often use the following theorem to prove the concentration inequality for quadratic forms. A modern proof of this theorem can be found in Rudelson & Vershynin (2013).

**Theorem E.6** (Hanson-Wright Inequality (Hanson & Wright, 1971)). *Let $\mathbf{x} = (X_1, \ldots, X_n) \in \mathbb{R}^d$ be a random vector with independent sub-gaussian components $X_i$ with $\mathsf{E}X_i = 0$. Let $\mathbf{D}$ be an $n \times n$ matrix. Then, for every $t \geq 0$, we have*

$$\mathbb{P}\left\{ \left| \mathbf{x}^\top \mathbf{D}\, \mathbf{x} - \mathsf{E}\left[ \mathbf{x}^\top \mathbf{D}\, \mathbf{x} \right] \right| > t \right\} \leq 2\exp\left[ -c\min\left( \frac{t^2}{\|\mathbf{D}\|_F^2}, \frac{t}{\|\mathbf{D}\|_{\mathrm{op}}} \right) \right],$$

*where $c$ is a constant that depends only on the subgaussian constants of $X_i$.*

### E.6. Stein's Lemma

We use the following simple lemma which is an application of integration by parts for Gaussian integrals.

**Lemma E.7** (Stein's Lemma). *Let $X$ be a random variables drawn from $\mathsf{N}(\mu, \sigma^2)$ and $g : \mathbb{R} \to \mathbb{R}$ be a differentiable function. We have $\mathsf{E}\left[ g(X)(X - \mu) \right] = \sigma^2 \mathsf{E}\left[ g'(X) \right]$.*

### E.7. Woodbury Matrix Identity

In the proofs, we use the following elementary identity which states that the inverse of a rank-$k$ correction of a matrix is equal to a rank-$k$ correction to the inverse of the original matrix.

**Theorem E.8** (Woodbury Matrix Identity (Woodbury, 1950)). *Let $\mathbf{A} \in \mathbb{R}^{n\times n}, \mathbf{C} \in \mathbb{R}^{k\times k}, \mathbf{U} \in \mathbb{R}^{n\times k}$, and $\mathbf{V} \in \mathbb{R}^{k\times n}$. The following matrix identity holds:*

$$(\mathbf{A} + \mathbf{U}\mathbf{C}\mathbf{V})^{-1} = \mathbf{A}^{-1} - \mathbf{A}^{-1}\mathbf{U}(\mathbf{C}^{-1} + \mathbf{V}\mathbf{A}^{-1}\mathbf{U})^{-1}\mathbf{V}\mathbf{A}^{-1},$$

*assuming that the inverse matrices in the expression exist.*

### E.8. Stieltjes Transform of Empirical Eigenvalue Distribution

For a distribution $\mu$ over $\mathbb{R}$, its Stieltjes transform is defined as

$$m_\mu(z) = \int \frac{d\mu(x)}{x - z}.$$

Let $H_d$ be the (discrete) empirical eigenvalue distribution of $\boldsymbol{\Sigma}_\mathbf{x} \in \mathbb{R}^{d\times d}$ and let $F_d$ be the (discrete) empirical eigenvalue distribution of the sample covariance matrix $\widehat{\boldsymbol{\Sigma}} \in \mathbb{R}^{d\times d}$. Consider the proportional limit where $d, n \to \infty$ with $d/n \to \phi > 0$. Suppose that the eigenvalue distribution $H_d$ converges to a limit population spectral distribution $H_{\boldsymbol{\Sigma}_\mathbf{x}}$; i.e., $H_d \Rightarrow H_{\boldsymbol{\Sigma}_\mathbf{x}}$ in distribution. Given the definition of $m(z)$ from equation (33), we have $m(z) = m_F(z)$. The following theorem characterizes $m_F$ in terms of $H_{\boldsymbol{\Sigma}_\mathbf{x}}$.

**Theorem E.9** (Silverstein Equation (Silverstein & Choi, 1995)). *Let $\nu_F(z) = \phi(m_F(z) + 1/z) - 1/z$. The function $\nu_F$ is the solution of the following fixed-point equation:*

$$-\frac{1}{\nu_F(z)} = z - \phi \int \frac{t}{1 + t\,\nu_F(z)} dH_{\boldsymbol{\Sigma}_\mathbf{x}}(t).$$

Thus, using this theorem, given $H_{\mathbf{\Sigma_x}}$, we can numerically compute $\nu_F$ (and hence, $m_F$) using fixed-point iteration. For example, for $\mathbf{\Sigma_x}^{(\varepsilon)}$ from equation (14), we have $F = 1/2\,\delta_{1-\varepsilon} + 1/2\,\delta_{1+\varepsilon}$.

# F. Additional Numerical Results and Details

## F.1. Details of the Experiment Setups

In the experiments, we generate $\mathbf{F}_0 \in \mathbb{R}^{d_Y \times k}$ with i.i.d. $\mathsf{N}(0, 1)$ entries. Then, for each task s we randomly draw a matrix $\mathbf{B_s} \in \mathbb{R}^{d_Y \times d_Y}$ and set $\mathbf{F_\star^s} = \exp\left(0.005(\mathbf{B_s} - \mathbf{B_s^\top})\right)\mathbf{F}_0$, where $\exp(\cdot)$ is the matrix exponential. The shared representation matrix $\mathbf{G_\star} \in \mathbb{R}^{k \times d_X}$ is generated by sampling uniformly from the space of row-orthonormal matrices in $\mathbb{R}^{k \times d_X}$.

We consider two settings for the covariance matrices $\mathbf{\Sigma_{x,s}}$; the *low-anisotropic*, and the *high-anisotropic* settings. In the low-anisotropic setting, we define $\mathbf{E} = 5\,\mathbf{I}_{d_X} + \mathbf{N}$ where $\mathbf{N} \in \mathbb{R}^{d_X \times d_X}$ has i.i.d. $\mathsf{N}(0, 1)$ entries, and set $\mathbf{\Sigma_{x,s}} = 0.5\,(\mathbf{E} + \mathbf{E}^\top)$. For the high-anisotropic setting, we first sample uniformly a rotation matrix $\mathbf{O} \in \mathbb{R}^{d_X \times d_X}$ and set $\mathbf{\Sigma_{x,,s}} = \mathbf{ODO}^\top$ where $\mathbf{D} = \mathrm{diag}(\texttt{logspace}(0, 5, d_X))$. In the experiments for the main paper, we always consider the high-anisotropic setting.

In the following experiments, in addition to the data generation process in equation (13) used in the experiment in the main paper, we also consider a Gaussian data setup where samples for task s are generated according to

$$\mathbf{y}_i^s = \mathbf{F_\star^s}\mathbf{G_\star}\mathbf{x}_i^s + \boldsymbol{\varepsilon}_i^s, \quad \mathbf{x}_i^s \sim \mathsf{N}(\mathbf{0}, \mathbf{\Sigma_{x,s}}), \quad \boldsymbol{\varepsilon}_i^s \overset{\text{i.i.d.}}{\sim} \mathsf{N}(0, \sigma_{\varepsilon,s}\mathbf{I}_{d_Y}), \quad \mathsf{s} \in \{\mathsf{test}, \mathsf{train}\}. \tag{38}$$

## F.2. Additional Experiments

### F.2.1. EFFECT OF BATCH NORMALIZATION

We used the same experiment setting described in Section 4 to generate the plots in Figure 9. Explicitly, we use data dimension $d_X = 100$, task dimension $d_Y = 15$, and representation dimension $k = 8$. We use the same learning rate $10^{-2}$ for each optimizer except for NGD, in which we used $10^{-4}$. The batch size is 1024. In Figure 3 we considered the Uniform data (13) with high anisotropy. Here, we consider the other three setting: Uniform data (13) with low anisotropy, Gaussian data (38) with low anisotropy, and Gaussian data (38) with high anisotropy.

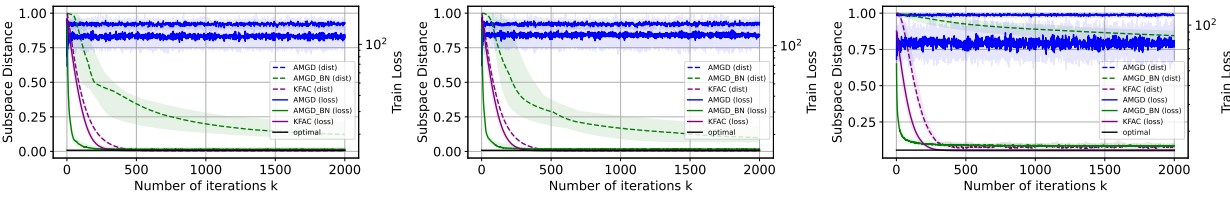

*Figure 4.* The effect of batch normalization (on AMGD) vs. KFAC in our experiment settings **(Left)** Uniform with low anisotropy. **(Middle)** Gaussian with low anisotropy. **(Right)** Gaussian with high anisotropy.

As discussed in the main paper Section 3.1.1, we expect AMGD with batch-norm to converge in training loss but to perform poorly with respect to the subspace distance from the optimal in settings in the case with high anistropy **(Right)**. However, in the experiment settings with low anisotropy **(Left and Center)**, we expect reasonable performance from this algorithm because $\mathrm{rowsp}(\mathbf{G_\star}\mathbf{\Sigma_x})$ is close to the target $\mathrm{rowsp}(\mathbf{G_\star})$.

### F.2.2. LEARNING RATE SWEEP

We further test the performance of each learning algorithm at different learning rates from $10^{-6}, 10^{-5.5}, \ldots, 10^{-0.5}, 10^0$, with results shown in Figure 5, where we plot the subspace distance at 1000 iterations for different algorithms. If the algorithm encounters numerical instability, then we report the subspace distance as the maximal value of $1.0$. We observe that KFAC and DFW coverage to a solution with small subspace distance to the true representation for a wide range of step sizes, whereas the set of suitable learning rates for other algorithms is much narrower. Furthermore, we observe the poor performance of various algorithms in Figure 1 and Figure 9 is not due to specific choice of learning rate.

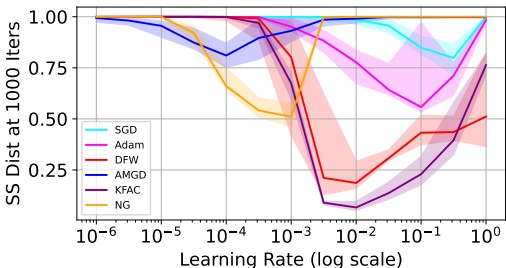

*Figure 5.* The subspace distance of representations learned by different algorithms after 1000 iterations and the true representation as a function of learning rate.

### F.2.3. HEAD-TO-HEAD EXPERIMENTS

We again consider the same experimental setting used for Figure 1. In particular, we use data dimension $d_x = 100$, task dimension $d_y = 15$, and representation dimension $k = 8$. We use the same learning rate $10^{-2}$ for each optimizer except for NGD optimizer, in which we used $10^{-4}$. The batch size is 1024. In Figure 1 we considered the Uniform data (13) with high anisotropy. Here, we consider the other three setting: Uniform data (13) with low anisotropy, Gaussian data (38) with low anisotropy, and Gaussian data (38) with high anisotropy. We plot the training loss, subspace distance to the ground truth shared representation, and the transfer loss obtained by different algorithms. See Figure 9. We observe that in all three settings, various algorithms converge in training loss. In the case with high anisotropy (second row), methods other than KFAC do not converge to the optimal representation in subspace distance and transfer loss. However, in the low anisotropy settings (first and third rows), the performance of other algorithms also improve, but are notably still suboptimal relative to KFAC, confirming the theoretical results showing that anisotropy is a root cause behind the sub-optimality of prior algorithms and analysis.

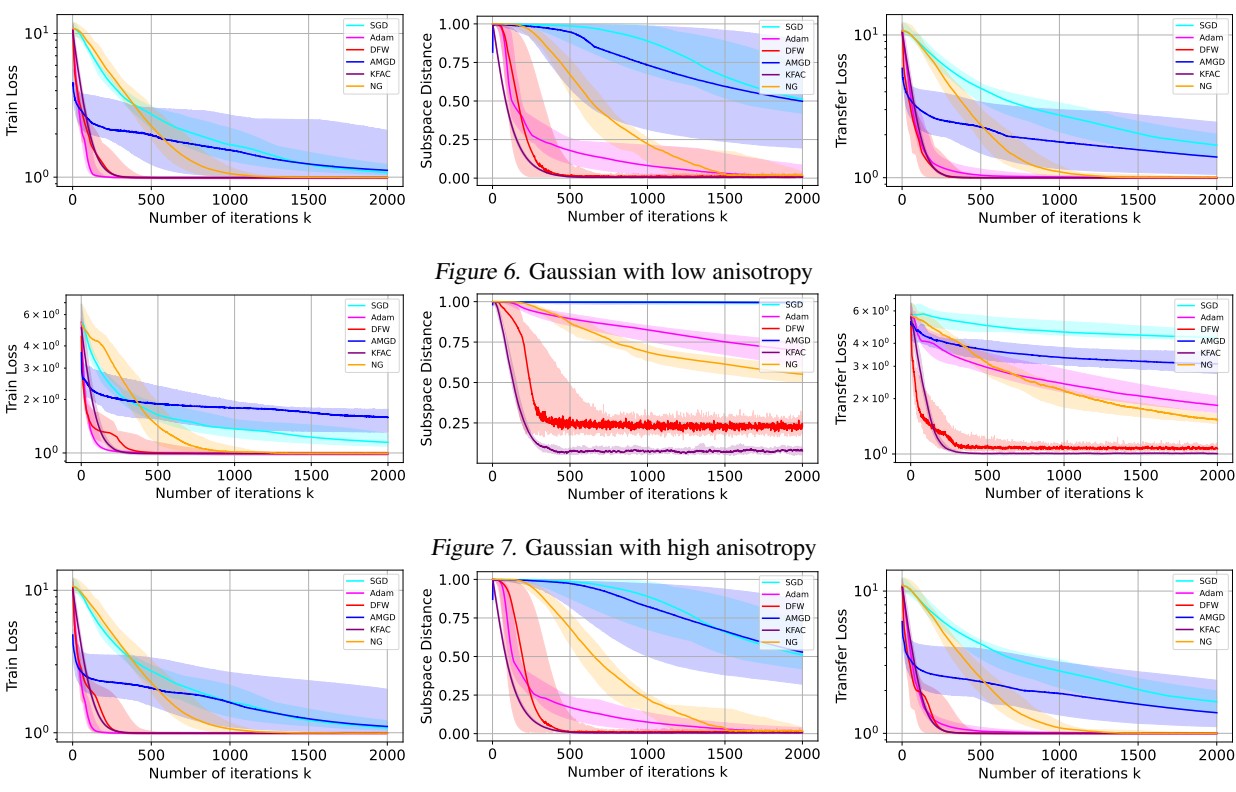

*Figure 6.* Gaussian with low anisotropy

*Figure 7.* Gaussian with high anisotropy

*Figure 8.* Bernoulli with low anisotropy

*Figure 9.* From **left** to **right**: the training loss, subspace distance, and transfer loss induced by various algorithms on a linear representation learning task.

