# OpenReview forum: "On The Concurrence of Layer-wise Preconditioning Methods and Provable Feature Learning"
_ICML.cc/2025/Conference — ICML 2025 poster_

### Official Review · Reviewer_aaxq · 2025-02-18

**Overall Recommendation:** 3

**Summary:**

The author show theoretically how, in the settings of linear representation and single index learning with data with non-trivial covariance structures using a two-layer network, SGD suffers from fundamental statistical biases and limitations, and how the use of a simple Kronecker-factored preconditioning scheme allows to overcome those issues. More precisely, in the case of linear representation learning, they establish how the preconditioned dynamics are endowed with condition number-free guarantees, in contrast to SGD, which is suboptimal. For single-index learning, they study the spike developed by the first layer weights following a single gradient step. They show that a SGD step leads to a biased spike compared to the target weights, while a preconditioned step does not. They corroborate their findings in numerical experiments, and include a comparison with Adam and batch-normalization schemes, which are suboptimal in the considered settings.

**Claims And Evidence:**

The theoretical claims of Theorem 3.9 are quantitatively verified in Fig. 2, where they are compared to numerical experiments, with convincing agreement. The theoretical insights on the suboptimality of SGD and benefits of KFAC are illustrated in numerical experiments in Figs. 1,3.

**Essential References Not Discussed:**

I did not identify any essential reference which the authors fail to discuss. However, I have limited familiarity with the literature on alternating or preconditioned descent methods, so it is possible I overlooked some works.

**Experimental Designs Or Analyses:**

I did not check the experimental designs in detail, however, the exposition of the setup provided in the main text is sufficiently detailed and I do not see any issue.

(Minor question) What is the value of $\lambda_G$ in Figs. 1,5,8? Do the qualitative behaviors of the curve depend on this parameter ?

**Methods And Evaluation Criteria:**

The paper is primarily theoretical in nature, and all experiments are conducted within the two stylized settings (linear representation task and single-index learning) considered in the theoretical parts.

**Other Comments Or Suggestions:**

I do not have particular comments or suggestions.

**Other Strengths And Weaknesses:**

I am overall in favor of acceptance. The results contained in the paper are interesting, as they highlight the fundamental limitations of vanilla SGD and how preconditioning schemes naturally emerge as pathways to mitigate these issues. Due to my limited expertise, I am unable to assess confidently the novelty or significance of the results presented in section 3.1. I have left a few questions and comments in the above sections.

**Questions For Authors:**

I do not have any particular question, beside those I listed in "Theoretical Claims" and "Experimental Design"

**Relation To Broader Scientific Literature:**

The alternating SGD scheme studied is related to algorithms appearing in (Collins et al., 2023) or (Zhang et al., 2024), but seems more general, although I do not have a large familiarity with this line of works.

The single-step analysis of Theorem 3.9 is a generalization of Proposition 2 and Theorem 3 of (Ba et al., 2022) to anisotropic covariates, which is to the best of my awareness a novel result.

**Theoretical Claims:**

From my reading, the claims look reasonable and sound, but I did not check the proofs.

I have minor concerns and questions regarding Proposition 3.5 and Theorem 3.6. In Theorem 3.6, the notation $\kappa$ and $\sigma_{min}$ are, unless I am mistaken, not introduced. I assume they refer to the condition number and the $\lambda_{min}$. In this case, would the range of application of Theorem 3.6 not also suffer from the conditioning number, as an increasingly close initialization to the ground truth is required? If this is the case, the statement under Theorem 3.6, although still valid, would gain to be nuanced by discussing this fact.
As a minor question, is it possible to reach a similar high-probability local convergence for SGD? Even if impossible or unclear, I believe it would aid to grasp better the comparison with KFAC, as for now Proposition 3.5 is a worst-case result. On the other hand, I do believe the experiments of Figs. 1,5,9 compellingly show the superiority of KFAC. This is meant as a minor suggestion.

---

> ### Author Rebuttal · Authors · 2025-03-29
>
> We thank the reviewer for their valuable feedback. We are glad that you found our results regarding the fundamental limitations of SGD and the natural emergence of preconditioning schemes as solutions to address these challenges interesting. Please find our detailed response to the comments and questions below.
>
> - **Defining $\kappa, \sigma_{\min}$ in notation.**
>
> We thank the reviewer for pointing this out. The parameters $\kappa$ and $\sigma_{\min}$ correspond to condition number and the smallest singular value respectively. We have added these to the revision.
>
> - **Dependence of the locality the requirement on $\kappa, \sigma_{\min}$.**
>
> The presence of $\kappa, \sigma_{\min}$ in the initialization requirement is rather subtle. Firstly, we note that locality requirements are distinct from the rate; an analogous bound for SGD in the isotropic setting will have both the initialization requirement *and* a rate that contains $\kappa(\mathbf F)$ [1, Thm 3.1-3.2], not to mention that SGD is non-convergent under anisotropy $\kappa(\Sigma_{\mathbf x}) \gg 1$ (cf. Figure 1). Secondly, similar requirements are ubiquitous in linear representation learning guarantees; however, it is generally believed that initialization is not required for this problem [2, Sec. 6], [3, Remark 3.2]. On the other hand, linear representation learning can be viewed as a low-rank matrix-sensing problem [1], and as far as we know there does not exist a general *global* convergence guarantee for that problem, let alone for the anisotropic sensing matrices our setting induces. We lastly remark that the initialization requirement does not appear in a stylized, noiseless setting $\sigma_{\varepsilon} = 0$, where we fix $\mathbf F_0$ at initialization and solely perform KFAC updates on $\mathbf G$. In particular, for each iteration $t$ we have
> $$
> \mathbf G_t = \mathbf R_0 \mathbf G_\star + \left(1-\eta_{\mathbf G}\right)^t \left(\mathbf G_0 - \mathbf R_0 \mathbf G_\star\right), \\;\\; \mathbf R_0 = (\mathbf F_0^\top \mathbf F_0)^{-1}\mathbf F_0 \mathbf F_\star.
> $$
> Thus, for $\eta_{\mathbf G}$ close to $1$, $\mathbf G_t$ converges linearly to $\mathbf R_0 \mathbf G_\star$, and for full-rank $\mathbf F_0$, $\mathrm{rowsp}(\mathbf R_0 \mathbf G_\star) = \mathrm{rowsp}(\mathbf G_\star)$.
>
>
> - **High-probability bound for SGD and role of Proposition 3.5.**
>
> We note that we identify two suboptimalities of SGD. Firstly, given anisotropic covariates, SGD can be non-convergent. However, even given isotropic covariates, SGD is unavoidably affected by the conditioning of the output layer $\mathbf F$. Convergence upper bounds analogous to Theorem 3.6 for SGD can be found in [1, 4]. The role of Proposition 3.5 is therefore to show that *even under extremely ideal conditions* such as isotropic inputs, SGD necessarily suffers when $\mathbf F_\star$ is ill-conditioned. This bound is indeed worst-case in that the adversary chooses $\mathbf G_0$, but should be interpreted as "SGD generally suffers from ill-conditioning", whereas KFAC's convergence rate is always condition-number-free. As noted after Proposition 3.5, an instance-specific SGD convergence rate *could* in principle be estimated and will almost surely involve a function of the spectrum of $\mathbf F_\star$, but is besides the point of our paper.
>
> - **Role of ridge parameter $\lambda_{\mathbf G}$ in linear representation experiments.**
>
> We have set $\lambda_{\mathbf G} \to 0$ in Figures 1, 5, and 8. We conduct additional experiments for the same experimental setup, but with varying degrees $\lambda_{\mathbf G}$ [(anon. link)](https://drive.google.com/file/d/1n6OmlzOpo0hRFp2YOwkHDun2bo63HjK3/view?usp=sharing). We observe that the regularization does not significantly alter the qualitative behaviors of the curves for a wide range of $\lambda_{\mathbf G}$.
>
> [1] https://arxiv.org/abs/2102.10217
>
> [2] https://arxiv.org/abs/2105.08306
>
> [3] https://arxiv.org/abs/2308.04428
>
> [4] https://arxiv.org/abs/2102.07078

---

> > ### Comment · Reviewer_aaxq · 2025-04-02
> >
> > I thank the reviewers for answering my questions and addressing my concerns. I wish to stand my evaluation and maintain my score.

---

### Official Review · Reviewer_AMdp · 2025-03-13

**Overall Recommendation:** 3

**Summary:**

The authors aim to study the effectiveness of layer-wise preconditioning methods with per-axis preconditioning. They show that the Kronecker-Factored structure naturally occurs for the problem of linear representation learning and single index learning. This provides a potential explanation for why these methods can outperform their exact second-order counterpart.

## update after rebuttal

In the rebuttal, the authors state that they avoid making general claims about Kronecker-Factored preconditioning being better than element-wise/diagonal preconditioning for feature learning, or that factored curvature estimates always lead to faster convergence than the dense versions.

If that is the case, then I believe some parts of the writing can be improved to avoid giving out this impression. For example, in the introduction section, they first state that KFAC generally outperforms its ideal counterpart NGD and then state that they want to give explanation to the performance of Kronecker-Factored methods. This could give the impression that they want to claim evidence for KFAC better than not only SGD but also NGD.

In either case, this means the paper only try to show that Kronecker-Factored methods are better than SGD, which is solid work, but also much more expected so I keep more original score.

I also verified the authors claim about rowspan is correct. I was originally thinking about left-multiplication instead of right multiplication.

**Claims And Evidence:**

It feels like some of the claims are only supported indirectly without being articulated clearly. Specifically, two of the more interesting claims,

1. Kronecker-Factored structured preconditioning is necessary and thus element-wise preconditioning like Adam is not enough,

2. There are theoretical justification for why Kronecker-Factored structured preconditioning can outperform their exact second-order counterpart,

are discussed very little in the paper. Most of the main paper is about how SGD can fail, which is more expected and less interesting to the readers.

**Essential References Not Discussed:**

I do not find any essential references that are missed by the authors.

**Experimental Designs Or Analyses:**

I do not find any issues with their analyses.

**Methods And Evaluation Criteria:**

The evaluation setting is quite simple so that it aligns with the analyzed theoretical setting. Since the purpose of the experiments is just to verify the theory, this seems reasonable.

**Other Comments Or Suggestions:**

I have no other suggestions.

**Other Strengths And Weaknesses:**

The writing seems to be a weakness of the paper. Some of the claims seem not to be clearly articulated.

**Questions For Authors:**

1. I am not sure if the equality about rowspan on line 302 is correct. It seems to me that right multiplication with a matrix will not change the rowspan. (The authors might be considering some noise term to the solution of G, which cannot be clearly seen from the equation)

**Relation To Broader Scientific Literature:**

The paper provides potential explanations to the empirical success of Kronecker-Factored preconditioning methods. This might lead to the development of new optimization methods.

**Theoretical Claims:**

I do not find any issues with their theoretical claims.

---

> ### Author Rebuttal · Authors · 2025-03-29
>
> We thank the reviewer for their valuable comments. We are glad that you find that these results provide explanations for the empirical success of Kronecker-Factored preconditioning methods, and that it can potentially guide algorithm design. Below please find our response to the comments and questions.
>
> - **Indirect support of certain claims.**
>
> We would like to clarify that the main goal of our paper is to address key issues of relying on SGD in feature learning theory, and in doing so establish a feature-learning motivation for layer-wise preconditioned optimizers. We avoid making general claims about Kronecker-Factored preconditioning being better than element-wise/diagonal preconditioning for feature learning, or that factored curvature estimates always lead to faster convergence than the dense versions. Rather, specific observations of these phenomena in prior literature are what *motivate* our study of Kronecker-Factored preconditioning. We provide some additional information and context for each.
>
> 1. **KF preconditioning is necessary vs element-wise.**
>
> Current literature notably lacks work showing provable feature learning capabilities of diagonal preconditioning methods like Adam, despite a huge amount of work studying SGD in this context. Anecdotally, this is because the entry-wise (non-linear) operations that Adam performs are somewhat unamenable to analyze given the linear-operator structure of weights in neural networks. Therefore, by straightforwardly deriving feature learning properties of a representative Kronecker-Factored method, we see our work as evidence that they are a more natural NN-oriented optimizer class. On the other hand, we note diagonal preconditioning is *not* a strict subset of Kronecker-Factored preconditioning: for a given entrywise product $\mathbf M \odot \mathbf G$, generally there does not exist $\mathbf P, \mathbf Q$ such that $\mathbf M \odot \mathbf G = \mathbf P \mathbf G \mathbf Q$, which is why we do not claim that diagonal preconditioning is generally worse than KF preconditioning for feature learning, even if KF preconditioning may be more mathematically natural.
>
> 2. **Outperforming dense 2nd-order variant.**
>
> As stated in our introduction, many deep learning optimizers, including Adagrad (Adam), KFAC, Shampoo etc. were initially derived as computationally-efficient approximations to full 2nd-order methods like (Gauss-)Newton method and Natural Gradient Descent. Therefore, a natural conclusion is that *if one had the resources*, the full 2nd-order method should yield better convergence. However, the behavior of full 2nd-order methods on neural networks is notoriously poorly understood from theory, with basic questions such as "are negative curvature directions (negative Hessian eigs) in Newton's method good?" and "is Gauss-Newton (guaranteed psd) preferable to Newton?" having no conclusive answers [1]. Therefore, our contribution can be seen as *directly* deriving the benefits of the approximating method *without* reasoning about the full 2nd-order method. The fact that the approximant numerically *outperforms* the full method is only further evidence that this is a fruitful path of analysis, though we note this observation is not original to us [2].
>
> - **Correctness of rowspan.**
>
> The math in line 302 should be correct. To see how right-multiplication by a matrix can change the rowspace, consider the simple example $\mathbf G = [1 \quad 0] \in \mathbb R^{1 \times 2}$, whose rowspace is  $\{[c \quad 0], c \in \mathbb R\}$. Multiplying by a (psd) matrix $\Sigma = \begin{bmatrix}2 & 1 \\\\ 1 & 2 \end{bmatrix}$ yields $\mathbf G \Sigma = [2\quad 1]$, which is not contained in $\mathrm{rowsp}(\mathbf G)$. To tie this back to batchnorm, our observation is that whitening the training-distribution covariates allows SGD to converge (since the covariates are now isotropic) but changes the rowspace, breaking the shared structure between the source and transfer distributions.
>
> [1] https://arxiv.org/abs/1503.05671
>
> [2] https://arxiv.org/abs/2201.12250

---

### Official Review · Reviewer_NYms · 2025-03-14

**Overall Recommendation:** 3

**Summary:**

This paper demonstrates KFAC is better at feature learning than vanilla SGD with two model examples. For linear representation learning, they show the convergence rate of SGD will suffer from the condition number while KFAC gets rid of it. For single-index learning, they show that one-step update of SGD can only learn the correct direction with isotropic data while the one-step update of KFAC can learn the direction under isotropy.  They conduct experiments to show the effectiveness of their theory.

**Claims And Evidence:**

The author claims they show SGD can be drastically slow on linear representation learning under anisotropy. But I only saw they cite previous papers on this claim and proposition 3.5 assumes isotropy. It can not viewed as one contribution by this paper without further explanation.

**Essential References Not Discussed:**

NA

**Experimental Designs Or Analyses:**

I checked the experiment setting and results and didn’t find any serious issue.

**Methods And Evaluation Criteria:**

The two representation learning models are commonly used in this area. There are two synthetic experiments for each models. They are already reasonable but I am hoping for some real world dataset.

**Other Comments Or Suggestions:**

1. It’s better to explicit show each algorithm with some algorithm block and keep a consistent notation for $G_+$. It is defined differently in (5) and (8) and looks very confusing every time it is mentioned later. For example, I think it should be $\bar{G}_+$ in the LHS of (9).
2. Typo in line 280: $F_\star^{(t)}$ should be $F_{ls}^{(t)}$?
3. Typo in line 401: The distribution of $x_i^s$ seems unfinished.

**Other Strengths And Weaknesses:**

Strength:
1. This paper makes a good connection between feature learning and kronecker-factored algorithms.
2. The theoretical claims and proof look well-written and relatively rigorous.

Weakness:
1. The theoretical results are only for one specific layer-wise preconditioning method. Since Shampoo is a more popular kronecker-factored optimization algorithm than KFAC, it should be more interesting to get the results for Shampoo.
2. Different $P_G$ are used for KFAC in the two models. It would be better if lemma 3.12 is proved with unregularized $P_G$.
3. The negative result is only shown for SGD. As mentioned in the introduction, it is unknown why kronecker-factored algorithm can be better than the idealized second-order methods. If we want to claim the reason might be KFAC is better at feature learning, then the negative results of second-order methods are expected.

**Questions For Authors:**

1. I am not convinced why we only care about the one-step update in the single index learning. Even if the one-step update of SGD is bad, we almost never update the parameter only for once so SGD might be able to learn the correct direction after several steps.
2. Do you try Shampoo for the first experiment? If the theoretical results can not be easily extended to Shampoo, I am curious whether Shampoo can learn features well in practice. That can help understand whether the connection to feature learning is applicable to a broader class of kronecker-factored algorithms rather than only KFAC.
3. I am also curious whether there can be any negative theoretical results for Adam. From figure 1, Adam can learn features much better than any other algorithms and is only slightly worse than KFAC. If there is no obvious negative theoretical result, is there any insight why Adam can be good at learning features?
4. As mentioned in weakness 2, I wonder there is any obstacle in proving lemma 3.12 with unregularized $P_G$.

**Relation To Broader Scientific Literature:**

It can help understand the difference between optimization algorithms and why some algorithms can succeed on some tasks.

**Theoretical Claims:**

I roughly checked the proof of proposition 3.5 and didn’t find any obvious mistakes.

---

> ### Author Rebuttal · Authors · 2025-03-29
>
> We thank the reviewer for their valuable comments. We are glad that you find our results rigorous, and that our results make a good connection between feature learning and KF algorithms.
>
> - **Anisotropy vs ill-conditioning.**
>
> We clarify that we make two distinct claims in Section 3.1. We identify *two* sources of suboptimality in SGD for linear rep. learning: the first sourced from the anisotropy of the inputs, and the second coming from the ill-conditioning of the output layer $\mathbf F$. The bias from anisotropy is critical, and can prevent convergence *altogether* (as seen in AMGD/SGD lines in Figure 1). As pointed out in our literature review, this observation has escaped notice, apart from a recent work [1], whose algorithm is a special case of our recipe in Eqs (4) and (5); see DFW in Fig. 1. The second issue is what the lower bound in Prop. 3.5 concerns. We assume isotropy therein (and multiple favorable assumptions) to make the problem *as benign as possible* for SGD, which we showed does not converge without isotropy. The point of Prop. 3.5 is that *even under ideal conditions* the convergence rate of SGD and related methods is suboptimal compared to KFAC as shown in Theorem 3.6. We hope this clarifies our separate contributions.
>
> - **Real-world experiments.**
>
> We believe it is well-documented that KFAC performs well in practice (see e.g. [2, Sec. 13] and [3]). In fact, KFAC was designed as a *practical* approximation of the Natural Gradient method. Our main goal is to resolve the limitations of SGD in prominent feature learning theory setups. As a byproduct, we derive KFAC as the natural solution, providing an alternate *feature-learning* justification for KFAC. Our experiments are solely to verify and complement our theoretical findings.
>
> - **Relevance of Shampoo.**
>
> We are aware that methods such as Shampoo and Muon have received a lot of attention lately. However, we should note that up until last year, KFAC has been largely the more well-known method, seeing many extensions and applications. We also mention that making Shampoo outcompete the Adam family (e.g. the winning submission in AlgoPerf) takes additional heuristics like learning-rate grafting [4, Sec. 2.4]. However, **we have implemented basic versions of Shampoo**, and run it in the set-up for Figure 1 [(anon link)](https://drive.google.com/file/d/1MS4PjZXTXVidzvzjmoM1P1X7mKz7_Ih6/view?usp=sharing). SHMP(G) corresponds to updating $\mathbf G$ with Shampoo, and $\mathbf F$ with least-squares (like AMGD and KFAC), and SHMP is Shampoo on both layers. We observe that Shampoo performs roughly on par with NGD, but not as well as KFAC. This is expected, since we derived KFAC to provably get *constant-factor* linear convergence. We believe that exploring the feature learning properties of Shampoo is an interesting future direction.
>
> - **Regularized $\mathbf P_{\mathbf G}$.**
>
> In the single-index setting, it is more natural to consider regularized $\mathbf P_{\mathbf G}$ because analyses are traditionally in the proportional limit $d_x \asymp n$, where a non-zero regularizer is known to be optimal. In fact, the result we provide (Lemma 3.10 and 3.12) are more general and hold even for $\lambda_{\mathbf G} \to 0$. In Figure 3 (Left), we still see that for $\lambda_{\mathbf G} \to 0$, our theory matches the simulations, and that the alignment of the direction learned by KFAC to the true direction is still significantly larger than that of SGD.
>
> - **Negative result for 2nd-order methods.**
>
> We clarify that the motivation for our paper is that many algorithms are designed to approximate 2nd-order methods, but neural network analyses for 2nd-order methods are still lacking. Therefore, our goal is not to debunk 2nd-order methods, but to suggest a direct motivation for the approximation methods themselves. We refer the reviewer to **Outperforming dense 2nd-order variant** of our response to Reviewer AMdp for more discussions.
>
> - **Algorithm block and $\overline{\mathbf G}$.**
>
> We thank the reviewer for pointing these out. They have been fixed in the revision.
>
> - **One giant step vs multi-step.**
>
> The one-step update for learning a single index model is a popular framework in deep learning theory and is the state of the art model to theoretically analyze feature learning properties of two-layer networks (cf. line 283 in the submission). In this setting, features are learned by taking a single (giant) step on the first layer as opposed to multiple small steps. A single step can often be shown to be equivalent to taking multiple steps with smaller step sizes (Section B.1.3 of [5]).
>
> - **Is Adam a good feature learner?**
>
> For insights on why Adam has fundamental limitations in our feature learning setting, we refer to **Adam in transfer learning settings.** in our response to Reviewer RnJ3.
>
> [1] arxiv.org/abs/2308.04428
>
> [2] arxiv.org/abs/1503.05671
>
> [3] arxiv.org/abs/2311.00636
>
> [4] arxiv.org/abs/2309.06497
>
> [5] arxiv.org/abs/2205.01445

---

### Official Review · Reviewer_RnJ3 · 2025-03-18

**Overall Recommendation:** 4

**Summary:**

This paper shows that layer-wise preconditioning is statistically necessary for efficient feature learning using two common models: linear representation learning and single-index learning. They prove that SGD struggles in non-isotropic inputs, and demonstrate theoretically and experimentally that this suboptimality is fundamental. They show that layer-wise preconditioning naturally addresses this issue. In addition, they show experimentally that Adam and batch normalization help only slightly, and that layer-wise preconditioning is uniquely beneficial even compared to the actual second-order methods that it is approximating.

**Claims And Evidence:**

The main claims about layer-wise preconditioners versus SGD are clear and well-supported by theoretical analysis, with well-explained intuition and implications. Overall, it seems to be a solid contribution.

Perhaps one limitation would be that the experiments are somewhat limited, and while the paper shows that Adam (diagonally preconditioned methods) and batch norm are insufficient, it lacks deeper intuition or analysis on why that’s the case.

**Essential References Not Discussed:**

I find the related work pretty well cited in general.

I have seen in the following work that Adam can outperform SGD when there is a class imbalance in the data, which could be related to the problem condition. Would be interesting to see the discussion on whether this is related (but feel free to use your judgment).

[F. Kunstner et.al, Neurips 2024. Why Adam Outperforms Gradient Descent on Language Models: A Heavy-Tailed Class Imbalance Problem]

**Experimental Designs Or Analyses:**

I didn't rerun the experiments to check correctness, they seem reasonable. My complaint is that the experiments seem to be limited to small-scale examples and transfer learning settings. Though I understand empirical results are not the main focus of this paper, it would be very interesting to see if these observations hold in bigger, more practical cases.

**Methods And Evaluation Criteria:**

The theoretical examples and analysis and interesting and I believe appropriate for the question. Experimentally, the subspace distance seems to correlate well with whether transfer learning is successful.

**Other Comments Or Suggestions:**

No

**Other Strengths And Weaknesses:**

Other strength: The paper is educational and well-written, providing a clear discussion of optimization history and why the problem is well-motivated.

Other weakness: The analysis is limited to specific settings, though this is fairly common in theoretical work. Additionally, the experiments are somewhat limited in scope.

**Questions For Authors:**

The experiments that contain Adam seem to be limited to the transfer learning settings. Is there any intuition on why layer-wise preconditioning is better than diagonal (Adam) in this setting? Is there a difference in general, outside of the transfer learning setting?

**Relation To Broader Scientific Literature:**

It’s observed in existing literature and practice that K-FAC-like methods (Shampoo for example) can outperform SGD in training and generalization. While this paper’s theoretical analysis is limited to Gaussian data and small-scale tasks (e.g., two-layer networks for linear representation learning and single-index models), it’s still a valuable step toward understanding the limitations of (S)GD and the necessity of layer-wise preconditioning. The paper also briefly touches on why Adam's advantage is limited in such settings, though more in-depth study of that could be helpful.

**Theoretical Claims:**

I didn't check the proofs line by line, but they seem reasonable. The authors explained the intuitions and implications of each lemma/theorem well.

---

> ### Author Rebuttal · Authors · 2025-03-29
>
> We thank the reviewer for their valuable comments. We are very glad to see that you find our paper to be a solid contribution  and a valuable step toward understanding the limitations of SGD and the necessity of layer-wise preconditioning. Below please find our response to the comments and questions.
>
> - **More intuition for insufficiency of Adam and Batchnorm.**
>
> Regarding batchnorm, the discussion following Lemma 3.7 can be mathematically formalized. We will reformat this to be a separate lemma that demonstrates how batchnorm/whitening causes convergence to the wrong feature space $\mathrm{rowsp}(\tilde{\mathbf G}) \triangleq \mathrm{rowsp}(\mathbf G_\star \Sigma_{\mathbf x}^{1/2})$, and further provide a simple estimate of $\mathrm{dist}(\tilde{\mathbf G}, \mathbf G_\star)$ in terms of the anistropy of $\Sigma_{\mathbf x}$.
>
> Regarding the insufficiency of Adam, one of the motivating factors to this paper is that Adam has proven difficult to analyze in *feature learning* contexts, i.e. fine-grained models of neural-network learning, thus motivating the search and analysis of optimizers that are naturally amenable to the compositional / layered structure of neural networks (see e.g. Eq (1)). In our opinion, the reason Adam is not optimal even in our idealized neural network learning settings is that diagonal preconditioning is not optimal for dealing with sources of ill-conditioning arising from the compositional structure of the parameter-space, e.g. $f_{\mathbf F, \mathbf G}(x) = \mathbf F \mathbf G x$. Since Adam is a diagonal preconditioner, it necessarily suffers from these issues. Furthermore, Adam takes a "-1/2" power of its curvature estimate, rather than the full inverse, which necessarily slows its convergence for cases where curvature estimates are reliable (though certainly has other practical merits). Initial probes into the relative effectiveness of diagonal preconditioning have been proposed in concurrent studies, see e.g. https://nikhilvyas.github.io/SOAP_Muon.pdf, https://arxiv.org/abs/2411.12135 (cf. Section 4.3).
>
> - **Related work showing that Adam outperforms SGD when there is class imbalance.**
>
> We thank the reviewer for pointing out this paper. We will add it to the related works section.
>
> - **Adam in transfer learning settings.**
>
> We emphasize that our transfer learning set-up is a proxy for more robust notions of generalization, which allows us to diagnose the quality of solutions returned by various algorithms in a more informative way than comparing training loss convergence. Not accounting for curvature (e.g. SGD) or using weaker forms of it (Adam) are therefore exposed by either slow convergence rate overall or by poor "generalization" (exposed by poor convergence in subspace distance); see additional plots [(anon. link)](https://drive.google.com/file/d/1yitFmd-Kvqr_bBmrb_L7OhCKQd9RUMyC/view?usp=sharing). Notably, we believe the latter notion is a promising way to diagnose whether curvature-aware/adaptive descent methods capture the geometry of neural-network optimization correctly. For example, despite Adam being an adaptive gradient method, it exhibits poor subspace distance recovery due to still being biased by the (spurious) curvature introduced by the anisotropy of the covariates (compare e.g. to low anisotropy settings in Figure 8).
>
> The essence of why Adam (or diagonal preconditioners) is a suboptimal feature-learner is the following: many adaptive methods are able to reduce training loss quickly by fitting the "low-frequency" features that explain most of the data. Therefore, many candidate solutions can attain low training loss. However, the "high-frequency" directions/features that explain less of the *training* objective correspond to *sharper* curvature; weak preconditioners have trouble smoothing these directions, and thus make very slow progress therein, reflected by the subspace distance plot. Mild distribution shifts (e.g. transfer learning) can lead to significant shift in which features are relevant, and thus of the many candidate solutions that get low training loss, it is important to also accurately detect all the possible relevant features like KFAC does in this setting.

---

### Decision · Program_Chairs · 2025-05-01

**Decision:**

Accept (poster)

**Comment:**

Dear Authors,

Thank you for submitting your paper to ICML and for contributing a theoretically grounded and well-motivated analysis of layer-wise preconditioning in the context of feature learning. Your work explores the statistical and algorithmic advantages of Kronecker-factored preconditioning methods, demonstrating that they uniquely resolve fundamental limitations of SGD.

The reviewers generally agreed that this is a well-written and impactful contribution to our understanding of optimization in deep learning. Several noted that your theoretical results were rigorous and complemented by clearly designed experiments on certain stylized tasks. While some reviewers expressed interest in further empirical validation on larger or real-world datasets, the conceptual clarity and theoretical insights were recognized as a core strength. The rebuttal further addressed reviewer questions about generalization to Shampoo, the role of initialization, and the scope of your claims regarding second-order methods, preconditioners etc.

Given the strength of the theoretical contributions, the clarity of exposition, and the positive consensus among reviewers, I am happy to recommend acceptance to ICML 2025. I encourage you to use the camera-ready version to further refine the exposition, clarify your scope of claims (particularly relative to full second-order methods), and highlight directions for extending the work to broader optimizer families or empirical settings.

Congratulations on your contribution.

Best regards,
AC